# Bias in Evaluation Processes:
# An Optimization-Based Model

**L. Elisa Celis**
Yale University

**Amit Kumar**
IIT Delhi

**Anay Mehrotra**
Yale University

**Nisheeth K. Vishnoi**
Yale University

## Abstract

Biases with respect to socially-salient attributes of individuals have been well documented in evaluation processes used in settings such as admissions and hiring. We view such an evaluation process as a transformation of a distribution of the true utility of an individual for a task to an observed distribution and model it as a solution to a loss minimization problem subject to an information constraint. Our model has two parameters that have been identified as factors leading to biases: the resource-information trade-off parameter in the information constraint and the risk-averseness parameter in the loss function. We characterize the distributions that arise from our model and study the effect of the parameters on the observed distribution. The outputs of our model enrich the class of distributions that can be used to capture variation across groups in the observed evaluations. We empirically validate our model by fitting real-world datasets and use it to study the effect of interventions in a downstream selection task. These results contribute to an understanding of the emergence of bias in evaluation processes and provide tools to guide the deployment of interventions to mitigate biases.

## 1 Introduction

Evaluation processes arise in numerous high-stakes settings such as hiring, university admissions, and fund allocation decisions [20, 30, 90, 122]. Specific instances include recruiters estimating the hireability of candidates via interviews [121, 30], reviewers evaluating the competence of grant applicants from proposals [147, 18], and organizations assessing the scholastic abilities of students via standardized examinations [99, 19]. In these processes, an evaluator estimates an individual's value to an institution. The evaluator need not be a person, they can be a committee, an exam, or even a machine learning algorithm [51, 122, 145]. Moreover, outcomes of real-world evaluation processes have at least some uncertainty or randomness [30, 18, 79]. This randomness can arise both, due to the features of the individual (e.g., their test scores or grades) that an evaluator takes as input [31, 76, 124], as well as, due to the evaluation process itself [50, 30, 140].

Biases against individuals in certain disadvantaged groups have been well-documented in evaluation processes [147, 74, 104, 110, 30]. For instance, in employment decisions and peer review, women receive systematically lower competence scores than men, even when qualifications are the same [147, 110], in standardized tests, the scores show higher variance in students from certain genders [21, 112], and in risk assessment–a type of evaluation–widely used tools were twice as likely to misclassify Black defendants as being at a high risk of violent recidivism than White defendants [6]. Here, neither the distribution of individuals' true evaluation depends on their socially-salient attributes nor is the process trying to bias evaluations, yet biases consistently arise [147, 74, 104, 110, 30]. Such evaluations are increasingly used by ML systems to learn or make decisions about individuals, potentially exacerbating inequality [51, 122, 145]. This raises the question of explaining the emergence of biases in evaluation processes which is important to understand how to mitigate them, and is studied here.

37th Conference on Neural Information Processing Systems (NeurIPS 2023).

**Related work.** A wide body of work has studied reasons why such differences may arise and how to mitigate the effect of such biases [75, 57, 54, 32, 88, 28]. For one, socioeconomic disadvantages (often correlated with socially-salient attributes) have been shown to impact an individual's ability to perform in an evaluation process, giving rise to different performance distributions across groups [55, 16]. Specifically, disparities in access to monetary resources are known to have a significant impact on individuals' SAT scores [55]. Moreover, because of differences between socially-salient attributes of individuals and evaluators, the same amount of resources (such as time or cognitive effort) spent by the evaluator and the individual, can lead to different outcomes for individuals in different groups [64, 95, 8]. For instance, it can be cognitively more demanding, especially in time-constrained evaluation processes, for the evaluator to interact with individuals who have a different cultural background than them, thus impacting the evaluations [95, 86, 70, 58, 142, 115, 113]. Further, such biases in human evaluations can also affect learning algorithms through biased past data that the algorithms take as input [75, 57, 54, 28].

Another factor that has been identified as a source of bias is "risk averseness:" the tendency to perceive a lower magnitude of increase in their utility due to a profit than the magnitude of decrease in their utility due to a loss of the same magnitude as the profit [84, 144, 151]. Risk averseness is known to play a role in high-stakes decisions such as who to hire, who to follow on social networks, and whether to pursue higher education [71, 23, 14]. In evaluations with an abundance of applicants, overestimating the value of an individual can lead to a downstream loss (e.g., because an individual is hired or admitted) whereas under-estimating may not have a significant loss [139, 65]. Thus, in the presence of risk averseness, the outputs of evaluation processes may skew the output evaluations to lower or higher values. The same skew can also arise from the perspective of individuals [13, 33, 111]. For instance, when negotiating salaries, overestimating their salary can lead to adverse effects in the form of evaluators being less inclined to work with the individual or in extreme cases denying employment [13, 33]. Moreover, these costs have been observed to be higher for women than for men, and are one of the prominent explanations for why women negotiate less frequently [13, 33].

A number of interventions to mitigate the adverse effects of such biases in evaluation processes have been proposed. These include representational constraints that, across multiple individuals, increase the representation of disadvantaged and minority groups in the set of individuals with high evaluations [46, 135, 131, 19, 77, 27, 116], structured evaluations which reduce the scope of unintended biases in evaluations [123, 68, 147, 15], and anonymized evaluations that, when possible, blind the decision makers to the socially-salient attributes of individuals being evaluated [72].

Mathematically, some works have modeled the outcomes of evaluation processes based on empirical observations [12, 22, 90, 61]. For instance, the implicit variance model of [61] models differences in the amount of noise in the utilities for individuals in different groups. Here, the output estimate is drawn from a Gaussian density whose mean is the true utility $v$ (which can take any real value) and whose variance depends on the group of the individual being evaluated: The variance is higher for individuals in the disadvantaged group compared to individuals in the advantaged group. Additive and multiplicative skews in the outputs of evaluation processes have also been modeled [90, 22] (also see Appendix A). [90] consider true utilities $v > 0$ distributed according to the Pareto density and they model the output as $v/\rho$ for some fixed $\rho \geq 1$; where $\rho$ is larger for individuals in the disadvantaged group. These models have been influential in the study of various downstream tasks such as selection [90, 61, 38, 129, 67, 106, 108, 29], ranking [40], and classification [28] in the presence of biases.

**Our contributions.** We propose a new optimization-based approach to model how an evaluation process transforms an (unknown) input density $f_{\mathcal{D}}$ representing the true utility of an individual or a population to an observed distribution in the presence of information constraints or risk aversion. Based on the aforementioned studies and insights in social sciences, our model has two parameters: the resource-information parameter ($\tau \in \mathbb{R}$) in the information constraint and the risk-averseness parameter ($\alpha \geq 1$) in the objective function; see (OptProg) in Section 2. The objective measures the inaccuracy of the estimator with respect to the true density $f_{\mathcal{D}}$, and involves a given loss function $\ell$ and the parameter $\alpha - \alpha$ is higher (worse) for individuals in groups facing higher risk aversion. The constraint places a lower bound of $\tau$ on the amount of information (about the density of the true value $v$) that the individual and evaluator can acquire or exchange in their interaction $- \tau$ is higher for individuals in groups that require more resources to gain unit information. We measure the amount of information in the output density by its differential entropy. Our model builds on the maximum-entropy framework in statistics and information theory [80] and is derived in Section 2 and can be viewed as extending this theory to output a rich family of biased densities.

In Section 3, we show various properties of the output densities of our model. We prove that the solution to (OptProg) is unique under general conditions and characterize the output density as a function of $f_\mathcal{D}$, $\ell$, $\tau$, and $\alpha$; see Theorem 3.1. By varying the loss function and the true density, our framework can not only output standard density functions (such as Gaussian, Pareto, Exponential, and Laplace), but also their appropriate "noisy" and "skewed" versions, generalizing the models studied in [90, 22, 61]. Subsequently, we investigate how varying the parameter $\tau$ affects the output density in Section 3. We observe that when $\tau \to -\infty$, there is effectively no constraint, and the output is concentrated at a point. For any fixed $\alpha$, as $\tau$ increases, the output density *spreads*–its variance and/or mean increases. We also study the effect of increasing $\alpha$ on the output density. We observe that when the true density is Gaussian or Pareto, the mean of the output density decreases as $\alpha$ increases for any fixed $\tau$. Thus, individuals in the group with higher $\alpha$ and/or $\tau$ face higher noise and/or skew in their evaluations as predicted by our model.

Empirically, we evaluate our model's ability to emulate biases present in real-world evaluation processes using two real-world datasets (JEE-2009 Scores and the Semantic Scholar Open Research Corpus) and one synthetic dataset (Section 4). For each dataset, we report the total variation (TV) distance between the densities of biased utilities in the data and the best-fitting densities output by our framework and earlier models. Across all datasets, we observe that our model can output densities that are close to the density of biased utilities in the datasets and has a better fit than the models of [61, 90]; Table 1. Further, on a downstream selection task, we evaluate the effectiveness of two well-studied bias-mitigating interventions: equal representation (ER) and proportional representation (PR) constraints, and two additional interventions suggested by our work: decreasing the resource-information parameter $\tau$ and reducing the risk-averseness parameter $\alpha$. ER and PR are constraints on the allowable outcomes, $\tau$ can be decreased by, e.g., training the evaluators to improve their efficiency, and $\alpha$ can be decreased using, e.g., structured interviews [30]. We observe that for each intervention, there are instances of selection, where it outperforms all other interventions (Figure 1). Thus, our model can be used as a tool to study the effectiveness of different types of interventions in downstream tasks and inform policy; see also Appendix L.1 and Appendix B.

## 2 Model

The evaluation processes we consider have two stakeholders–an evaluator and an individual–along with a societal context that affects the process. In an evaluation process, an evaluator interacts with an individual to obtain an estimate of the individual's utility or value. We assume that each individual's true utility $v$ is drawn from a probability distribution. This not only captures the case that the same individual may have variability in the same evaluation (as is frequently observed in interviews, examinations, and peer-review [31, 50, 76, 124, 30, 140, 18]) but also the case that $v$ corresponds to the utility of an individual drawn from a population. For simplicity, we consider the setting where $v$ is real-valued and its density is supported on a continuous subset $\Omega \subseteq \mathbb{R}$. This density gives rise to a distribution over $\Omega$ with respect to the Lebesgue measure $\mu$ over $\mathbb{R}$. For instance, $\Omega$ could be the set of all real numbers $\mathbb{R}$, the set of positive real number $\mathbb{R}_{>0}$, an open interval such as $[1, \infty)$, or a closed interval $[a, b]$. Following prior work modeling output densities [90, 40, 61], we assume that the true utility of all individuals is drawn from the *same* density $f_\mathcal{D}$.

We view an evaluation process as a transformation of an (unknown) true density $f_\mathcal{D}$ into an observed density $f_\mathcal{E}$ over $\Omega$. In real-world evaluation processes, this happens through various means: by processing features of an individual (e.g., past performance on exams or past employment), through interaction between the evaluator and the individual (e.g., in oral examinations), or by requesting the individual to complete an assessment or test [121, 99, 19]. We present an optimization-based model that captures some of the aforementioned scenarios and outputs $f_\mathcal{E}$. The parameters of this model encode factors that may be different for different socially-salient groups, thus, making $f_\mathcal{E}$ group dependent even though $f_\mathcal{D}$ is not group dependent. We derive our model in four steps.

**Step 1: Invoking the entropy maximization principle.** In order to gain some intuition, consider a simple setting where the utility of an individual is a fixed quantity $v$ (i.e., $f_\mathcal{D}$ is a Dirac-delta function around $v$). We first need to define an *error* or loss function $\ell : \Omega \times \Omega \to \mathbb{R}$; given a guess $x$ of $v$, the loss function $\ell(x, v)$ indicates the gap between the two values. We do not assume that $\ell$ is symmetric but require $\ell(x, v) \geq 0$ when $x \geq v$. Some examples of $\ell(x, v)$ are $(x - v)^2$, $|x - v|$, $x/v$, and $\ln(x/v)$. The right choice of the loss function can be context-dependent, e.g., $(x - v)^2$ is a commonly used loss function for real-valued data, and $\ln(x/v)$ is sometimes better at capturing relative error for

heavy-tailed distributions over positive domains [83]. For a density $f$ for $x$, $\mathbb{E}_{x \sim f} \, \ell(x, v)$ denotes the expected error of the evaluation process. One can therefore consider the following problem: Given a value $\Delta$, can we find an $f$ such that $\mathbb{E}_{x \sim f} \, \ell(x, v) \leq \Delta$? This problem is under-specified as there may be (infinitely) many densities $f$ satisfying this constraint. To specify $f$ uniquely, we appeal to the maximum entropy framework in statistics and information theory [81]: Among all the feasible densities, one should select the density $f$ which has the maximum entropy. This principle leads to the selection of a density that is consistent with our constraint and makes no additional assumption. We use the notion of the (differential) entropy of a density with respect to the Lebesgue measure $\mu$ on $\mathbb{R}$:

$$\mathsf{Ent}(f) := - \int_{x \in \Omega} f(x) \ln f(x) d\mu(x), \tag{1}$$

where $f(x) \ln f(x) = 0$ whenever $f(x) = 0$. Thus, we get the following optimization problem:

$$\mathrm{argmax}_{f: \text{ density on } \Omega} \, \mathsf{Ent}(f), \quad s.t., \quad \mathbb{E}_{x \sim f} \, \ell(x, v) \leq \Delta. \tag{2}$$

This optimization problem is well-studied and it is known that by using different loss functions, we can derive many families of densities [102, 148]. For instance, for $\ell(x, v) := (x - v)^2$, we recover the Gaussian density with mean $v$, and for $\ell(x, v) := \ln(x/v)$, we obtain a Pareto density.

**Step 2: Incorporating the resource-information parameter.** We now extend the above formulation to include information constraints in the evaluation process. In an evaluation process, both the evaluator and the individual spend resources such as time, cognitive effort, or money to communicate the information related to the utility of the individual to the evaluator. For instance, in interviews, both the interviewer and the interviewee spend time and cognitive effort. In university admissions, the university admissions office needs to spend money to hire and train application readers who, in turn, screen applications for the university's admission program, and the applicants need to spend time, cognitive effort, and money to prepare and submit their applications [24, 150]. The more resources are spent in an evaluation process, the more additional information about $v$ is acquired. We model this using a resource-information parameter $\tau$, which puts a lower bound on the entropy of $f$. Thus, we modify the optimization problem in Equation (2) in the following manner. We first flip the optimization problem to an equivalent problem where we minimize the expected loss subject to a lower bound on the entropy, $\mathsf{Ent}(f)$, of $f$. A higher value of the resource-information parameter $\tau$ means that one needs to spend more resources to obtain the same information and corresponds to a stronger lower bound on $\mathsf{Ent}(f)$ (and vice-versa) in our framework.

$$\mathrm{argmin}_{f: \text{ density on } \Omega} \, \mathbb{E}_{x \sim f} \, \ell(x, v), \quad s.t., \quad \mathsf{Ent}(f) \geq \tau. \tag{3}$$

When $\tau \to -\infty$, the optimal density tends to a point or delta density around $v$ (recovering the most information), and when $\tau \to \infty$, it tends to a uniform density on $\Omega$ (learning nothing about $v$). Since differential entropy can vary from negative to positive infinity, the value of $\tau$ is to be viewed relative to an arbitrary reference point. $\tau$ may vary with the socially-salient group of the individual in real-world contexts. For instance, in settings where the evaluator needs to interact with individuals (e.g., interviews), disparities can arise because it is less cognitively demanding for an evaluator to communicate with individuals who speak the same language as themselves, compared to individuals who speak a different language [95]. In settings where the evaluator assesses individuals based on data about their past education and employment (e.g., at screening stages of hiring or in university admissions), disparities can arise because the evaluator is more knowledgeable about a specific group's sociocultural background compared to others, and would have to spend more resources to gather the required information for the other groups [49, 62].

**Step 3: Incorporating the risk-averseness parameter.** We now introduce the parameter $\alpha$ that captures risk averseness. Roughly speaking, risk averseness may arise in an evaluation process because of the downstream impact of the output. The evaluator may also benefit or may be held accountable for the estimated value, and hence, would be eager or reluctant to assign values much higher than the true utility [71, 23, 14]. Further, the individual may also be risk averse, e.g. during a hiring interview, the risk of getting rejected may prompt the individual to quote less than the expected salary [13, 33, 111]. To formalize this intuition, for a given $\ell$, we define a risk-averse loss function $\ell_\alpha : \Omega \times \Omega \to \mathbb{R}$ that incorporates the parameter $\alpha \geq 1$ in $\ell$ as follows:

$$\ell_\alpha(x, v) := \alpha \cdot \ell(x, v) \text{ if } x \geq v \text{ and } \ell_\alpha(x, v) := \ell(x, v) \text{ if } x < v. \tag{4}$$

Not only does this loss function penalize overestimation versus underestimation, but in addition, the more the overestimation, the more the penalization is. This intuition is consistent with the theory

of risk averseness [10, 119]. Our choice is related to the notion of hyperbolic absolute risk aversion [78, 109], and one may pick other ways to incorporate risk averseness in the loss function [144, 98]. As an example, if $\ell(x, v) = (x - v)^2$, then the $\ell_2^2$-loss is $\ell_\alpha(x, v) = \alpha \cdot (x - v)^2$, if $x \geq v$ and $(x - v)^2$ otherwise. If $\ell(x, v) = \ln(x/v)$, then the log-ratio loss is $\ell_\alpha(x, v) = \alpha \cdot \ln(x/v)$, if $x \geq v$; $\ln(x/v)$ otherwise. Plots of these two loss functions are in Figure 2.[1] Thus, the analog of Equation (3) becomes

$$\text{argmin}_{f:\text{ density on } \Omega} \; \mathbb{E}_{x \sim f} \, \ell_\alpha(x - v, v), \quad s.t., \quad \text{Ent}(f) \geq \tau. \tag{5}$$

We note that, for the risk-averse loss function defined in (4), the following hold: For all $\alpha \geq \alpha' \geq 1$, $\ell_\alpha(x, v) \geq \ell_{\alpha'}(x, v)$ for all $x, v \in \Omega$, and $\ell_\alpha(x, v) - \ell_{\alpha'}(x, v)$ is an increasing function of $x$ for $x \geq v$. Beyond (4), one could consider other $\ell_\alpha(x, v)$ satisfying these two properties in our framework; we omit the details. One can also incorporate the (opposite) notion of "risk eager," where values of $x$ lower than $v$ are penalized more as opposed to values of $x$ higher than $v$ by letting $\alpha \in (0, 1]$.

**Step 4: Generalizing to arbitrary $f_\mathcal{D}$.** To extend Equation (5) to the setting when $v$ comes from a general density $f_\mathcal{D}$, we replace the loss function by its expectation over $f_\mathcal{D}$ and arrive at the model for the evaluation process that we propose in this paper:

$$\text{argmin}_{f:\text{ density on } \Omega} \; \text{Err}_{\ell,\alpha}(f_\mathcal{D}, f) := \int_{v \in \Omega} \left[ \int_{x \in \Omega} \ell_\alpha(x, v) f(x) d\mu(x) \right] f_\mathcal{D}(v) d\mu(v), \text{(OptProg)}$$
$$\text{such that} \quad - \int_{x \in \Omega} f(x) \log f(x) d\mu(x) \geq \tau.$$

For a given $\ell$ and parameters $\alpha$ and $\tau$, this optimization framework can be viewed as transforming the true utility density $f_\mathcal{D}$ of a group of individuals to the density $f_\mathcal{E}$ (the solution to this optimization problem). It is worth pointing out that neither the evaluator nor the individual is solving the above optimization problem – rather (OptProg) models the evaluation process and the loss function $\ell$, $\alpha$, and $\tau$ depend on the socially-salient attribute of the group of an individual; see also Appendix B.

## 3 Theoretical results

**Characterization of the optimal solution.** We first characterize the solution of the optimization problem (OptProg) in terms of $f_\mathcal{D}$, $\ell_\alpha$, $\tau$, and $\alpha$. Given a probability density $f_\mathcal{D}$, a parameter $\alpha \geq 1$, and a loss function $\ell$, consider the function $I_{f_\mathcal{D},\ell,\alpha}(x) := \int_{v \in \Omega} \ell_\alpha(x, v) f_\mathcal{D}(v) d\mu(v)$. This integral captures the expected loss when the estimated utility is $x$. Further, for a density $f$, the objective function of (OptProg) can be expressed as $\text{Err}_{\ell,\alpha}(f_\mathcal{D}, f) := \int_{x \in \Omega} I_{f_\mathcal{D},\ell,\alpha}(x) f(x) d\mu(x)$.

**Theorem 3.1** (**Informal version of Theorem C.1 in Appendix C**). *Under general conditions on $f_\mathcal{D}$ and $\ell$, for any finite $\tau$ and $\alpha \geq 1$, (OptProg) has a unique solution $f^\star(x) \propto \exp\left(-I_{f_\mathcal{D},\ell,\alpha}(x)/\gamma^\star\right)$, where $\gamma^\star > 0$ is unique and also depends on $\alpha$ and $\tau$. Further, $\text{Ent}(f^\star) = \tau$.*

The uniqueness in Theorem 3.1 implies that, if $\alpha = 1$, $\tau = \text{Ent}(f_\mathcal{D})$, and $\ell(x, v) := \ln f_\mathcal{D}(x) - \ln f_\mathcal{D}(v)$, then the optimal solution is $f^\star(x) = f_\mathcal{D}(x)$. To see this, note that in this case $I_{f_\mathcal{D},\ell,\alpha}(x) = \ln f_\mathcal{D}(x) + \text{Ent}(f_D)$. Hence, $f^\star(x) = f_\mathcal{D}(x)$ satisfies Theorem 3.1's conclusion (see Section C.7 for details). Thus, in the absence of risk averseness, and for an appropriate choice of resource-information parameter, the output density is the same as the true density.

Theorem 3.1 can be viewed as a significant extension of results that show how well-known probability distributions arise as solutions to the entropy-maximization framework. Indeed, the standard maximum-entropy formulation only considers the setting where the input utility is given by a single value, i.e., the distribution corresponding to $f_\mathcal{D}$ is concentrated at a single point; and the risk-averseness parameter $\alpha = 1$. While the optimal solution to the maximum-entropy framework (2) restricted to the class of well-known loss functions, e.g. $\ell_2^2$-loss or linear loss, can be understood by using standard tools from convex optimization (see [48]), characterizing the optimal solution to the general formulation (OptProg) is more challenging because of several reasons: (i) The input density $f_\mathcal{D}$ need not be concentrated at a single point, and hence one needs to understand conditions on $f_\mathcal{D}$ when the formulation has a unique optimal solution. (ii) The loss function can be arbitrary and one needs to formulate suitable conditions on the loss function such that (OptProg) has a unique optimal solution. (iii) The risk-averseness parameter $\alpha$ makes the loss function asymmetric (around

---

[1]A variation of (4) that we use in the empirical part is the following: For a fixed "shift" $v_0$, let $\ell_\alpha(x, v) := \alpha \cdot \ell(x, v + v_0)$ if $x \geq v + v_0$ and $\ell_\alpha(x, v) := \ell(x, v + v_0)$ if $x < v + v_0$.

any fixed value $v$) and makes the analysis of the error in the objective function non-trivial. Roughly speaking, the only restrictions, other than standard integrability assumptions, that we need on the input are: (a) Monotonicity of the loss function $\ell(x, v)$ with respect to either $x$ or $v$, (b) the growth rate of the loss function $\ell(x, v)$ is at least logarithmic, and (c) the function $I_{f_\mathcal{D}, \ell, \alpha}(x)$ has a unique global minimum, which is a much weaker assumption than convexity of the function. Note that for the $\ell_2^2$-loss function given by $\ell(x, v) = (x - v)^2$, the first two conditions hold trivially; and when the input density $f_\mathcal{D}$ is Gaussian, it is not hard to show that $I_{f_\mathcal{D}, \ell, \alpha}(x)$ is strongly convex and hence the third condition mentioned about holds (see Appendix H for details). These conditions are formally stated in Appendix C.1 and we show that they hold for the cases of Gaussian, Pareto, Exponential, and Laplace densities in Sections H, I, J, K respectively.

The proof of Theorem 3.1 is presented in Appendix C and the following are the key steps in it: (i) The proof starts by considering the dual of (OptProg) and shows that strong duality holds (see Appendix C.2 and Appendix C.3). (ii) The next step is to show that the optimal solution $f^\star$ of (OptProg) exists and is unique. This requires proving that the dual variable $\gamma^\star$ (corresponding to the entropy constraint in (OptProg)) is positive – while this variable is always non-negative, the main technical challenge is to show that it is *non-zero*. In fact, there are instances of (OptProg) where $\gamma^\star$ is zero, and an optimal solution does not exist (or an optimal solution exists, but is not unique). (iii) The proof of $\gamma^\star \neq 0$ requires us to understand the properties of the integral $I_{f_\mathcal{D}, \ell, \alpha}(x)$ (abbreviated as $I(x)$ when the parameters $f_\mathcal{D}, \ell, \alpha$ are clear from the context). In Appendix C.4 we show that $I(x)$ can be expressed as a sum of two monotone functions (see Theorem C.10). This decomposition allows us to show that the optimal value of (OptProg) is finite. (iv) In Appendix C.5, we show that the optimal value of (OptProg) is strictly larger than $I(x^\star)$ (Lemma C.13, Lemma C.14), where $x^\star$ is the minimizer of $I(x)$. This requires us to understand the interplay between the growth rate of the expected loss function and the entropy of a density as we place probability mass away from $x^\star$. Indeed, these technical results do not hold true if the loss function $\ell(x, v)$ grows very slowly (as a function of $x/v$ or $(|x - v|)$. (v) Finally, in Theorem C.15, we show that $\gamma^\star$ is nonzero. This follows from the fact that if $\gamma^\star = 0$, then the optimal value of (OptProg) is equal to $I(x^\star)$, which contradicts the claim in (iv) above. Once we show $\gamma^\star > 0$, the expression for the (unique) optimal solution, i.e., $f^\star(x) \propto \exp\left(-I(x)/\gamma^\star\right)$, follows from Theorem C.16.

We conclude this section with two remarks. 1) In Appendix E, we show that Theorem 3.1 implies that $\tau = \left(\mathrm{Err}_{\ell, \alpha}\left(f_\mathcal{D}, f^\star\right)/\gamma^\star\right) + \ln Z^\star$, where $Z^\star := \int_\Omega \exp\left(-I_{f_\mathcal{D}, \ell, \alpha}(x)/\gamma^\star\right) d\mu(x)$ is the partition function or the normalizing constant that makes $f^\star$ a probability density; This equation is an analog of the Gibbs equation in statistical physics and gives a physical interpretation of $I_{f_\mathcal{D}, \ell, \alpha}(x)$ and $\gamma^\star$: $\gamma^\star$ is the *temperature* and $I_{f_\mathcal{D}, \ell, \alpha}(x)$ is the *energy* corresponding to *state* $x$. This may be useful in understanding the effects of different parameters on the output density. 2) If one wishes, one can use (OptProg) to understand the setting where a single individual is being evaluated by setting the input density $f_\mathcal{D}$ to be concentrated at their true utility $v$. For instance, if we set the loss function to be the $\ell_2^2$-loss, using Theorem 3.1, one can show that for any given values of the parameters, $\alpha$ and $\tau$ and loss function being $\ell_2^2$-loss, the mean of the output density is $v - \sqrt{\frac{\gamma^\star}{\pi}} \cdot \frac{\sqrt{\alpha}-1}{\sqrt{\alpha}}$; see Appendix D for a proof. Therefore for $\alpha > 1$, the mean of the output density is strictly less than $u$. This gives a mapping from the "true ability" to the (mean of the) "biased ability" in this case. This mapping can be used to understand how the parameters $\tau$ and $\alpha$ in the evaluation process transform the true ability.

**Effect of varying $\tau$ for a fixed $\alpha$.** We first study the effect of changing the resource-information parameter $\tau$ on $f^\star$ for a fixed value of the risk-averseness parameter $\alpha \geq 1$. To highlight this dependency on $\tau$, here we use the notation $f^\star_\tau$ to denote the optimal solution $f^\star$. We start by noting that as $\gamma^\star$ increases, the optimal density becomes close to uniform, and as it goes towards zero, the optimal density concentrates around a point $x^\star$ that minimizes energy: $\mathrm{argmin}_{x \in \Omega} I_{f_\mathcal{D}, \ell, \alpha}(x)$. Note that the point $x^\star$ does not depend on $\tau$. However, $\gamma^\star$ may depend in a complicated manner on both $\tau$ and $\alpha$, and it is not apparent what effect changing $\tau$ has on $\gamma^\star$. We show that, for any fixed $\alpha \geq 1$, as $\tau$ decreases, the output density gets concentrated around $x^\star$ (see Theorem F.1). This confirms the intuition that as we reduce $\tau$ by adding more resources in the evaluation process, the uncertainty in the output density should be reduced. Similarly, if $\tau$ increases because of a reduction in the resources invested in the evaluation, the uncertainty in $f^\star_\tau$ should increase, and hence, the output density should converge towards a uniform density. For specific densities, one can obtain sharper results. Consider the case when $f_\mathcal{D}$ is a Gaussian with mean $m$ and variance $\sigma^2$, and $\ell_\alpha(x, v) := \alpha(x - v)^2$ if $x \geq v$, and $(x - v)^2$ if $x < v$. The uncertainty in the Gaussian density is captured by the variance, and hence, we expect the output density to have a higher variance when the parameter $\tau$ is increased. Indeed,

when $\alpha = 1$, we show that the optimal density $f_\tau^\star$ is a Gaussian with mean $m$ and variance $e^{2\tau}\sigma^2$; see Appendix H. Thus, if one increases $\tau$ from $-\infty$ to $\infty$, the variance of the output density changes *monotonically* from 0 to $\infty$. When $\alpha \geq 1$, numerically, it can be seen that for any fixed $\alpha \geq 1$, increasing $\tau$ increases the variance, and also decreases the mean of $f_\tau^\star$; see Figure 3. Intuitively, the decrease in mean occurs because higher variance increases the probability of the estimated value being much larger than the mean, and the risk-averseness parameter imposes a high penalty when the estimated value is larger than the true value. In fact, we show in Theorem F.5 that the variance of $f_\tau^\star$ for any continuous input density $f_\mathcal{D}$ supported on $\mathbb{R}$ is at least $\frac{1}{2\pi}e^{2\tau-1}$. This follows from the well-known fact that among all probability densities supported on $\mathbb{R}$ with variance $\sigma^2$, the Gaussian density with variance $\sigma^2$ maximizes the (differential) entropy [120]; see Section F.2 for details. In a similar vein, we show that the mean of $f_\tau^\star$ is at least $e^{\tau-1}$ when the input density $f_\mathcal{D}$ is supported on $[0, \infty)$, and hence approaches $\infty$ as $\tau$ goes to $\infty$ (see Theorem F.7). This result relies on the fact that among all densities supported on $[0, \infty)$ and with a fixed expectation $1/\lambda$ (for $\lambda > 0$), the one maximizing the entropy is the exponential density with parameter $\lambda$ [48].

We now consider the special setting when the input density $f_\mathcal{D}$ is Pareto with parameter $\beta > 0$ ($f_\mathcal{D}(x) := \beta x^{-\beta-1}$ for $x \in [1, \infty)$), and $\ell_\alpha(x, v) := \alpha \ln(x/v)$ if $x \geq v$, and $\ln(x/v)$ if $x < v$. When $\alpha = 1$, we show that the optimal density $f_\tau^\star$ is also a Pareto density with parameter $\beta^\star$ satisfying the following condition: $1 + (1/\beta^\star) - \ln \beta^\star = \tau$. Using this, it can be shown that, for $\alpha = 1$, both the mean and variance of $f_\tau^\star$ *monotonically* increase to $\infty$ as $\tau$ goes to $\infty$; see Appendix I. The increase in variance reflects the fact that increasing $\tau$ increases the uncertainty in the evaluation process. Unlike the Gaussian case, where the mean of $f_\tau^\star$ could shift to the left of 0 with an increase in $\tau$, the mean of the output density in this setting is constrained to be at least 1, and hence, increasing the variance of $f_\tau^\star$ also results in an increase in its mean. Numerically, for any fixed $\alpha \geq 1$, increasing $\tau$ increases both the mean and variance of $f_\tau^\star$; see Figures 3 and 6.

**Effect of varying $\alpha$ for a fixed $\tau$.** Let $f_\alpha^\star$ denote the optimal solution $f^\star$ with $\alpha$ for a fixed $\tau$. We observe that, for any fixed $x$, $I_{f_\mathcal{D}, \ell, \alpha}(x)$, which is the expected loss when the output is $x$, and $\mathrm{Err}_{\ell, \alpha}(f_\alpha^\star, f_\mathcal{D})$ are increasing functions of $\alpha$; see Appendix G. Thus, intuitively, the mass of the density should shift towards the minimizer of $I_{f_\mathcal{D}, \ell, \alpha}(x)$. Moreover, the minimizer of $I_{f_\mathcal{D}, \ell, \alpha}(x)$ itself should reduce with increasing $\alpha$. Indeed, as the evaluation becomes more risk averse, the expected loss, $I_{f_\mathcal{D}, \ell, \alpha}(x)$, for an estimated value $x$, increases. However, the asymmetry of the loss function leads to a more rapid rate of increase in $I_{f_\mathcal{D}, \ell, \alpha}(x)$ for larger values of $x$. As a result, minimizer of $I_{f_\mathcal{D}, \ell, \alpha}(x)$ decreases with increasing $\alpha$. Thus, as we increase $\alpha$, the output densities should *shrink* and/or *shift* towards the left. We verify this for Pareto and Gaussian densities numerically: for fixed $\tau$, increasing $\alpha$ decreases the mean $f_\alpha^\star$ for both Pareto and Gaussian $f_\mathcal{D}$; see Figures 4 and 7. As for the variance, with fixed $\tau$, increasing $\alpha$ increases the variance when $f_\mathcal{D}$ is Gaussian and decreases the variance when $f_\mathcal{D}$ is Pareto; see Figures 4 and 7. See also discussions in Appendices H and I.

**Connection to the implicit variance model.** Our results confirm that increasing $\tau$ effectively increases the "noise" in the estimated density by moving it closer to the uniform density. The implicit variance model of [61] also captures this phenomenon. More concretely, in their model, the observed utility of the advantaged group is a Gaussian random variable with mean $\mu$ and variance $\sigma_0^2$, and the observed utility of the disadvantaged group is a Gaussian random variable with mean $\mu$ and variance $\sigma_0^2 + \sigma^2$. This model can be derived from our framework where the input density $f_\mathcal{D}$ is Gaussian, the risk-averseness parameter $\alpha = 1$, the loss function is $\ell(x, v) = (x - v)^2$ and the disadvantaged group is associated with a higher value of the resource-information parameter $\tau$; see Section H for details.

**Connection to the multiplicative bias model.** In the multiplicative-bias model of [90], the true utility of both the groups is drawn from a Pareto distribution, and the output utility for the disadvantaged group is obtained by scaling down the true utility by a factor $\rho > 1$. This changes the domain of the distribution to $[1/\rho, \infty)$ from $[1, \infty)$ and, hence, does not fit exactly in our model which does not allow for a change in the domain. Nevertheless, we argue that when the input density $f_\mathcal{D}$ is Pareto with a parameter $\beta$, $\tau = \mathrm{Ent}(f_\mathcal{D})$ and $\alpha > 1$, and the loss function is given by $\ell(x, v) = \ln x - \ln v$, then output density $f_\alpha^\star$ has a smaller mean than that of the input density. As we increase $\alpha$, the evaluation becomes more risk-averse and hence decreases the probability of estimating higher utility values. Hence, the mean of the output density decreases. We first show that for any fixed $\beta$, as the parameter $\alpha$ increases, the output density converges to a density $g^\star$. We then show numerically that, for all the Pareto distributions considered by our study, the mean of the density $g^\star$ is less than that of the output density $f_\alpha^\star$ when $\alpha = 1$. We give details of this argument in Section I.

| Dataset | This work | | | Multiplicative bias [90] | Implicit variance [61] |
|---|---|---|---|---|---|
| | Vary $\alpha$ and $\tau$ | Fix $\alpha{=}1$ | Fix $\tau{=}\mathsf{Ent}(f_\mathcal{D})$ | | |
| JEE-2009 (Birth category) | **0.09** | 0.15 | 0.21 | 0.14 | 0.10 |
| JEE-2009 (Gender) | **0.07** | 0.15 | 0.19 | **0.07** | 0.08 |
| Semantic Scholar (Gender) | **0.03** | 0.09 | 0.23 | 0.08 | 0.13 |
| Synthetic Network | **0.03** | 0.05 | 10.0 | 0.05 | 0.22 |

Table 1: *TV distances between best-fit densities and real data (Section 4) with 80%-20% training and testing data split:* Each dataset consists of two densities $f_{G_1}$ and $f_{G_2}$ of utility, corresponding to the advantaged and disadvantaged groups. We fix $f_\mathcal{D}{=}f_{G_1}$ and report the best-fit TV distance between $f_{G_2}$ and densities output by (a) our model, (b) the multiplicative bias model, and (c) the implicit variance model. We compare our model to variants where we fix $\alpha{=}1$ and $\tau{=}\mathsf{Ent}(f_\mathcal{D})$. Our model achieves a small TV distance on all datasets.

# 4 Empirical results

**Ability to capture biases in data.** First, we evaluate our model's ability to output densities that are "close" to the densities of biased utility in one synthetic and two real-world datasets.

*Setup and discussion.* In all datasets we consider, there are natural notions of utility for individuals: scores in college admissions, number of citations in research, and degree in (social) networks. In each dataset, we fix a pair of groups $G_1$ and $G_2$ (defined by protected attributes such as age, gender, and race) and consider the empirical density of utilities $f_{G_1}$ and $f_{G_2}$ for the two groups. Suppose group $G_1$ is more privileged or advantaged than $G_2$. In all datasets, we observe notable differences between $f_{G_1}$ and $f_{G_2}$ that advantaged $G_1$ (e.g., $f_{G_1}$'s mean is at least 34% higher than $f_{G_2}$'s).

*Implementation details.* Our goal is to understand whether our model can "capture" the biases or differences between $f_{G_1}$ and $f_{G_2}$. To evaluate this, we fix $f_\mathcal{D} = f_{G_1}$, i.e., $f_{G_1}$ is the true density, and compute the minimum total variation distance between $f_{G_2}$ and a density output by our model, i.e., $\min_{\alpha,\tau} d_{\mathrm{TV}}\left(f^\star_{\alpha,\tau}, f_{G_2}\right)$; where $f^\star_{\alpha,\tau}$ is the solution to (OptProg) with inputs $\alpha$ and $\tau$. (The total variation distance between two densities $f$ and $g$ over $\Omega$ is $\frac{1}{2}\int_\Omega |f(x) - g(x)|\, d\mu(x)$ [97].) To illustrate the importance of both $\alpha$ and $\tau$, we also report the TV-distances achieved with $\alpha = 1$ (no skew) and with $\tau = \tau_0 := \mathsf{Ent}(f_\mathcal{D})$ (vacuous constraint), i.e., $\min_\tau d_{\mathrm{TV}}\left(f^\star_{1,\tau}, f_{G_2}\right)$ and $\min_\alpha d_{\mathrm{TV}}\left(f^\star_{\alpha,\tau_0}, f_{G_2}\right)$, respectively. As a further comparison, we also report the minimum TV distances achieved by existing models of biases in evaluation processes: the multiplicative-bias model and the implicit variance model [90, 61]. Concretely, we report $\min_{\mu,\rho} d_{\mathrm{TV}}\left(f_{\mathcal{D},\mu,\rho}, f_{G_2}\right)$ and $\min_{\mu,\sigma} d_{\mathrm{TV}}\left(f_{\mathcal{D},\mu,\sigma}, f_{G_2}\right)$ where $f_{\mathcal{D},\mu,\rho}$ is the density of $\rho v + \mu$ for $v \sim f_\mathcal{D}$ and $f_{\mathcal{D},\mu,\sigma}$ is the density of $v + \mu + \sigma\zeta$ where $v \sim f_\mathcal{D}$ and $\zeta \sim \mathcal{N}(0,1)$.

Below we present brief descriptions of the datasets; detailed descriptions and additional implementation appear in Appendix L.

*Dataset 1 (JEE-2009 scores).* Indian Institutes of Technology (IITs) are, arguably, the most prestigious engineering universities in India. Admission into IITs is decided based on students' performance in the yearly Joint Entrance Exam (JEE) [20]. This dataset contains the scores, birth category (official SES label [135]), and (binary) gender of all students from JEE-2009 (384,977 total) [91]. We consider the score as the utility and run two simulations with birth category ($G_1$ denotes students in GEN category) and gender ($G_1$ denotes male students) respectively as the protected attributes. We set $\Omega$ as the discrete set of possible scores. We fix $\ell_2^2$-loss as $f_{G_1}$ and $f_{G_2}$ appear to be Gaussian-like (unimodal with both a left-tail and a right-tale; see Figure 13 in Appendix L).

*Dataset 2 (Semantic Scholar Open Research Corpus).* This dataset contains the list of authors, the year of publication, and the number of citations for 46,947,044 research papers on Semantic Scholar. We consider the total first-author citations of an author as their utility and consider their gender (predicted from first name) as the protected attribute ($G_1$ denotes male authors). We fix $\Omega = \{1, 2, \dots\}$ and $\ell$ as the log-ratio loss as $f_{G_1}$ and $f_{G_2}$ have Pareto-like density.

*Dataset 3 (Synthetic network data).* We generate a synthetic network with a biased variant of the Barabási–Albert model [5, 17, 35, 93]. The vertices are divided into two groups $G_1$ and $G_2$. We start with a random graph $G_0$ with $m{=}50$ vertices where each vertex is in $G_1$ w.p. $\frac{1}{2}$ independently. We extend $G_0$ to $n{=}10,000$ vertices iteratively: at each iteration, one vertex $u$ arrives, $u$ joins $G_1$ w.p. $\frac{1}{2}$ and otherwise $G_2$, and $u$ forms one edge with an existing vertex $v$–where $v$ is chosen w.p. $\propto d_v$ if $v \in G_1$ ($d_v$ is $v$'s current degree) and $\propto \frac{1}{2}d_v$ otherwise. We use a vertex's degree as its utility, fix $\Omega = \{1, 2, ...\}$, and use log-ratio loss as $f_{G_1}$ and $f_{G_2}$ have Pareto-like density.

*Observations.* We report the TV distances for all simulations in Table 1 (also see Figure 11 and Figure 12) for the plots of the corresponding best-fit densities). We observe that across all simulations our model can output densities that are close in TV distance ($\leq 0.09$) to $f_{G_2}$. Moreover, both $\alpha$ and $\tau$ parameters are important, and dropping either can increase the TV distance significantly (e.g., by 1.5 times on JEE-2009 data with birth category and 2.66 times on the Semantic Scholar data). Compared to the implicit variance model, our model has a better fit on the JEE-2009 (Birth category), Semantic Scholar, and Synthetic Network data because the implicit variance model does not capture skew and in these datasets $f_{G_1}$ is a skewed version of $f_{G_2}$. Compared to the multiplicative bias model, our model has a better fit on the JEE-2009 (Birth category), as here the utilities have Gaussian-like distributions due to which multiplicative bias largely has a translation effect. Finally, on the JEE-2009 (Gender) data, our model's performance is similar to multiplicative bias and implicit variance models because in this data $f_{G_1}$ and $f_{G_2}$ are similar (TV distance$\leq 0.08$)

**Effect of interventions on selection.** Next, we illustrate the use of our model to study the effectiveness of different bias-mitigating interventions in downstream selection tasks (e.g., university admissions, hiring, and recommendation systems).

*Subset selection tasks.* There are various types of selection tasks [59, 4, 30]. We consider the simplest instantiation where there are $n$ items, each item $i$ has a true utility $v_i \geq 0$, and the goal is to select a size-$k$ subset $S_{\mathcal{D}}$ maximizing $\sum_{i \in S} v_i$. If $V=(v_1, \ldots, v_n)$ is known, then this problem is straightforward: select $k$ items with the highest utility. However, typically $V$ is unknown and is estimated via a (human or algorithmic) evaluation process that outputs a possibly skewed/noisy estimate $X=(x_1, \ldots, x_n)$ of $V$ [125, 152, 110, 118, 36]. Hence, the outputs is $S_{\mathcal{E}} := \mathrm{argmax}_{|S|=k} \sum_{i \in S} x_i$, which may be very different from $S_{\mathcal{D}}$ and possible has a much lower true utility: $\sum_{i \in S_{\mathcal{E}}} v_i \ll \sum_{i \in S_{\mathcal{D}}} v_i$.

*Interventions to mitigate bias.* Several interventions have been proposed to counter the adverse effects of bias in selection, including, representational constraints, structured interviews, and interviewer training. Each of these interventions tackles a different dimension of the selection task. Representational constraints require the selection to include at least a specified number of individuals from unprivileged groups [30, 37, 135]. Structured interviews reduce the scope of unintended skews by requiring all interviewees to receive the same (type of) questions [30, 123, 68]. Interviewer training aims to improve efficiency: the amount of (accurate) information the interviewer can acquire in a given time [30]. Which intervention should a policymaker enforce?

*Studying the effectiveness of interventions.* A recent and growing line of work [90, 40, 28, 61, 67, 38, 107] evaluates the effectiveness of representational constraints under specific models of bias: they ask, given $\mathcal{C}$ of subsets satisfying some constraint, when does the constraint optimal set, $S_{\mathcal{E},\mathcal{C}} := \mathrm{argmax}_{S \in \mathcal{C}} \sum_{i \in S} x_i$, have a higher utility than the unconstrained optimal $S_{\mathcal{E}}$, i.e., when is $\sum_{i \in S_{\mathcal{E},\mathcal{C}}} v_i > \sum_{i \in S_{\mathcal{E}}} v_i$? Based on their analysis [90, 40, 28, 61, 67, 38, 107] demonstrate the benefits of different constraints including, equal representation (ER), which requires the output $S$ to satisfy $|S \cap G_1| = |S \cap G_2|$ and, proportional representation (PR), which requires $S$ to satisfy $|S \cap G_1|/|G_1| = |S \cap G_2|/|G_2|$. A feature of our model is that its parameters $\alpha$ and $\tau$ have a physical interpretation, which enables the study of other interventions: for instance, structured interviews aim to reduce skew in evaluation, which corresponds to shifting $\alpha$ closer to 1, and interviewer-training affects the information-to-resource trade-off, i.e., reduces $\tau$.

Using our model we compare ER and PR with two new interventions: change $\alpha$ by 50% ($\alpha$-intervention) and change $\tau$ by 50% ($\tau$-intervention). Here, 50% is an arbitrary amount for illustration.

*Setup.* We consider a selection scenario based on the JEE 2009 data: we fix $f_{\mathcal{D}} = f_{G_1}$, $\alpha', \tau'$ to be the best-fit parameters on the JEE 2009 data (by TV distance), and $|G_1| = 1000$. Let $f_\alpha$ and $f_\tau$ be the densities obtained after applying the *alpha* intervention and the *tau* intervention respectively. (Formally, $f_\alpha = f^\star_{\alpha'/2, \tau'}$ and $f_\tau = f^\star_{\alpha', 3\tau'/2}$.) We vary $|G_2| \in \{500, 1000, 1500\}$. For each $i \in G_1$, we draw $v_i \sim f_{\mathcal{D}}$ and set $x_i = v_i$ (no bias). For each $i \in G_2$, we draw $v_i \sim f_{\mathcal{D}}$, $x_i \sim f^\star_{\alpha,\tau}$, $x_i^\alpha \sim f_\alpha$, and $x_i^\tau \sim f_\tau$ coupled so that the CDFs of the respective densities at $v_i, x_i, x_i^\alpha$, and $x_i^\tau$ are the same. We give ER and PR the utilities $\{x_i\}_i$ as input, we give $\alpha$-intervention utilities $\{x_i^\alpha\}_i$ as input, and the $\tau$-intervention utilities $\{x_i^\tau\}_i$. For each $|G_1|$ and $|G_2|$, we vary $50 \leq k \leq 1000$, sample utilities and report the expected utilities of the subset output by each intervention over 100 iterations (Figure 1). Here, 1000 is the largest value for which ER is satisfiable across all group sizes $|G_1|$ and $|G_2|$.

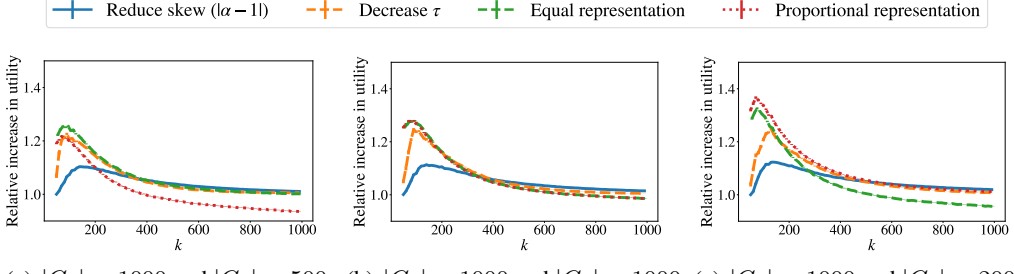

(a) $|G_1| = 1000$ and $|G_2| = 500$  (b) $|G_1| = 1000$ and $|G_2| = 1000$  (c) $|G_1| = 1000$ and $|G_2| = 2000$

Figure 1: *Effectiveness of different interventions on the selection-utility–as estimated by our model:* The $x$-axis shows $k$ (size of selection) and the $y$-axis shows the ratio of the (true) utility of the subset output with an intervention to the (true) utility of the subset output without any intervention. The main observation across the figures is that, for each intervention, there is a choice of $k$, and group sizes $|G_1|$ and $|G_2|$, where the intervention outperforms all other interventions. Hence, each intervention has a different effect on the latent utility of selection and a policymaker can use our model to study their effect and decide which intervention to enforce. Error bars represent the standard error of the mean over 100 repetitions.

*Observations.* Our main observation is that there is no pair of interventions such that one always achieves a higher utility than the others. In fact, for each intervention, there is a value of $k$, $|G_1|$, and $|G_2|$, such that the subset output with this intervention has a higher utility than the subsets output with other interventions. Thus, each intervention has a very different effect on the latent utility of the selection and a policymaker can use our model to study the effects in order to systematically decide which interventions to enforce; see Appendix L.1 for a case study of how a policymaker could potentially use this model to study bias-mitigating interventions in the JEE context.

## 5   Conclusion, limitations, and future work

We present a new optimization-based approach to modeling bias in evaluation processes ((OptProg)). Our model has two parameters, risk averseness $\alpha$ and resource-information trade-off $\tau$, which are well documented to lead to evaluation biases in a number of contexts. We show that it can generate rich classes of output densities (Theorem 3.1) and discuss how the output densities depend on the two parameters (Section 3). Empirically, we demonstrate that the densities arising from our model have a good fit with the densities of biased evaluations in multiple real-world datasets and a synthetic dataset; often, leading to a better fit than models of prior works [90, 61] (Table 1). We use our model as a tool to evaluate different types of bias-mitigating interventions in a downstream selection task–illustrating how this model could be used by policymakers to explore available interventions (Figure 1 and Appendix L.1); see also Appendix B. Our work relies on the assumptions in prior works that there are no differences (at a population level) between $G_1$ and $G_2$; see, e.g., [89, 61, 40]. If this premise is false, then the effectiveness of interventions can be either underestimated or overestimated which may lead a policymaker to select a suboptimal intervention. That said, if all the considered interventions reduce risk aversion and/or resource constraints, then the chosen intervention should still have a positive impact on the disadvantaged group. Our model can be easily used to study multiple socially-salient groups by considering a group-specific risk-aversion parameter and a group-specific information constraint. For example, if two groups $G_1, G_2$ overlap, then we can consider three disjoint subgroups $G_1 \cap G_2$, $G_1 \backslash G_2$ and $G_2 \backslash G_1$. Our model of evaluation processes considers scenarios where candidates are evaluated along a single dimension. It can also be applied – in a dimension-by-dimension fashion – to scenarios where individuals are evaluated along multiple dimensions, but the evaluation in any dimension is independent of the evaluation in other dimensions. Modeling evaluation processes involving multiple correlated dimensions is an interesting direction. While we illustrate the use of our model in a downstream selection task, utilities generated from biased evaluation processes are also used in other decision-making tasks (such as regression and clustering), and studying the downstream impact of evaluation biases on them is an important direction. Moreover, the output of or model can be used by policymakers to assess the impact of interventions in the supermodular set aggregation setting, where the utility of the selected group is more than the sum of the individuals. Our model cannot be directly used to understand the effect of interventions in the long term. Additional work would be required to do so, perhaps as in [38], and would be an important direction for future work. Finally, any work on debiasing could be used adversarially to achieve the opposite goal. We need third-party evaluators, legal protections, and available recourse for affected parties – crucial components of any system – though beyond the scope of this work.

**Acknowledgments.** This project is supported in part by NSF Awards CCF-2112665 and IIS-2045951.

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

# Contents

# A   Other related work

**Models of bias in Economics.**    There are two prominent models of discrimination in the Economics literature: taste-based discrimination and statistical discrimination [114]. These capture different types of biases [117, 45, 49, 12, 22, 41, 11] (also see [66, 25, 114]). Taste-based discrimination [22] models explicit biases (e.g., racism and sexism) and, in the vanilla taste-based discrimination model, individuals are divided into two groups and a decision-maker pays an additional cost for interacting with individuals in the disadvantaged group. This additional additive cost diminishes the value of disadvantaged individuals for the decision-maker. While we do not model explicit biases, such an additive bias also arises in our model of evaluation processes with specific parameter choices, suggesting that additive biases may also arise due to resource constraints and risk averseness (Appendix C). Statistical discrimination models how group-wise disparities in the noise in the inputs to a Bayesian decision-maker propagate to systematic disparities in decisions [117, 11]. Our mathematical model can be viewed as giving an explanation of why such disparities may arise in the input.

**Implicit biases in Psychology.**    There is a long and rich history of the study of implicit (and explicit) biases in Psychology, e.g., [85, 101, 74, 104, 69, 127, 149]. This body of works proposes various theories about why implicit biases arise [69] and their relation to real-world stimuli [130, 63, 42]. We refer the reader to [73, 87] for an overview. [69] explains that the ease of categorizing individuals into categories (defined by, e.g., color, race, or gender) provides an incentive to the evaluator to use their prior (possibly biased) knowledge, and this leads to implicit biases. In contrast, we show that even when the evaluator has the same prior estimate for all social groups, biases can arise in evaluation processes when the information-to-resource trade-off or the degree of risk averseness is different for different groups. Further, since resource constraints and risk averseness are not specific to the setting of a single evaluator, our model of evaluation processes also models scenarios where the evaluator denotes a group or an organization.

**Optimization and human behavior.**    The use of optimization to model human behavior dates back to (at least) von Neumann and Morgenstern's work that showed that, under a small number of assumptions, the behavior of an evaluator is as if they are optimizing the expected value of a "utility function" [144]. Since then, numerous apparent deviations from this theory were discovered [84]. These deviations, broadly referred to as irrational behavior or cognitive biases, laid the foundation of Behavioral Economics [138]. Several theories have been proposed to account for the deviations in human behavior from utility maximization. Including prospect theory that models risk averseness of individuals – the empirical observation that humans process losses and gains of equal amounts (monetary or otherwise) asymmetrically [84] – bounded rationality that reconciles irrational human behavior by proposing that humans solve underlying optimization problems approximately (instead of optimally) [132], and resource rational analysis that proposes that humans trade-off the utility of with the costs (e.g., such as effort and time) required to find a solution with higher utility [100]. These works model hidden costs and constraints on humans that lead to deviations from the traditional "rational" utility maximization. Like this work, these works also use optimization to explain human behavior, but while they focus on irrational behaviors and cognitive biases, our work focuses on biases with respect to socially-salient attributes in evaluation processes.

**Other entropy-based models.**    Maximum entropy distributions have been widely deployed in machine learning [56] and theoretical computer science [133]. Maximum entropy distributions have been shown to be "stable" [136]. The maximum-entropy framework over discrete distributions has been used to preprocess data to debias it [39]. In our optimization program, we use entropy to measure the amount of "information" in a distribution — this appears as a constraint in our program. An entropy-constrained viewpoint is also prevalent in explanations of other phenomena. For instance, various optimization algorithms can be captured using an optimization- and entropy-based viewpoint [9, 96, 143], it is also known to arise in explanations of biological phenomena [43], and leads to Page rank and other popular random-walk-based procedures [105]. Finally, taking an optimization viewpoint when studying a dynamical system also has additional benefits, such as providing a "potential function" that functions that gives an efficient and interpretable method of tracking the progress of a complex dynamical system [137].

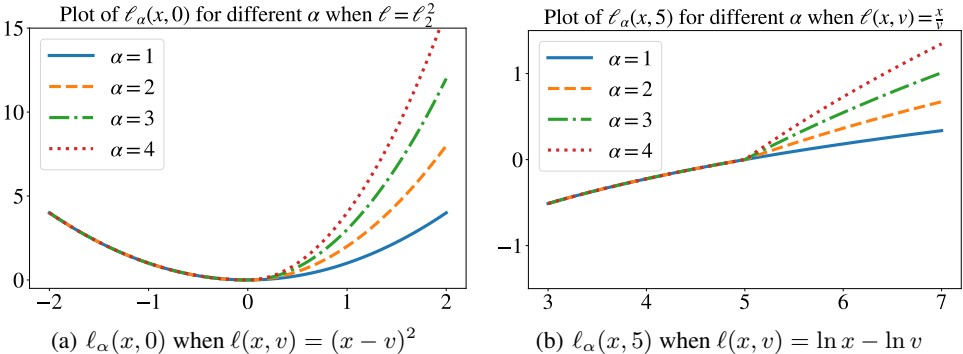

(a) $\ell_\alpha(x,0)$ when $\ell(x,v) = (x-v)^2$   (b) $\ell_\alpha(x,5)$ when $\ell(x,v) = \ln x - \ln v$

Figure 2: Plots of risk-averse loss function $\ell_\alpha(x,v)$ for different $\alpha$ and $\ell$.

## B  Specific examples of our model

In this section, we present some specific examples of mechanisms by which information constraints and risk aversion lead to bias. One context is college admissions. It is well known that SAT scores are implicitly correlated with income (and, hence, test preparation) in addition to student ability [1]. While the true ability may be $v$, the score is skewed depending on the amount/quality of test preparation, which depends on socioeconomic status that may be correlated to socially-salient attributes. The parameter $\alpha$ in our model can be used to encode this. As for $\tau$, while an evaluator may know what a GPA means at certain universities well known to them, they may not understand what GPA means for students from lesser-known schools. This lack of knowledge can be overcome, but takes effort/time, and without effort entrenches the status quo.

Another example is the evaluation of candidates using a standardized test. In time-constrained settings, a high value of the resource-information parameter $\tau$ for the disadvantaged group indicates that such candidates may not be able to comprehend a question as well as someone from an advantaged group. This could be due to various factors including less familiarity with the language used in the test or the pattern of questions, as opposed to someone who had the resources to invest in a training program for the test. Similarly, a high value of the risk-averseness parameter captures that an evaluator, when faced with a choice of awarding low or high marks to an answer given by a candidate from the disadvantaged group, is less likely to give high marks. More concretely, suppose there are several questions in a test, where each question is graded either $0$ or $1$. Assume that a candidate has true utility $v \in [0,1]$, and hence, would have received an expected score $v$ for each of the questions if one were allowed to award grades in the continuous range $[0,1]$. However, the fact that the true scores have to be rounded to either $0$ or $1$ can create a bias for the disadvantaged group. Indeed, the probability that an evaluator rounds such an answer to $1$ may be less than $v$ – the risk-averseness parameter measures the extent to which this probability gets scaled down.

Most of the prior works on interventions in selection settings have focused on adding representational constraints for disadvantaged groups. Such constraints, often framed as a form of affirmative action, could be beneficial but may not be possible to implement in certain contexts. For instance, in a landmark 2023 ruling, the US Supreme Court effectively prohibited the use of race-based affirmative action in college admissions [2]. Our work, via a more refined model of how bias might arrive at a population level in evaluation processes, allows for evaluating additional interventions that focus on procedural fairness; this allows working towards diversity and equity goals without placing affirmative-action-like constraints.

In this framing, we can consider decreasing either $\alpha$ or $\tau$. Improving either would work towards equity, but which ones to target or to what extent and via which method would be context-dependent and vary in cost. A decrease in $\alpha$ can be achieved by reducing risk-averseness in the evaluation process; e.g. by investing in better facilities for disadvantaged groups, or making the evaluation process blind to group membership. A reduction in $\tau$ may follow by allocating additional resources to the evaluation process, e.g., by letting a candidate choose an evaluation in their native language. Our framework allows a policymaker to study these trade-offs, and we discuss specific examples in Appendix L.

# C   Characterization of optimal solution to the optimization problem

In this section, we present the formal version of Theorem 3.1. We first state the general conditions under which this result is true, and give an outline of its proof. Recall that an instance of the optimization problem is given by a tuple $\mathcal{I} := (\Omega, f_\mathcal{D}, \ell, \alpha, \tau)$, where $\Omega$ is a closed interval in $\mathbb{R}$ (i.e., $\Omega$ is one of $[a, b], [a, \infty), (-\infty, b], (-\infty, \infty)$ for some suitable real values $a$ or $b$), $f_\mathcal{D}$ is the density of the true utility that is Lebesgue measurable, $\ell$ is the loss function, $\tau$ is the resource-information parameter and $\alpha \geq 1$ is the risk-averseness parameter. The main result, Theorem C.1, states that under mild conditions, the instance $\mathcal{I}$ has a unique optimal solution. We present applications of Theorem C.1 when the input density is Gaussian, Pareto, Exponential, and Laplace in Sections H, I, J, and K respectively.

In Section C.1, we state the primal optimization formulation and the assumptions needed by Theorem C.1. Clearly, we need that the instance $\mathcal{I}$ has a non-empty set of solutions, i.e., there exists a density of entropy at least $\tau$ (assumption **(A0)**). In Section C.2, we state the dual of the convex program PrimalOpt and show that weak duality holds. This proof requires integrability of the loss function with respect to the measure induced by the density $f_\mathcal{D}$ (assumption **(A3)**). This assumption also ensures that the optimal value does not become infinite.

In Section C.3, we show that strong duality holds. We use Slater's condition to prove this. This proof also shows that there exist optimal Lagrange dual variables. However, this does not show that there is an optimal solution $f^\star$ to the primal convex program. In fact, one can construct examples involving natural loss functions and densities where strong duality holds, but the optimal solution does not exist. In order to prove the existence of an optimal solution $f^\star$ to the instance $\mathcal{I}$, we show that the optimal Lagrange dual variable $\gamma^\star$ (corresponding to the entropy constraint in PrimalOpt) is strictly positive. This proof requires several technical steps.

We first study properties of the function $I_{f_\mathcal{D}, \alpha, \ell}(x)$ in Section C.4, which is the expected loss if the estimated value is $x$. It is easy to verify that the objective function of PrimalOpt, $\mathrm{Err}_{\ell, \alpha}(f_\mathcal{D}, f)$, is the integral of $I_{f_\mathcal{D}, \alpha, \ell}(x) f(x)$ over $\Omega$. Therefore, the optimal solution $f^\star(x)$ would place a higher probability mass on regions where $I_{f_\mathcal{D}, \alpha, \ell}(x)$ is small. Under natural monotonicity conditions on the loss function (assumption **(A1)**), we show that $I_{f_\mathcal{D}, \alpha, \ell}(x)$ can be expressed as a sum of an increasing and a decreasing function. This decomposition shows that for natural loss functions, e.g., those which or concave or convex in each of the coordinates, the function $I_{f_\mathcal{D}, \alpha, \ell}(x)$ is unimodal; we state this as an assumption **(A5)** to take care of general loss function settings.

In Section C.5, we show that $\gamma^\star > 0$. This proof hinges on the following result: For every instance $\mathcal{I} = (\Omega, f_\mathcal{D}, \ell, \alpha, \tau)$ with finite optimal value, there is a positive constant $\eta$ such that the optimal value is at least $I_{f_\mathcal{D}, \ell, \alpha}(x^\star) + \eta$, where $x^\star$ is the minimizer of $I_{f_\mathcal{D}, \ell, \alpha}(x)$. This requires unimodality of the function $I_{f_\mathcal{D}, \ell, \alpha}(x)$ and bounded mean (and median) of $f_\mathcal{D}$ (assumption A4). Somewhat surprisingly, this result also requires that the loss function $\ell(v, x)$ grows at least logarithmically as a function of $|v - x|$ (assumption **(A2)**). Without this logarithmic growth, an optimal solution need not exist: it may happen that there are near-optimal solutions that place vanishingly small probability mass outside the global minimum $x^\star$ of $I_{f_\mathcal{D}, \ell, \alpha}(x)$. Thus, even though we have an entropy constraint, there will be a sequence of solutions, whose error converges to the optimal value, that converges to a delta function.

In Section C.6, we use the positivity of $\gamma^\star$ to explicitly write down an expression for the optimal primal solution. We use strict convexity of the feasible region for PrimalOpt to show that the optimal solution is unique. Finally, in Section C.7, we show that if $f_\mathcal{D}$ has bounded entropy, then there is a suitable choice for the loss function $\ell$, and parameters $\alpha$ and $\tau$, such that we recover $f_\mathcal{D}$ as the unique optimal solution in the resulting instance.

## C.1   The primal optimization problem and assumptions

We re-write the optimization problem OptProg here. An instance $\mathcal{I}$ of this problem is given by a tuple $\mathcal{I} := (\Omega, f_\mathcal{D}, \ell, \alpha, \tau)$, where $\Omega$ is a closed interval in $\mathbb{R}$, $f_\mathcal{D}$ is the density of the true utility that is Lebesgue measurable, $\ell$ is the loss function, $\tau$ is the resource-information parameter and $\alpha \geq 1$ is the risk-averseness parameter. Recall that $\mu$ is the Lebesgue measure on $\mathbb{R}$. Note that $\Omega$ can be of the form $(-\infty, \infty)$, or $[a, \infty)$ for a real $a$, $(-\infty, b]$ for a real $b$, or $[a, b]$ for real $a, b, a < b$.

The primal problem for $\mathcal{I} = (\Omega, f_\mathcal{D}, \ell, \alpha, \tau)$ is as follows. Here, Dom denotes the set $\{f : \Omega \to \mathbb{R}_{\geq 0}\}$.

$$\min_{f \in \text{Dom}} \text{Err}_{\ell,\alpha}(f_\mathcal{D}, f) \coloneqq \int_\Omega \left[ \int_\Omega \ell_\alpha(x, v) f(x) d\mu(x) \right] f_\mathcal{D}(v) d\mu(v)$$

(PrimalOpt)

$$\text{such that} \quad \text{Ent}(f) \coloneqq - \int_\Omega f(x) \ln f(x) d\mu(x) \geq \tau \tag{6}$$

$$\int_\Omega f(x) d\mu(x) = 1. \tag{7}$$

We state the assumptions and justify each of these:

**A0** (**Strict feasibility of the instance**) The interval $\Omega$ has length strictly larger than $e^\tau$. This is satisfied trivially if $\Omega$ is infinite.

*Remark: In order to ensure that the set of feasible solutions to an instance of PrimalOpt is non-empty, we require that $\Omega$ has length at least $e^\tau$; otherwise even the uniform distribution on $\Omega$ shall have entropy less than $\tau$. The strict inequality is needed to ensure strong duality (Slater's condition).*

**A1** (**Monotonicity of the loss function**) We assume that the loss function $\ell : \Omega \times \Omega \to \mathbb{R}$ is continuous and $\ell(x, x) = 0$ for all $x \in \Omega$. We consider two types of loss functions, TYPEP and TYPEN. The TYPEP loss functions have the following monotonicity property: For any fixed $v \in \Omega$, $\ell(v, x)$ strictly increases as $|v - x|$ increases. It follows that $\ell(v, x) \geq 0$ with equality if and only if $v = x$. The TYPEN loss functions have the following property: For a fixed $v \in \Omega$, $\ell(v, x)$ is a strictly increasing function of $x$. It follows that $\ell(v, x) \geq 0$ if $x \geq v$ and $\ell(v, x) < 0$ if $x < v$. For example, $\ell(x, v) = (x - v)^2, |x - v|$ are of TYPEP, whereas $\ell(x, v) = \ln x - \ln v, (x - v)$ are of TYPEN.

*Remark: These are natural monotonicity properties. A TYPEP loss function ensures that the optimal density to an instance $\mathcal{I}$ of PrimalOpt assigns higher values to points where the input density $f_\mathcal{D}$ is more concentrated. A TYPEN loss function ensures that the optimal density does not place high probability mass at points that are much larger than the mean of $f_\mathcal{D}$.*

**A2** (**Growth rate of the loss function**) We would like to assume that the loss function $\ell(x, v)$ grows reasonably rapidly as $|x - v|$ increases. It turns out that the "right" growth rate would be at least logarithmic. For instance, we could require that $|\ell(x, v)|$ is $\Omega(|\ln(x - v)|)$. However, unless we assume some form of the triangle inequality, such a lower bound would not imply a lower bound on $|\ell(x_2, v) - \ell(x_1, v)|$ for any $x_2, x_1, v \in \Omega$. Thus, we make the following assumption: There is a constant $C$, such that for all $x_2, x_1, v \in \Omega$, with $|x_1 - x_2| \geq C, v \notin (x_1, x_2)$,

$$|\ell(x_2, v) - \ell(x_1, v)| \geq \ln |x_2 - x_1| - \theta_{x_1},$$

where $\theta_{x_1}$ is a value which depends on $x_1$ only. For example, when $\ell(x, v) = (x - v)$, the above is satisfied with $\theta_x = 0, C = 1$; and when $\ell(x, v) = \ln x - \ln v$, the above property holds with $\theta_x = \ln x, C = 0$.

*Remark: This is a subtle condition, and is needed to ensure that an optimal solution exists. Consider for example, an instance $\mathcal{I}$ with $\Omega = [0, \infty)$, $f_\mathcal{D}$ being the exponential density, $\ell(x, v) = \ln \ln(x + 2) - \ln \ln(v + 2)$ (the "+2" factor is to ensure that $\ln(x + 2)$ does not become negative). The parameters $\alpha, \tau$ can be arbitrary. Let $\varepsilon > 0$ be an arbitrarily small constant. Now, consider a solution that places $1 - \varepsilon$ probability mass at $x = 0$ and spreads the remaining $\varepsilon$ probability mass uniformly over an interval of length $e^{\tau/\varepsilon}$ to achieve entropy $\tau$. Since the loss function grows slowly, the expected loss of this solution decreases with decreasing $\varepsilon$. Thus, even though strong duality holds in this instance, an optimal solution does not exist. In fact, we have a sequence of solutions, with error converging to the optimal value of $\mathcal{I}$, converging to the delta-function at 0.*

**A3** (**Integrability of the loss function**) We assume that the function $\ell(x, v) f_\mathcal{D}(v)$ is in $L^1(\mu)$ for every $x \in \Omega$. In other words, for each $x \in \Omega$,

$$\int_\Omega |\ell(x, v)| f_\mathcal{D}(v) d\mu(v) < \infty.$$

**Remark:** *This is needed in order to carry out basic operations like the swapping of integrals (Fubini's Theorem) on the objective function. Further, this ensures that the objective function does not become infinite under reasonable conditions.*

**A4** (**Bounded mean and half-radius**) We assume that the true utility density $f_{\mathcal{D}}$ has a bounded mean $m$. Moreover, we assume that there is a finite value $R$ such that at least half of the probability mass of $f_{\mathcal{D}}$ lies in the range $[m - R, m + R]$, i.e.,

$$\int_{\Omega \cap [m-R, m+R]} f_{\mathcal{D}}(v) d\mu(v) \geq 1/2.$$

**Remark:** *This condition ensures that $f_{\mathcal{D}}$ decays at a reasonable rate (though it is much milder than requiring bounded variance).*

**A5** (**Unique global minimum for estimated loss function**) Let $I_{f_{\mathcal{D}}, \ell, \alpha}(x)$ denote the expected loss when the estimated value is $x$, i.e.,

$$I_{f_{\mathcal{D}}, \ell, \alpha}(x) := \int_{\Omega} \ell_{\alpha}(x, v) f_{\mathcal{D}}(v) d\mu(v).$$

We assume that this function has a unique minimum on $\Omega$. Moreover, if $x^{\star}$ denotes $\operatorname{argmin}_{x \in \Omega} I_{f_{\mathcal{D}}, \ell, \alpha}(x)$, we also assume that for all other local minima $x$ of this function, $I_{f_{\mathcal{D}}, \ell, \alpha}(x)$ is larger than $I_{f_{\mathcal{D}}, \ell, \alpha}(x^{\star})$ by a fixed constant. We state this condition formally as follows: Given any $\delta > 0$, there is an $\varepsilon_{\delta} > 0$ such that for all $x$ satisfying $|x - x^{\star}| > \delta$, we have $I_{f_{\mathcal{D}}, \ell, \alpha}(x) \geq I_{f_{\mathcal{D}}, \ell, \alpha}(x^{\star}) + \varepsilon_{\delta}$.

**Remark:** *This condition is needed to ensure the uniqueness of the optimal solution. The optimal density for an instance $\mathcal{I} = (\Omega, f_{\mathcal{D}}, \ell, \alpha, \tau)$ tries to place higher probability mass in regions where $I_{f_{\mathcal{D}}, \ell, \alpha}(x)$ is low. Therefore, a unique global minimum (and well-separatedness from other local minima) is needed to ensure that the optimal solution is unique. When the loss function is* TYPEN*, this condition is always satisfied (unless the optimal value is $-\infty$). For a* TYPEP *loss function, this condition is satisfied if the loss function $\ell(x, v)$ is concave or convex in $x$ (for each fixed value of $v$), e.g., when $\ell(x, v) = (x - v)^2, |x - v|, |\ln x - \ln v|$.*

The following is the formal version of Theorem 3.1.

**Theorem C.1** (**Characterization of optimal density**). *Consider an instance $\mathcal{I}$ of the optimization problem PrimalOpt defined on a closed interval $\Omega \subseteq \mathbb{R}$. Let $\ell, \alpha, \tau, f_{\mathcal{D}}$ denote the loss function, risk-averseness parameter, resource-information parameter, and the density of the true utility respectively in $\mathcal{I}$. If assumptions* (**A0**)–(**A5**) *are satisfied, then there is a unique solution $f^{\star}$ to the instance $\mathcal{I}$. This solution satisfies the following condition:*

$$f^{\star}(x) \propto \exp\left(-\frac{I_{f_{\mathcal{D}}, \ell, \alpha}(x)}{\gamma^{\star}}\right), \tag{8}$$

*where $\gamma^{\star}$ is the Lagrange variable corresponding to the entropy constraint* (6) *and is strictly positive. Moreover, $\mathsf{Ent}(f^{\star}) = \tau$.*

## C.2 Dual formulation and weak duality

Let $\gamma \geq 0$ and $\phi \in \mathbb{R}$ denote the Lagrange variables for the constraints (6) and (7) respectively. Then, the Lagrangian is

$$L(f, \gamma, \phi) := \mathsf{Err}_{\ell, \alpha}(f_{\mathcal{D}}, f) + \gamma(\tau - \mathsf{Ent}(f)) + \phi\left(\int_{\Omega} f(x) d\mu(x) - 1\right).$$

Given $\gamma \geq 0, \phi$, define

$$g(\gamma, \phi) := \inf_{f \in \mathsf{Dom}} L(f, \gamma, \phi). \tag{9}$$

The dual problem is

$$\max_{\gamma \in \mathbb{R}_{\geq 0}, \phi \in \mathbb{R}} g(\gamma, \phi). \tag{DualOpt}$$

We first show weak duality.

**Theorem C.2 (Weak duality).** *Consider an instance $\mathcal{I} = (\Omega, f_{\mathcal{D}}, \ell, \alpha, \tau)$ of* PrimalOpt *satisfying assumption* **(A3)**. *Let $g(\gamma, \phi)$ be as defined in* (9). *Then,*

$$\max_{\gamma \in \mathbb{R}_{\geq 0}, \phi \in \mathbb{R}} g(\gamma, \phi) \leq \min_{f : f \text{ feasible for PrimalOpt}} \mathrm{Err}_{\ell, \alpha}(f_{\mathcal{D}}, f).$$

*Proof.* We assume that the primal problem has a feasible solution, otherwise the desired inequality follows trivially. Let $f$ be a feasible solution to PrimalOpt and let $\gamma, \phi$ be real values with $\gamma \geq 0$. Then,

$$g(\gamma, \phi) \leq L(f, \gamma, \phi)$$
$$= \mathrm{Err}_{\ell, \alpha}(f_{\mathcal{D}}, f) + \gamma(\tau - \mathsf{Ent}(f)) + \phi \left( \int_{\Omega} f(x) d\mu(x) \right)$$
$$\leq \mathrm{Err}_{\ell, \alpha}(f_{\mathcal{D}}, f),$$

where the first inequality follows from the definition of $g(\gamma, \phi)$, and the last inequality follows from the fact that $f$ is a feasible solution to the instance $\mathcal{I}$ of PrimalOpt. $\qquad \square$

## C.3   Strong duality

**Theorem C.3 (Strong duality).** *Consider an instance $\mathcal{I} = (\Omega, f_{\mathcal{D}}, \ell, \alpha, \tau)$ of* PrimalOpt *satisfying assumptions* **(A0)**, **(A1)** *and* **(A3)**. *Let $f^{\star}$ and $(\gamma^{\star}, \phi^{\star})$ be the optimal solutions to* PrimalOpt *and* DualOpt *respectively. Then $g(\gamma^{\star}, \phi^{\star}) = \mathrm{Err}_{\ell, \alpha}(f_{\mathcal{D}}, f^{\star})$.*

*Proof.* We first observe that there is a feasible solution to the instance $\mathcal{I}$. This is so because, by assumption **(A0)**, $\Omega$ must contain an interval $I$ of length at least $e^{\tau}$. Now, we define $f$ as the uniform distribution on $I$. Then $\mathsf{Ent}(f) = \tau$, and hence, $f$ is a feasible solution to the instance $\mathcal{I}$. We now argue that $\mathrm{Err}_{\ell, \alpha}(f_{\mathcal{D}}, f)$ is finite.

**Claim C.4.** *Consider the function $f$ and the interval $I$ defined above. Then, $\mathrm{Err}_{\ell, \alpha}(f_{\mathcal{D}}, f) < \infty$.*

*Proof.* Let $a, b$ denote the left and the right end-points of the interval $I$ respectively. For any $x \in I$ and $v \in \Omega$, we claim that $\ell_{\alpha}(x, v) \leq \max(\ell_{\alpha}(a, v), \ell_{\alpha}(b, v))$. First, consider the case when the loss function is TYPEP. If $v$ is at least $x$, then $\ell(x, v) \leq \ell(a, v)$; otherwise $\ell(x, v) \leq \ell(b, v)$. If the loss function is TYPEN, then we know that it is an increasing function of $x$, and therefore, $\ell(x, v) \leq \ell(b, v)$. Thus, we see that for any $x \in I$, $v \in \Omega$, $\ell_{\alpha}(x, v) \leq \max(\ell_{\alpha}(a, v), \ell_{\alpha}(b, v)) \leq |\ell_{\alpha}(a, v)| + |\ell_{\alpha}(b, v)|$. Therefore (recall that $f$ is the uniform distribution on $I$),

$$\mathrm{Err}_{\ell, \alpha}(f_{\mathcal{D}}, f) \leq \frac{1}{|I|} \int_{I} |\ell(a, v)| f_{\mathcal{D}}(v) d\mu(v) + \frac{1}{|I|} \int_{I} |\ell(b, v)| f_{\mathcal{D}}(v) d\mu(v) < \infty,$$

where the last inequality follows from assumption **(A3)**. $\qquad \square$

Thus, we see that the optimal value of PrimalOpt is either finite or $-\infty$. If it is $-\infty$, then weak duality implies that DualOpt has optimal value $-\infty$ as well. Thus, we have shown strong duality in this case.

For the rest of the proof, assume that the optimal value, $p^{\star}$, of PrimalOpt is finite. We show strong duality using Slater's condition. Towards this, we define two sets $\mathcal{A}$ and $\mathcal{B}$, both of which are contained in $\mathbb{R}^3$. Define

$$\mathcal{A} := \left\{ (u, v, t) : u \geq \tau - \mathsf{Ent}(f), v = \int_{\Omega} f(x) d\mu(x) - 1, t \geq \mathrm{Err}_{\ell, \alpha}(f_{\mathcal{D}}, f), \text{for some } f \in \mathsf{Dom} \right\},$$

and

$$\mathcal{B} := \{(0, 0, q) : q < p^{\star}\}.$$

It is easy to see that $\mathcal{B}$ is convex. We show that $\mathcal{A}$ is also convex.

**Claim C.5.** *The set $\mathcal{A}$ as defined above is a convex subset of $\mathbb{R}^3$.*

*Proof.* The proof follows from the fact that $\mathsf{Ent}(f)$ is a concave function of $f$ and $\mathrm{Err}_{\ell,\alpha}(f_{\mathcal{D}}, f)$ is linear in $f$. Formally, suppose $(u_1, v_1, t_1), (u_2, v_2, t_2) \in \mathcal{A}$ and $\lambda \in [0, 1]$. We need to show that $(u, v, t) := \lambda(u_1, v_1, t_1) + (1 - \lambda)(u_2, v_2, t_2) \in \mathcal{A}$. By the definition of the set $\mathcal{A}$, there exist $f_1, f_2 \in \mathsf{Dom}$ such that

$$u_i \geq \tau - \mathsf{Ent}(f_i), \; v_i = \int_\Omega f_i(x) d\mu(x) - 1, \; t_i \geq \mathrm{Err}_{\ell,\alpha}(f_{\mathcal{D}}, f_i), \quad i \in \{1, 2\}.$$

Consider $f = \lambda f_1 + (1 - \lambda) f_2$. Then $f$ is also a density and is in $\mathsf{Dom}$. Further,

$$\mathrm{Err}_{\ell,\alpha}(f_{\mathcal{D}}, f) = \lambda_1 \mathrm{Err}_{\ell,\alpha}(f_{\mathcal{D}}, f_1) + (1 - \lambda_1) \mathrm{Err}_{\ell,\alpha}(f_{\mathcal{D}}, f_2) = \lambda t_1 + (1 - \lambda) t_2 = t.$$

Similarly, since $-x \ln x$ is concave,

$$\mathsf{Ent}(f) = - \int_\Omega f(x) \ln f(x) d\mu(x)$$
$$\geq -\lambda \int_\Omega f_1(x) \ln f_1(x) d\mu(x) - (1 - \lambda) \int_\Omega f_2(x) \ln f_2(x) d\mu(x) \geq \tau.$$

Thus, $(u, v, t) \in \mathcal{A}$. $\qquad\square$

We argue that $\mathcal{A}$ and $\mathcal{B}$ are disjoint. Indeed, otherwise, there is an $f \in \mathsf{Dom}$ such that $f$ is a density, $\mathsf{Ent}(f) \geq \tau$ and $\mathrm{Err}_{\ell,\alpha}(f_{\mathcal{D}}, f) < p^\star$, a contradiction. By the hyperplane separation theorem [34, 143], there is a hyperplane $(\tilde{\gamma}, \tilde{\phi}, \tilde{\nu})^\top (x_1, x_2, x_3) = c$ such that $\mathcal{A}$ and $\mathcal{B}$ lie on different sides of this hyperplane. In other words, for every $(x_1, x_2, x_3) \in \mathcal{A}$,

$$\tilde{\gamma} x_1 + \tilde{\phi} x_2 + \tilde{\nu} x_3 \geq c,$$

and for every $(x_1, x_2, x_3) \in \mathcal{B}$,

$$\tilde{\gamma} x_1 + \tilde{\phi} x_2 + \tilde{\nu} x_3 \leq c.$$

Therefore, for each $f \in \mathsf{Dom}$,

$$\tilde{\gamma}(\tau - \mathsf{Ent}(f)) + \tilde{\phi} \left( \int_\Omega f(x) d\mu(x) - 1 \right) + \tilde{\nu} \mathrm{Err}_{\ell,\alpha}(f_{\mathcal{D}}, f) \geq c, \tag{10}$$

and

$$\tilde{\nu} p^\star \leq c. \tag{11}$$

**Claim C.6.** *$\tilde{\gamma}$ and $\tilde{\nu}$ are non-negative.*

*Proof.* Suppose, for the sake of contradiction, that $\tilde{\gamma} < 0$. Consider $f \in \mathsf{Dom}$ as given by Claim C.4. Choose values $(x_1, x_2, x_3)$ as follows: $x_1$ is a large enough value greater than $\tau - \mathsf{Ent}(f)$, $x_2 = 0$, $x_3 = \mathrm{Err}_{\ell,\alpha}(f_{\mathcal{D}}, f)$. Claim C.4 shows that $x_3$ is finite. Since $(x_1, x_2, x_3) \in \mathcal{A}$, $\tilde{\gamma} x_1 + \tilde{\nu} x_3 \geq \alpha$. Since $\tilde{\gamma} < 0$, we can choose $x_1$ large enough to make $\tilde{\gamma} x_1 + \tilde{\nu} x_3$ go below $\alpha$, which is a contradiction. Similarly, we can show that $\tilde{\nu} \geq 0$. $\qquad\square$

Thus, two cases arise (i) $\tilde{\nu} > 0$, or (ii) $\tilde{\nu} = 0$. First, consider the case when $\tilde{\nu} > 0$. Define $\gamma^\star = \tilde{\gamma}/\tilde{\nu}, \phi^\star = \tilde{\phi}/\tilde{\nu}$. Inequalities (10) and (11) show that for all $f \in \mathsf{Dom}$,

$$\gamma^\star(\tau - \mathsf{Ent}(f)) + \phi^\star \left( \int_\Omega f(x) d\mu(x) - 1 \right) + \mathrm{Err}_{\ell,\alpha}(f_{\mathcal{D}}, f) \geq p^\star,$$

i.e., $g(\gamma^\star, \phi^\star) \geq p^\star$. It follows from weak duality (Theorem C.2) that $g(\gamma^\star, \phi^\star) = p^\star$ and the dual optimum value is equal to $p^\star$.

Now consider the case when $\tilde{\nu} = 0$. Again, inequalities (10) and (11) show that for all $f \in \mathsf{Dom}$,

$$\tilde{\gamma}(\tau - \mathsf{Ent}(f)) + \tilde{\phi} \left( \int_\Omega f(x) d\mu(x) - 1 \right) \geq 0.$$

Next, we observe that there is an $f_0 \in \mathsf{Dom}$ such that $f_0$ is a density and $\mathsf{Ent}(f_0) > \tau$. Indeed, let $f_0$ be the uniform distribution over an interval of length strictly larger than $e^\tau$ (such an interval exists by assumption **(A0)**). Substitution $f = f_0$ in the above inequality, and assuming $\tilde{\gamma} > 0$, the l.h.s. of the above inequality becomes strictly less than 0, which is a contradiction. Therefore, it must be the case that $\tilde{\gamma} = 0$. Hence, we see that for all $f \in \mathsf{Dom}$,

$$\tilde{\phi}\left(\int_\Omega f(x)d\mu(x) - 1\right) \geq 0.$$

Since all the three quantities $\tilde{\gamma}, \tilde{\phi}, \tilde{\nu}$ cannot be 0, it must be the case that $\phi^\star \neq 0$. But for any density $f_0 \in \mathsf{Dom}$, by suitably scaling it by a positive real, we can make the quantity $\left(\int_\Omega f(x)d\mu(x) - 1\right)$ strictly larger than or strictly smaller than 0, which is again a contradiction. Hence, we have concluded that $\tilde{\nu}$ cannot be 0. This completes the proof of strong duality. $\qquad\square$

**Corollary C.7.** *Consider an instance $\mathcal{I} = (\Omega, f_\mathcal{D}, \ell, \alpha, \tau)$ of PrimalOpt and assume that the optimal value $p^\star$ is finite. Then there exists a solution $(\gamma^\star, \phi^\star)$ to DualOpt such that $p^\star = g(\gamma^\star, \phi^\star)$.*

*Proof.* This follows from the proof of Theorem C.3. When $p^\star$ is finite, the parameter $\tilde{\nu} >$ and, hence, $\gamma^\star = \tilde{\gamma}/\tilde{\nu}, \phi^\star = \tilde{\phi}/\tilde{\nu}$ as defined in the proof of this result satisfy the property that $g(\gamma^\star, \phi^\star) = p^\star$. $\qquad\square$

We would now like to prove that, assuming that the optimal primal value is finite, the optimal dual variable $\gamma^\star$ is non-zero. This allows us to write an explicit expression for an optimal solution to PrimalOpt. We first need to understand the properties of the following integral, which was defined in assumption **(A5)**:

$$I_{f_\mathcal{D}, \ell, \alpha}(x) = \int_\Omega \ell_\alpha(x, v) f_\mathcal{D}(v) d\mu(v).$$

When the parameters $f_\mathcal{D}, \ell, \alpha$ will be clear from the context, we shall often abbreviate $I_{f_\mathcal{D}, \ell, \alpha}(x)$ as $I(x)$.

### C.4  Properties of the integral $I(x)$

We study some of the key properties of the integral $I(x)$. We shall assume that assumptions **(A0)–(A4)** hold. The integral $I(x)$ can be split into two parts:

$$I(x) = \alpha \underbrace{\int_{\Omega \cap (-\infty, x]} \ell(x, v) f_\mathcal{D}(v) d\mu(v)}_{:= I^L(x)} + \underbrace{\int_{\Omega \cap [x, \infty)]} \ell(x, v) f_\mathcal{D}(v) d\mu(v)}_{:= I^R(x)}.$$

**Lemma C.8.** *The integral $I^L(x)$ is a strictly increasing function of $x$. Further, $I^L(x_2) - I^L(x_1) \geq \alpha(\ln(x_2 - x_1) - \theta_{x_1})F_\mathcal{D}(x_1)$, for all $x_2, x_1 \in \Omega$ satisfying $x_2 - x_1 \geq C$. Here, $F_\mathcal{D}$ denotes the cumulative distribution function (c.d.f.) of $f_\mathcal{D}$.*

Recall that the parameter $\theta_{x_1}$ appears in the assumption **(A3)**.

*Proof.* Consider $x_1, x_2 \in \Omega$ with $x_1 < x_2$. Then

$$I^L(x_2) - I^L(x_1) = \alpha \int_{-\infty}^{x_1} (\ell(x_2, v) - \ell(x_1, v)) f_\mathcal{D}(v) d\mu(v) + \alpha \int_{[x_1, x_2] \cap \Omega} \ell(x_2, v) f_\mathcal{D}(v) d\mu(v).$$

By assumption **(A1)**, $\ell(x_2, v) > \ell(x_1, v)$ for all $v \leq x_1$ and $\ell(x_2, v) \geq 0$ for all $v \leq x_2$, we see that $I^L(x_2) > I^L(x_1)$. Suppose $x_2 - x_1 \geq C$. Then $\ell(x_2, v) - \ell(x_1, v) \geq \ln|x_2 - x_1| - \theta_{x_1}$. Therefore, using A3, the first integral above is at least

$$\alpha(\ln(|x_2 - x_1| - \theta_{x_1}) \int_{-\infty}^{x_1} f_\mathcal{D}(v) d\mu(v) = \alpha(\ln(x_2 - x_1) - \theta_{x_1})F_\mathcal{D}(x_1).$$

$\qquad\square$

**Lemma C.9.** *The integral $I^R(x)$ is a strictly increasing function of $x$ when the loss function is* TYPEN, *and is a strictly decreasing function of $x$ when the loss function is* TYPEP.

*Proof.* Consider $x_1, x_2 \in \Omega$ with $x_1 < x_2$. Then

$$I^R(x_2) - I^R(x_1) = - \int_{x_1}^{x_2} \ell(x_1, v) f_{\mathcal{D}}(v) d\mu(v) + \int_{[x_2,\infty)\cap\Omega} (\ell(x_2, v) - \ell(x_1, v)) f_{\mathcal{D}}(v) d\mu(v).$$

First, consider the case of TYPEN loss function. Then for all $v \geq x_1$, $\ell(x_1, v) \leq \ell(v, v) = 0$, and hence, the first integrand above is positive. We also know that for all $v$, $\ell(x_2, v) \geq \ell(x_1, v)$, and hence, the second integrand above is also positive. This shows that $I^R(x)$ is an increasing function of $x$ when the loss function is TYPEN.

Now consider the case when the loss function is TYPEP. For any value $v \geq x_1$, $\ell(x_1, v) \geq \ell(v, v) = 0$. Similarly, for $v \geq x_2$, $\ell(x_2, v) < \ell(x_1, v)$. Thus, $I^R(x_2) < I^R(x_1)$ in this case. $\square$

Combining Lemma C.8 and Lemma C.9, we get the main result about the variation of $I(x)$.

**Theorem C.10** (**Monotonicity of $I(x)$ with respect to $x$**). *Assume that conditions* **(A0)**–**(A4)** *hold. The function $I(x)$ is a continuous function of $x$. For a* TYPEN *loss function, $I(x)$ is a monotonically strictly increasing function of $x$, with $I(x)$ going to $-\infty$ as $x$ goes to $-\infty$ (assuming $\Omega$ is unbounded from below). For a* TYPEP *loss function, $I(x)$ is the sum of an increasing and a decreasing function, and has a global minimum on $\Omega$. Further, $I(x)$ goes to $\infty$ as $x$ goes to $\infty$ or $-\infty$ (assuming these values lie in $\Omega$).*

*Proof.* First, consider the case of TYPEN loss functions. It follows from Lemma C.8 and Lemma C.9 that $I(x)$ is a strictly increasing function of $x$. The integrand in $I^L(x)$ is $\mathbf{I}[x \geq v]\ell(x, v)f_{\mathcal{D}}(v)$, where $\mathbf{I}[\,]$ denotes the indicator function. As shown in the proof of Lemma C.8, this is a monotonically decreasing function of $x$. Therefore, the monotone convergence theorem [126] implies that $I^L(x)$ goes to 0 as $x$ goes to $-\infty$. Similarly, $I^R(x)$ goes to $-\infty$ as $x$ goes to $-\infty$. Similarly, in the case of TYPEP loss function, $I^L(x)$ goes to 0 and $I^R(x)$ goes to $\infty$ as $x$ goes to $-\infty$. Thus, $I(x)$ goes to $\infty$. Similarly, $I(x)$ goes to $\infty$ as $x$ goes to $\infty$. Since $I(x)$ is the sum of an increasing and decreasing function, and is infinite as $x$ goes to $\infty$ or $-\infty$, it must have a global minimum on $\Omega$. $\square$

**Corollary C.11.** *Consider an instance $\mathcal{I} := (\Omega, f_{\mathcal{D}}, \ell, \alpha, \tau)$ of PrimalOpt. If the loss function is* TYPEP, *then the optimal value is always finite. If the loss function is* TYPEN *and the optimal value is finite, then $\Omega$ is of the form $[a, \infty)$ or $[a, b]$ for some $a, b \in \mathbb{R}$.*

*Proof.* First, consider the case when the loss function is TYPEP. We exhibit a solution $f$ with finite objective value. Let $f$ be the uniform distribution over an interval $A$ of length $e^\tau$ (by assumption **(A0)**, such an interval always exists). Since $I(x)$ is continuous, it achieves a maximum value on $A$ – let this value be $p$. Then $\text{Err}_{\ell,\alpha}(f_{\mathcal{D}}, f) \leq p$. Thus, we see that the optimal value for this instance is finite.

Now we prove the second assertion. Suppose, for the sake of contradiction, that the loss function is TYPEN and $\Omega$ is unbounded from below. We claim that the optimal value for this instance is $-\infty$, which will be a contradiction. To see this, consider a density $f \in \text{Dom}$ which is uniform on an interval $A = [s, t]$ of length $e^\tau$ (by assumption **(A0)**). The entropy of this density is $\tau$ and, hence, this is a feasible solution to the instance $\mathcal{I}$. However,

$$\text{Err}_{\ell,\alpha}(f_{\mathcal{D}}, f) = \int_\Omega I(x)f(x)d\mu(x) = \frac{1}{e^\tau} \int_A I(x)d\mu(x) \leq I(t).$$

Thus, we can keep moving the interval $A$ to the left, which would mean that $I(t)$ would tend to $-\infty$ (using Theorem C.10). This shows that the optimal value of this instance is $-\infty$. $\square$

We shall use the following fact about the finiteness of optimal value when condition **(A5)** is also satisfied.

**Claim C.12.** *Consider an instance $\mathcal{I} = (\Omega, f_{\mathcal{D}}, \ell, \alpha, \tau)$ of PrimalOpt satisfying conditions* **(A0)**–**(A5)**. *Then the optimal value is always finite.*

*Proof.* If the loss function is TYPEP, this follows from Corollary C.11. In the case of a TYPEN loss function, the optimal value is finite unless $\Omega$ is unbounded from below. In this case, Theorem C.10 shows that $\text{argmin}_{x \in \Omega} I(x)$ does not exist, and hence, assumption **(A5)** is violated. $\qquad\square$

## C.5   Positivity of the optimal dual variable

We now prove that the optimal dual variable $\gamma^\star$ is strictly positive. In this section, we assume that assumptions **(A0)**–**(A5)** hold. For this, we need certain technical results.

**Lemma C.13.** *Consider an instance $\mathcal{I} = (\Omega, f_{\mathcal{D}}, \ell, \alpha, \tau)$ of PrimalOpt with the loss function being* TYPEP. *Let $x^\star$ be $\text{argmin}_{x \in \Omega} I(x)$. Then there is a value $\eta > 0$, such that any feasible solution $f$ must have $\text{Err}_{\ell,\alpha}(f_{\mathcal{D}}, f) \geq I(x^\star) + \eta$.*

*Proof.* Let $x^\star$ denote the unique global minimum of $I(x)$ on $\Omega$ (using assumption **(A5)**). We show that there is a small enough value $\varepsilon > 0$ such that any feasible solution $f$ must place at least $\varepsilon$ amount of probability mass outside $I_\varepsilon := [x^\star - \varepsilon, x^\star + \varepsilon]$. This suffices for the following reason. We know by the assumption **(A5)** that there is a positive value $\zeta$ such that $I(x) > I(x^\star) + \zeta$ for all $x \notin I_\varepsilon$. Thus,

$$
\text{Err}_{\ell,\alpha}(f_{\mathcal{D}}, f) = \int_{\Omega \cap I_\varepsilon} I(x)f(x)d\mu(x) + \int_{\Omega \setminus I_\varepsilon} I(x)f(x)d\mu(x)
$$
$$
\geq I(x^\star) + \varepsilon\zeta.
$$

Hence, we can choose the desired value $\eta$ to be $\varepsilon\zeta$.

It remains to find such a $\varepsilon$. We consider such a value $\varepsilon$ and assume that a feasible solution $f$ places strictly more than $1 - \varepsilon$ probability mass inside $I_\varepsilon$. We need one notation: For an interval $A$, define

$$
\text{Ent}_A(f) := -\int_{\Omega \cap A} f(x) \ln f(x) d\mu(x).
$$

Let $m$ be the mean of $f_{\mathcal{D}}$ and let $R$ be the half-radius of $f_{\mathcal{D}}$, i.e., $f_{\mathcal{D}}$ places at least half of its mass in the interval $[m - R, m + R] \cap \Omega$. Let $A_0$ be a large enough interval containing $x^\star$ and the interval $[m - R, m + R]$ – assume that the end-points of $A_0$ are at least $A_0/2$ away from any point in $[m - R, m + R]$. We consider intervals of growing size around $A_0$. Let $A_0 := [s_0, t_0]$. Let $A_1$ be the union of the intervals on both sides of $A_0$, each of length $e^{2/\varepsilon}$. Similarly, having defined $A_{i-1}$ (which will be a union of two intervals), define $A_i$ to be the union of two intervals on both sides of $A_{i-1}$, each of length $e^{2^i}$.

Consider the feasible solution $f$. Let $\beta_i$ be the total probability mass placed by $f$ on $A_i$. Thus,

$$
\text{Ent}(f) = \text{Ent}_{I_\varepsilon}(f) + \text{Ent}_{A_0 \setminus I_\varepsilon}(f) + \sum_{i \geq 1} \text{Ent}_{A_i}(f).
$$

We bound each of the terms above. Note that $\text{Ent}_{A_i}(f)$ is maximized when we distribute $\beta_i$ mass uniformly over $A_i$. Similarly, $\text{Ent}_{I_\varepsilon}(f)$ is at most $\ln(\varepsilon)$ and $\text{Ent}_{A_0 \setminus I_\varepsilon} \leq \ln|A_0| = \ln D$. Thus, we see that

$$
\tau \leq \text{Ent}(f) \leq \ln(\varepsilon D) + \sum_{i \geq 1} \beta_i \ln \frac{|A_i|}{\beta_i} \leq \ln(\varepsilon D) + \sum_{i \geq 1} \beta_i \ln|A_i|.
$$

In other words,

$$
\sum_{i \geq 1} \beta_i \ln|A_i| \geq \tau - \ln(\varepsilon D).
$$

Observe that we can still choose $\varepsilon$ and, hence, we will choose it such that the r.h.s. above becomes as large as we want (ideally, much larger than $p^\star$). Observe that any point in $A_i$ is at least $|A_{i-1}|$

distance away from any point in $[m - R, m + R]$. Therefore, (and this is where we use non-negativity of the loss function, which has been assumed to be TYPEP)

$$\text{Err}_{\ell,\alpha}(f_{\mathcal{D}}, f) \geq \sum_i \frac{\beta_i}{2} \left(\ln |A_{i-1}| - \theta_x\right) \geq \sum_i \frac{\beta_i \ln |A_i|}{4} - \theta_x,$$

where we have used the fact that $|\ln |A_i| = 2 \ln |A_{i-1}|$. It follows from the above two inequalities that if we choose $\varepsilon = \left(\frac{e^\tau}{4\theta_x p^\star D}\right)^4$, then $\text{Err}_{\ell,\alpha}(f_{\mathcal{D}}, f) \geq 2p^\star$. Hence, either $f$ places less than $1 - \varepsilon$ mass inside $I_\varepsilon$ or its objective function value is more than $2p^\star$. □

The above proof only worked for TYPEP loss functions. We need a similar result for TYPEN loss functions. The ideas are similar, but we need to use the function $I(x)$ in a more subtle manner.

**Lemma C.14.** *Consider an instance $\mathcal{I} = (\Omega, f_{\mathcal{D}}, \ell, \alpha, \tau)$ of PrimalOpt with the loss function being TYPEN. Let $x^\star$ be $\operatorname{argmin}_{x \in \Omega} I(x)$. Then there is a value $\eta > 0$, such that any feasible solution $f$ must have $\text{Err}_{\ell,\alpha}(f_{\mathcal{D}}, f) \geq I(x^\star) + \eta$.*

*Proof.* As in the proof of Lemma C.13, we would like to argue that there is a positive $\varepsilon > 0$ such that any feasible solution places at least $\varepsilon$ amount of mass outside the interval $I_\varepsilon$. Since the optimum is finite, Corollary C.11 shows that $\Omega$ is bounded from below, i.e., it is of the form $[a, b]$ or $[a, \infty)$. Theorem C.10 now implies that $x^\star = a$.

Again, define the sets $A_i$ as in the proof of Lemma C.13, but now, we make $A_0$ start from the lower limit $a$ of $\Omega$. Thus, each of the $A_i$ will now be a single interval, with $A_i$ being to the right of $A_{i-1}$. Now we use the fact that when the loss function is TYPEN, both the functions $I^L(x)$ and $I^R(x)$ are monotonically increasing (Lemma C.8 and Lemma C.9). Further, for a point $x \in A_i$, Lemma C.8 shows that

$$I^L(x) - I^L(m + R) \geq \frac{\alpha}{2}\left(\ln |A_{i-1}| - \theta_{m+R}\right),)$$

because the c.d.f. of $f_{\mathcal{D}}$ at $m + R$ is at least $1/2$. Hence,

$$\int_{A_i \cap \Omega} (I^L(x) - I^L(x^\star)f(x)d\mu(x) \geq \frac{\alpha\beta_i}{2}(\ln |A_{i-1}| - \theta_{m+R}).$$

Summing the above over all $i \geq 1$, and proceeding as in the proof of Lemma C.13, we see that by choosing a small enough $\varepsilon$, (here $t_0$ denotes the right end-point of $A_0$)

$$\sum_{[t_0, \infty) \cap \Omega} I^L(x)f(x)d\mu(x) - I^L(x^\star)$$

can be made larger than a desired quantity $Q$, which depends only on the parameters of the input instance $\mathcal{I}$. Since $I^R(x)$ is an increasing function of $x$, we also get

$$\int_{[t_0, \infty) \cap \Omega} I(x)f(x)d\mu(x) \geq I(x^\star) + Q.$$

One remaining issue is that the integral on the l.h.s. does not include the terms for $[a, t_0)$ (recall that $I(x)$ can be negative). However, observe that $I(x)$ is an increasing function of $x$, and hence $I(x) \geq I(a)$ for all $x$. Therefore,

$$\begin{aligned}
\text{Err}_{\ell,\alpha}(f_{\mathcal{D}}, f) &= \int_\Omega I(x)f(x)d\mu(x) \\
&= \int_a^{t_0} I(x)f(x)d\mu(x) + \int_{[t_0, \infty) \cap \Omega} I(x)f(x)d\mu(x) \\
&\geq I(a) + I(a) + Q.
\end{aligned}$$

Again, by choosing $Q$ large enough, we can set the above to more than $2I(a)$. □

As an immediate consequence of this result, we see that for an instance of PrimalOpt satisfying conditions **(A0)**–**(A5)**, the optimal dual solution $\gamma^\star$ is non-zero.

**Theorem C.15.** *Consider an instance $\mathcal{I} = (\Omega, f_{\mathcal{D}}, \ell, \alpha, \tau)$ of PrimalOpt satisfying conditions* **(A0)–** **(A5)**. *Then there is an optimal dual solution $(\gamma^\star, \phi^\star)$ to DualOpt satisfying $\gamma^\star > 0$.*

*Proof.* Let $p^\star$ denote the optimal value of this instance (it is finite by Claim C.12). Corollary C.11 implies that $I(x)$ has a (unique) global minimum $x^\star$ in $\Omega$. Assume, for the sake of contradiction, that $\gamma^\star = 0$. We claim that $g(\gamma^\star, \phi^\star) = I(x^\star)$. Indeed, consider a function $f$ which places unit probability mass at $x^\star$. Then it is easy to verify that $L(f, \gamma^\star, \phi^\star) = I(x^\star)$. However, Lemma C.13 and Lemma C.14 show that $p^\star > I(x^\star)$, which is a contradiction (because by Theorem C.3, strong duality holds). $\qquad\square$

## C.6 Optimality conditions and uniqueness of optimal solution

We now show that, under suitable conditions, there is an optimal solution to an instance of PrimalOpt.

**Theorem C.16** (**Optimality condition**). *Consider an instance $\mathcal{I}$ of the optimization problem PrimalOpt defined on a space $\Omega$. Let $\ell, \alpha, \tau, f_{\mathcal{D}}$ denote the loss function, risk-averseness parameter, resource-information parameter, and the density of the true utility respectively in $\mathcal{I}$. Assume that the optimal value for this instance is finite and assumptions* **(A0)**, **(A3)** *hold. Let $(\gamma^\star, \phi^\star)$ be an optimal solution to the corresponding DualOpt with $\gamma^\star > 0$. Consider a function $f^\star \in$ Dom defined as follows:*

$$I_{f_{\mathcal{D}}, \ell, \alpha}(x) + \gamma^\star(1 + \ln f^\star(x)) + \phi^\star = 0, \qquad \forall x \in \Omega. \tag{12}$$

*Then $f^\star$ is an optimal solution to the instance $\mathcal{I}$.*

*Proof.* Let $p^\star$ denote the value of the optimal solution to $\mathcal{I}$. Since $p^\star$ is finite, Theorem C.15 shows that there is an optimal dual solution satisfying $\gamma^\star > 0$. Recall that

$$g(\gamma^\star, \phi^\star) := \min_{f \in \mathsf{Dom}} L(f, \gamma^\star, \phi^\star).$$

Let $f^\star$ be the function defined by (12) ($f^\star(x)$ is well-defined since $\gamma^\star > 0$, this is where we need strict positivity of $\gamma^\star$). We argue that $L(f^\star, \gamma^\star, \phi^\star) = g(\gamma^\star, \phi^\star)$.

Indeed, consider any other function $h \in$ Dom. Define a function $\theta : [0, \infty) \to \mathbb{R}$ as follows:

$$\theta(t) := L((1 - t)f^\star(x) + t\,h(x), \gamma^\star, \phi^\star) = L(f^\star(x) + te(x), \gamma^\star, \phi^\star),$$

where $e(x) = h(x) - f^\star(x)$. We first claim that $\theta(t)$ is a convex function of $t$.

**Claim C.17.** *The function $\theta : [0, 1] \to \mathbb{R}$ is a convex function.*

*Proof.* We observe that $\mathrm{Err}(f^\star + t \cdot e, f_{\mathcal{D}})$ is a linear function of $t$. The function $\gamma(\tau - \mathsf{Ent}(f^\star + t \cdot e))$ is a convex function of $t$, and $\phi\left(\int_\Omega (f(x) + te(x))d\mu(x) - 1\right)$ is a linear function of $t$. Therefore, $\theta(t)$ which is the sum of these three functions, is a convex function of $t$. $\qquad\square$

We shall show that $\frac{d\theta(t)}{dt}\big|_{t=0^+} = 0$. Along with the convexity of $\theta(t)$, this implies that $\theta(0) \leq \theta(1)$ and hence $L(f^\star, \gamma^\star, \phi^\star) \leq L(h, \gamma^\star, \phi^\star)$. A routine calculation shows that $\frac{d\theta(t)}{dt}\big|_{t=0^+}$ is equal to

$$\int_\Omega \left(I_{f_{\mathcal{D}}, \ell, \alpha}(x) + \gamma^\star(1 + \ln f^\star(x)) + \phi^\star\right) e(x)d\mu(x).$$

Inequality (12) implies that the above expression is 0. This proves the desired result. Thus, we have shown that $L(f^\star, \gamma^\star, \phi^\star) = g(\gamma^\star, \phi^\star)$, and therefore, $f^\star$ is an optimal solution to the instance $\mathcal{I}$. $\quad\square$

We now show the uniqueness of the optimal solution to PrimalOpt. Consider an instance of this optimization problem specified by $\Omega$, loss function $\ell$, density $f_{\mathcal{D}}$ and parameters $\alpha, \tau$. Let $\mathcal{F}$ denote the set of feasible densities, i.e.,

$$\mathcal{F} := \{f : f \in \mathsf{Dom}, f \text{ is a density on } \Omega \text{ and } \int_\Omega f(x) \ln f(x)d\mu(x) \leq -\tau\}.$$

**Lemma C.18.** $\mathcal{F}$ *is strictly convex.*

*Proof.* Let $f_1, f_2 \in \mathcal{F}$ and $\lambda$ be a parameter in $[0, 1]$. We first show that the density $f$ defined by $f(x) := \lambda f_1(x) + (1 - \lambda) f_2(x)$, $x \in \Omega$, is also in $\mathcal{F}$. Clearly, $f$ is a density, because $f(x) \geq 0$ and

$$\int_\Omega f(x) d\mu(x) = \lambda \int_\Omega f_1(x) d\mu(x) + (1 - \lambda) \int_\Omega f_2(x) d\mu(x) = \lambda + (1 - \lambda) = 1.$$

Hence, the fact that $g(y) := y \ln y$ is a strongly convex function on $\mathbb{R}_{\geq 0}$ implies that

$$f(x) \ln f(x) \leq \lambda f_1(x) \ln f_1(x) + (1 - \lambda) f_2(x) \ln f_2(x)),$$

with equality if and only if $f_1(x) = f_2(x)$. Integrating both sides, and using the fact that $f_1, f_2$ belong to $\mathcal{F}$, we get

$$\int_\Omega f(x) \ln f(x) d\mu(x) \leq -\lambda \tau - (1 - \lambda) \tau$$
$$= -\tau.$$

Thus, if $f_1(x)$ and $f_2(x)$ differ on a set of positive measure, then $f(x) \ln f(x) < \lambda f_1(x) + (1 - \lambda) f_2(x)$ on a set of positive measure. Integrating both sides, we get $\mathsf{Ent}(f) > \tau$. This shows that the set $\mathcal{F}$ is strictly convex. $\qquad\square$

Strict convexity of $\mathcal{F}$ now allows us to show the uniqueness of the optimal solution.

**Theorem C.19** (**Uniqueness of optimal solution**). *Consider an instance $\mathcal{I} = (\Omega, f_\mathcal{D}, \ell, \alpha, \tau)$ of PrimalOpt satisfying the property that the optimal value is finite and there is an optimal dual solution with $\gamma^\star > 0$. Then there is a unique optimal solution to this instance.*

*Proof.* Let $p^\star$ denote the optimal value of this instance. Theorem C.16 already shows the existence of an optimal solution. We show the uniqueness of an optimal solution. We have assumed that there is an optimal dual solution $(\gamma^\star, \phi^\star)$ such that $\gamma^\star > 0$. The complementary slackness condition shows that for any optimal solution $f$ to the instance $\mathcal{I}$, $\mathsf{Ent}(f)$ must equal $\tau$.

Suppose, for the sake of contradiction, that there are two distinct optimal solutions $f_1^\star$ and $f_2^\star$ to $\mathcal{I}$ (i.e., $f_1^\star(x) \neq f_2^\star$ on a subset of positive measure). Consider a solution $f^\star = t f_1^\star + (1 - t) f_2^\star$ for some $t \in (0, 1)$. Linearity of $\mathrm{Err}_{\ell, \alpha}(f_\mathcal{D}, f)$ shows that $\mathrm{Err}_{\ell, \alpha}(f_\mathcal{D}, f^\star) = p^\star$ as well. Now $f^\star$ is also a feasible solution by Lemma C.18; in fact, this lemma shows that $\mathsf{Ent}(f^\star) > \tau$. But this contradicts the fact that every optimal solution must have entropy equal to $\tau$. Thus, we see that there must be a unique optimal solution to the instance $\mathcal{I}$. $\qquad\square$

Combining Lemma F.2, Theorem C.16 and Theorem C.19, we see that Theorem C.1 holds.

## C.7  Conditions under which optimal solution is $f_\mathcal{D}$

We show that our optimization framework can output the true utility $f_\mathcal{D}$ for a choice of the parameters $\alpha, \tau$, and the loss function $\ell$.

**Theorem C.20** (**Parameters that recover the true utility**). *Consider an instance $\mathcal{I} = (\Omega, f_\mathcal{D}, \ell, \alpha, \tau)$ of the optimization problem PrimalOpt. Assume that the density $f_\mathcal{D}$ has finite entropy. Suppose, $\alpha := 1, \tau := \mathsf{Ent}(f_\mathcal{D})$, and the loss function $\ell(x, v) := \ln f_\mathcal{D}(v) - \ln f_\mathcal{D}(x)$. Then the density $f_\mathcal{D}$ is the unique optimal solution to PrimalOpt for $\mathcal{I}$.*

*Proof.* We claim that, for the values of dual variables $\gamma := 1$ and $\phi := \mathsf{Ent}(f_\mathcal{D}) - 1$, $g(\gamma, \phi) = L(f_\mathcal{D}, \gamma, \phi)$. Towards this, we show that $f_\mathcal{D}$ minimizes $L(f, \gamma, \phi)$ over all $f \in \mathsf{Dom}$. We first show that $f_\mathcal{D}$ satisfies the condition (12) with $\gamma^\star = \gamma, \phi^\star = \phi, f^\star = f_\mathcal{D}$; it shall then follow from exactly

the same arguments as in Theorem C.16 that $f_\mathcal{D} = \operatorname{argmin}_{f \in \mathsf{Dom}} L(f, \gamma, \phi)$. Now, we check that $f_\mathcal{D}$ satisfies condition (12) (recall that the loss function $\ell(x, v) := \ln f_\mathcal{D}(v) - \ln f_\mathcal{D}(x)$):

$$I(x) + \gamma(1 + \ln f_\mathcal{D}(x)) + \phi = \int_\Omega \ell(x, v) f_\mathcal{D}(v) d\mu(v) + \gamma(1 + \ln f_\mathcal{D}(x)) + \phi$$

$$= \int_\Omega f_\mathcal{D}(v) \ln f_\mathcal{D}(v) d\mu(v) - \ln f_\mathcal{D}(x) \int_\Omega f_\mathcal{D}(v) d\mu(v) + (1 + \ln f_\mathcal{D}(x)) + \mathsf{Ent}(f_\mathcal{D}) - 1$$

$$= -\mathsf{Ent}(f_\mathcal{D}) - \ln f_\mathcal{D}(x) + (1 + \ln f_\mathcal{D}(x)) + \mathsf{Ent}(f_\mathcal{D}) - 1 = 0.$$

This proves the desired claim.

Thus,

$$g(\gamma, \phi) = L(f_\mathcal{D}, \gamma, \phi) = \mathrm{Err}_{\ell, \alpha}(f_\mathcal{D}, f_\mathcal{D}).$$

Thus, $g(\gamma, \phi)$ is equal to the objective function value of PrimalOpt at $f = f_\mathcal{D}$. Hence, $f_\mathcal{D}$ is an optimal solution to PrimalOpt for $\mathcal{I}$. In order to prove uniqueness, note that the above argument also yields an optimal solution $(\gamma, \phi)$ to DualOpt. Since $\gamma > 0$, complementary slackness conditions imply that any optimal solution must have entropy exactly equal to $\tau$. Now, arguing as in Theorem C.19, we see that there is a unique optimal solution. $\qquad\square$

# D   Derivation of output density for a single individual

In this section, we consider the setting when a single individual with true utility $u$ is being evaluated. In this case, the input density $f_\mathcal{D}(x)$ is specified by the Dirac-delta function centered at $v$, denoted $\delta_v(x)$. Using Theorem 3.1, we can characterize the output density as follows:

**Lemma D.1.** *Consider an instance $\mathcal{I} = (\Omega, f_\mathcal{D}, \ell, \alpha, \tau)$ of (OptProg) where $\Omega = \mathbb{R}$, $f_\mathcal{D} = \delta_v(x)$ for some real $v$, $\ell(x, u) = (x - u)^2$, and $\alpha$ and $\tau$ are arbitrary real parameters. Then the optimal density is given by*

$$f^\star(x) = \begin{cases} Ke^{-\frac{(x-v)^2}{\gamma^\star}} & \text{if } x \leq v \\ Ke^{-\frac{\alpha(x-v)^2}{\gamma^\star}} & \text{otherwise} \end{cases},$$

*where $K$ is the normalization constant and $\gamma^\star$ is the optimal dual variable for the entropy constraint. The mean of this density is equal to $v - \sqrt{\frac{\gamma^\star}{\pi}} \cdot \frac{\sqrt{\alpha} - 1}{\sqrt{\alpha}}$.*

*Proof.* We first evaluate the integral $I(x)$ as follows:

$$I(x) := \int_\Omega f_\mathcal{D}(u) \ell_\alpha(x, u) dv = \alpha \int_{-\infty}^x \delta_v(u)(x - u)^2 + \int_x^\infty \delta_v(u)(x - u)^2.$$

When $x \leq v$, the first integral on the r.h.s. is 0, and hence, the above integral is equal to $(x - v)^2$. Similarly, if $x > v$, the above integral is equal to $\alpha(x - v)^2$. The expression for $f^\star(x)$ in the statement of the Lemma now follows from (12).

A routine calculation shows that the normalization constant $K$ is equal to $\frac{2\sqrt{\alpha}}{\sqrt{\pi\gamma^\star}(1+\sqrt{\alpha})}$. Now the mean of this density turns out to be

$$v - \sqrt{\frac{\gamma^\star}{\pi}} \cdot \frac{\sqrt{\alpha} - 1}{\sqrt{\alpha}}.$$

$\qquad\square$

# E  Connection to the Gibbs equation

**Theorem E.1** (**Gibbs equation**).  *Consider an instance $\mathcal{I} = (\Omega, f_{\mathcal{D}}, \ell, \alpha, \tau)$ of the optimization problem PrimalOpt that satisfies the assumptions of Theorem C.1. Then, the following holds:*

$$-\gamma^{\star} \ln Z^{\star} = \mathrm{Err}_{\ell,\alpha}(f^{\star}, f_{\mathcal{D}}) + \gamma^{\star} \mathsf{Ent}(f^{\star}). \tag{13}$$

*Here, $f^{\star}(x) \propto e^{-\frac{I(x)}{\gamma^{\star}}}$ is the solution to PrimalOpt, $Z^{\star} := \int e^{-\frac{I(x)}{\gamma^{\star}}} d\mu(x)$, and $\gamma^{\star} > 0$ is the solution to DualOpt. Recall that $I(x) := \int_{\Omega} \ell_{\alpha}(x, v) f_{\mathcal{D}}(v) d\mu(v)$.*

*Proof.* Theorem C.1 implies that there exists $f^{\star}(x)$ and $\gamma^{\star} > 0$ such that

$$f^{\star}(x) \propto e^{-\frac{I(x)}{\gamma^{\star}}}.$$

Thus, if we let $Z^{\star} := \int e^{-\frac{I(x)}{\gamma^{\star}}} d\mu(x)$, then

$$f^{\star}(x) = \frac{e^{-\frac{I(x)}{\gamma^{\star}}}}{Z^{\star}}.$$

Thus, from the optimality condition (12), we obtain that there is a $\phi^{\star}$ such that

$$I(x) + \gamma^{\star}(1 + \ln f^{\star}(x)) + \phi^{\star} = 0.$$

Since $\gamma^{\star} > 0$, we can divide by it to obtain $Z^{\star} = e^{1 + \frac{\phi^{\star}}{\gamma^{\star}}}$. We integrate the above with respect to the density $f^{\star}(x)$ to get

$$\mathrm{Err}_{\ell,\alpha}(f^{\star}, f_{\mathcal{D}}) + \gamma^{\star} - \gamma^{\star}\tau + \phi^{\star} = 0.$$

Thus, we obtain:

$$\gamma^{\star}\tau = \gamma^{\star} \mathsf{Ent}(f^{\star}) = \mathrm{Err}_{\ell,\alpha}(f^{\star}, f_{\mathcal{D}}) + \gamma^{\star} \ln Z^{\star}. \tag{14}$$

Rearranging this equation we obtain the theorem. $\qquad\square$

In analogy with the Gibbs equation in statistical physics [94], $Z^{\star}$ can be viewed as the *partition function* corresponding to the energy function $I(x)$, $\gamma^{\star}$ corresponds to the *temperature*, $-\gamma^{\star} \ln Z^{\star}$ corresponds to the *free energy* and $\mathrm{Err}(f^{\star}, f_{\mathcal{D}})$ is the *internal energy*.

It follows from the theorem that we can write $f^{\star}(x)$ as

$$f^{\star}(x) = e^{-\tau} \exp\left(-\frac{I(x) - \mathrm{Err}_{\ell,\alpha}(f^{\star}, f_{\mathcal{D}})}{\gamma^{\star}}\right). \tag{15}$$

# F  Effect of changing the resource-information parameter $\tau$

## F.1  Effect of decreasing $\tau$

**Theorem F.1** (**Effect of decreasing $\tau$**).  *Fix $\alpha \geq 1$, a density $f_{\mathcal{D}}$, and loss function $\ell$. Assume that the function $I_{f_{\mathcal{D}},\ell,\alpha}(x)$ satisfies condition (**A5**), and let $x^{\star} := \operatorname{argmin}_{x \in \Omega} I_{f_{\mathcal{D}},\ell,\alpha}(x)$. Given a $\tau$, let $f_{\tau}^{\star}$ denote the optimal solution to PrimalOpt. For every $\delta > 0$, there exists a value $T_{\delta}$ such that when $\tau \leq T_{\delta}$, the solution $f_{\tau}^{\star}$ has the following property: For every $x \in \Omega, |x - x^{\star}| \geq \delta$, we have*

$$\frac{f_{\tau}^{\star}(x)}{f_{\tau}^{\star}(x^{\star})} \leq \delta.$$

*In other words, the density outside an interval of length $\delta$ around $x^{\star}$ has a much smaller value than at $x^{\star}$.*

In order to prove the above result, we first show that the optimal dual value $\gamma_\tau^\star$ goes to 0 as $\tau$ goes to $-\infty$. Then we shall use Theorem E.1 to show that the optimal density is highly sensitive to small changes in $I(x)$.

**Lemma F.2.** *Fix an interval $\Omega \subseteq \mathbb{R}$, parameter $\alpha \geq 1$, density $f_\mathcal{D}$ of true utility and loss function $\ell$. Assume that the function $I_{f_\mathcal{D},\ell,\alpha}(x)$ has a unique global minimum. Let $\mathcal{I}_\tau$ denote the instance $(\Omega, f_\mathcal{D}, \ell, \alpha, \tau)$. Let $\gamma_\tau^\star$ be the optimal Lagrange variable corresponding to this instance (assuming it has a non-empty solution). Then*

$$\lim_{\tau \to -\infty} \gamma_\tau^\star = 0.$$

*Proof.* We first observe that the instance $\mathcal{I}_\tau$ will always have a feasible solution for small enough $\tau$. Indeed, when $e^\tau$ is less than the length of $\Omega$, there is always a feasible solution to $\mathcal{I}_\tau$. Let $\tau_0$ be a value of $\tau$ for which the instance $\mathcal{I}_\tau$ has a feasible solution. For sake of brevity, let $I(x)$ denote $I_{f_\mathcal{D},\ell,\alpha}(x)$, and let $x^\star := \operatorname{argmin}_{x \in \Omega} I(x)$. We first argue that $\mathrm{Err}(f_\mathcal{D}, f_\tau^\star)$ remains in a bounded range.

**Claim F.3.** *Consider a value of $\tau \leq \tau_0$. Let $f_\tau^\star$ be the optimal solution to the instance $\mathcal{I}_\tau$. Then*

$$I(x^\star) \leq \mathrm{Err}_{\ell,\alpha}(f_\mathcal{D}, f_\tau^\star) \leq \mathrm{Err}_{\ell,\alpha}(f_\mathcal{D}, f_{\tau_0}^\star).$$

*Proof.* The first inequality follows from the fact that

$$\mathrm{Err}_{\ell,\alpha}(f_\mathcal{D}, f_\tau^\star) = \int_\Omega I(x) f_\tau^\star(x) d\mu(x) \geq I(x^\star).$$

The second inequality follows from the fact that the solution $f_{\tau_0}^\star$ is also a feasible solution for the instance $\mathcal{I}_\tau$. $\qquad\square$

Suppose for the sake of contradiction,

$$\lim_{\tau \to -\infty} \gamma_\tau^\star > 0.$$

In other words, $\theta_0 := \limsup_{\tau \to -\infty} \gamma_\tau^\star > 0$. It follows that there is an infinite sequence $A := (\tau_0 > \tau_1 > \cdots)$ going to $-\infty$ such that $\gamma_{\tau_i}^\star \geq \theta_0/2$ for all $\tau_i \in A$.

Consider an interval of finite but non-zero length in $\Omega$ – let $U = [s, t]$ be such an interval. Since $U$ is closed and $I(x)$ is a continuous function, there are finite values $a_0, b_0$ such that $a_0 \leq I(x) \leq b_0$ for all $x \in U$. Thus we get:

**Claim F.4.** *There is a positive real $\eta$, such that*

$$\exp\left(\frac{\mathrm{Err}_{\ell,\alpha}(f_\mathcal{D}, f_\tau^\star) - I(x)}{\gamma_\tau^\star}\right) \geq \eta$$

*holds for all $x \in U, \tau \in A$.*

*Proof.* It follows from Claim F.3 and the observation above that for all $x \in U$ and all $\tau \leq \tau_0$, $|\mathrm{Err}_{\ell,\alpha}(f_\mathcal{D}, f_\tau^\star) - I(x)|$ lies in the range $[I(x^\star) - b_0, \mathrm{Err}_{\ell,\alpha}(f_\mathcal{D}, f_\tau^\star) + a_0]$. Since $\gamma_\tau \geq \theta_0/2$ for all $\tau \in A$, the result follows. $\qquad\square$

The above claim along with Theorem E.1 shows that $f_\tau^\star(x) \geq \eta\, e^{-\tau}$ for all $x \in U, \tau \in A$. Since $A$ contains an infinite sequence of values going to $-\infty$, we can choose a value $\tau \in A$ such that $\eta e^{-\tau} > 1/|U|$. But then $f_\tau^\star(x) > \frac{1}{|U|}$ for all $x \in U$, which is not possible because $f^\star$ is a density. This proves the lemma. $\qquad\square$

We now prove Theorem F.1. Consider the instance $\mathcal{I}$ as stated in this statement of this theorem. Let $\tau_0$ be a value of the information-resource parameter for which there is a density with entropy $\tau$ in $\Omega$ (i.e., when PrimalOpt has a feasible solution). Recall that $x^\star = \operatorname{argmin}_{x \in \Omega} I(x)$. Consider a $\delta > 0$. Assumption **(A5)** shows that there is a value $\varepsilon_\delta > 0$ such that $I(x) - I(x^\star) > \varepsilon_\delta$ for all $x$ satisfying

$|x - x^\star| \geq \delta$. Now consider an $x$ such that $|x - x^\star| \geq \delta$. Using Theorem E.1, we see that, for all $\tau \leq \tau_0$

$$\frac{f_\tau^\star(x)}{f_\tau^\star(x^\star)} = \exp\left(\frac{I(x^\star) - I(x)}{\gamma_\tau^\star}\right) \leq \exp\left(-\varepsilon_\delta / \gamma_\tau^\star\right),$$

where the last inequality follows from the fact that $\gamma_\tau^\star$ is positive. Now, Lemma F.2 shows that there a value $T_\delta$ such that for all $\tau \leq T_\delta$, $0 < \gamma_\tau^\star \leq \frac{\delta\,\varepsilon_\delta}{\ln(1/\delta)}$. Therefore, for all $x, |x - x^\star| \geq \delta$,

$$\frac{f_\tau^\star(x)}{f_\tau^\star(x^\star)} \leq \delta.$$

This proves Theorem F.1.

### F.2 Effect of increasing $\tau$

In this section, we consider the effect of an increase in $\tau$ on the variance and the mean of the optimal density.

**Theorem F.5** (**Effect of increasing $\tau$ on variance**). *Consider a continuous density $f_\mathcal{D}$ on $\mathbb{R}$, loss function $\ell$ and information-resource parameter $\alpha$. For a given risk-averse parameter $\tau$, let $\mathcal{I}_\tau$ denote the instance $(\mathbb{R}, f_\mathcal{D}, \ell, \alpha, \tau)$ of PrimalOpt. Let $f_\tau^\star$ be the optimal solution to the instance $\mathcal{I}_\tau$. Then the variance of $f_\tau^\star$ is at least $\frac{1}{2\pi} e^{2\tau-1}$.*

The proof relies on the following result.

**Theorem F.6** (**Gaussian maximizes entropy; Theorem 3.2 in [48]**). *For a continuous probability density function $f$ on $\mathbb{R}$ with variance $\sigma^2$, $\mathsf{Ent}(f) \leq \frac{1}{2} + \frac{1}{2}\ln\left(2\pi\sigma^2\right)$ with equality if and only if $f$ is a Gaussian density with variance $\sigma^2$.*

*Proof of Theorem F.5.* Consider the instance $\mathcal{I}_\tau$ and the optimal solution $f_\tau^\star$ for this instance. Since $f_\tau^\star$ is a feasible solution, $\mathsf{Ent}(f_\tau^\star) \geq \tau$. If $f_\tau^\star$, has unbounded variance, then the desired result follows trivially. Hence, assume that $f^\star$ has bounded variance, say $\sigma^2$. Now, Theorem F.6 shows that

$$\mathsf{Ent}(f_\tau^\star) \leq \frac{1}{2} + \frac{1}{2}\ln(2\pi\sigma^2).$$

Using the fact that $\mathsf{Ent}(f_\tau^\star) \geq \tau$, the above inequality implies that $\sigma^2 \geq \frac{1}{2\pi} e^{2\tau-1}$. This proves the desired result. □

We now show that the mean of the optimal density also increases with increasing $\tau$ when the input density $f_\mathcal{D}$ is supported on $[0, \infty)$.

**Theorem F.7** (**Effect of increasing $\tau$ on mean**). *Consider a continuous density $f_\mathcal{D}$ on $\Omega := [0, \infty)$, loss function $\ell$ and information-resource parameter $\alpha$. For a given risk-averse parameter $\tau$, let $\mathcal{I}_\tau$ denote the instance $(\Omega, f_\mathcal{D}, \ell, \alpha, \tau)$ of PrimalOpt, and $f_\tau^\star$ denotes the optimal solution to the instance $\mathcal{I}_\tau$. Then the mean of $f_\tau^\star$ is at least $e^{\tau-1}$.*

The proof relies on the following result.

**Theorem F.8** (**Exponential maximizes entropy; Theorem 3.3 in [48]**). *For a continuous probability density function $f$ on $[0, \infty)$ with mean $\lambda$, $\mathsf{Ent}(f) \leq 1 + \ln(\lambda)$.*

*Proof of Theorem F.7.* The proof proceeds along similar lines as that of Theorem F.5. We know that $\mathsf{Ent}(f_\tau^\star) \geq \tau$ because it is a feasible solution to the instance $\mathcal{I}_\tau$. If $f_\tau^\star$ has unbounded mean, we are done; therefore, assume its mean, denoted $\lambda$, is finite. Theorem F.8 shows that $\mathsf{Ent}(f) \leq 1 + \ln\lambda$. Since $\mathsf{Ent}(f_\tau^\star) \geq \tau$, we see that $\lambda \geq e^{\tau-1}$. This proves the theorem. □

# G  Effect of changing the risk-averseness parameter $\alpha$

**Theorem G.1** (**Monotonicity of** $I(x)$ **with respect to** $\alpha$). *Consider an instance $\mathcal{I} = (\Omega, f_{\mathcal{D}}, \ell, \alpha, \tau)$ of the optimization problem PrimalOpt. Then, for any $x \in \Omega$, $I_{f_{\mathcal{D}}, \ell, \alpha}(x)$ is an increasing function of $\alpha$.*

*Proof.* By definition

$$I_\alpha(x) \coloneqq I_{f_{\mathcal{D}}, \ell, \alpha}(x) = \int_{v \leq x} \ell_\alpha(x, v) f_{\mathcal{D}}(v) d\mu(v) + \int_{v > x} \ell(x, v) f_{\mathcal{D}}(v) d\mu(v).$$

Recall from (4), that for any $x, v \in \Omega$, $\ell_\alpha(x, v) \coloneqq \ell(x, v)$ if $x < v$ and $\ell_\alpha(x, v) \coloneqq \alpha \ell(x, v)$ when $x \geq v$. Since our model requires $\ell(x, v) \geq 0$ whenever $x \geq v$, $\ell_\alpha(x, v) \geq \ell_{\alpha'}(x, v)$ for any $x \geq v$ and $\alpha \geq \alpha'$. Thus, we have that $I_\alpha(x) \geq I_{\alpha'}(x)$ for all $x \in \Omega$. $\qquad\square$

**Theorem G.2** (**Monotonicity of** Err **with respect to** $\alpha$). *Consider an instance $\mathcal{I}_\alpha = (\Omega, f_{\mathcal{D}}, \ell, \alpha, \tau)$ of the optimization problem PrimalOpt. Suppose $\mathcal{I}_\alpha$ satisfies the assumption of Theorem C.1 and let $f_\alpha^\star$ be the optimal solution to instance $\mathcal{I}_\alpha$. Then, the function $\mathrm{Err}_{\ell, \alpha}(f_\alpha^\star, f_{\mathcal{D}})$ is an increasing function of $\alpha$.*

*Proof.* As noted in Section C, if the instance $\mathcal{I}$ satisfies the assumptions for $\alpha = 1$, then the instances obtained by changing $\alpha$ continue to satisfy the assumptions needed in Theorem C.1. Thus, we may assume the optimal solution exists for each version of $\mathcal{I}$ where we vary $\alpha \geq 1$, and let $f_\alpha^\star$ denote the optimal density. We first show that for any fixed density $f$, $\mathrm{Err}_{\ell, \alpha}(f, f_{\mathcal{D}})$ is an increasing function of $\alpha$. This is so because

$$\mathrm{Err}_{\ell, \alpha}(f, f_{\mathcal{D}}) = \int_{v \in \Omega} \left( \int_{x < v} \ell(x, v) f(x) d\mu(x) \right) f_{\mathcal{D}}(v) d\mu(v)$$
$$+ \int_{v \in \Omega} \left( \int_{x \geq v} \ell_\alpha(x, v) f(x) d\mu(x) \right) f_{\mathcal{D}}(v) d\mu(v)$$

and $\ell_\alpha(x, v)$ is an increasing function of $\alpha$ for any $x, v \in \Omega$. Consider two values of the parameter $\alpha$: $1 \leq \alpha_1 < \alpha_2$. Note that the instances corresponding to both $\alpha_1$ and $\alpha_2$ are feasible as $\alpha$ only appears in the objective and, hence, does not affect feasibility. Suppose for the sake of contradiction that $\mathrm{Err}_{\ell, \alpha_2}(f_{\alpha_2}^\star, f_{\mathcal{D}}) < \mathrm{Err}_{\ell, \alpha_1}(f_{\alpha_1}^\star, f_{\mathcal{D}})$. $f_{\alpha_2}^\star$ satisfies $\mathrm{Ent}(f_{\alpha_2}^\star) \geq \tau$ as it is a feasible solution of the problem defined by $\alpha_2$ and, hence, it is also a feasible solution for the problem instance defined by $\alpha_1$. This and the definition of $f_{\alpha_1}^\star$ imply that $\mathrm{Err}_{\ell, \alpha_1}(f_{\alpha_1}^\star, f_{\mathcal{D}}) \leq \mathrm{Err}_{\ell, \alpha_1}(f_{\alpha_2}^\star, f_{\mathcal{D}})$. Thus, we get $\mathrm{Err}_{\ell, \alpha_2}(f_{\alpha_2}^\star, f_{\mathcal{D}}) < \mathrm{Err}_{\ell, \alpha_1}(f_{\alpha_2}^\star, f_{\mathcal{D}})$, which contradicts the (above observed) monotonicity of $\mathrm{Err}_{\ell, \alpha}(f_{\alpha_2}^\star, f_{\mathcal{D}})$ with respect to $\alpha$. $\qquad\square$

# H  Gaussian density

The Gaussian density is defined as follows over $\Omega = \mathbb{R}$ and has parameters $m \in \mathbb{R}$ and $\sigma$:

$$f_{\mathcal{G}}(x) \coloneqq \frac{1}{\sqrt{2\pi\sigma^2}} e^{-\frac{(x-m)^2}{2\sigma^2}}, \quad x \in \mathbb{R}.$$

$m$ is the mean and $\sigma^2$ is the variance. The differential entropy of $f_{\mathcal{G}}$ is $\frac{1}{2} + \frac{1}{2}\ln(2\pi\sigma^2)$ [148]. We consider the loss function to be $\ell(x, v) \coloneqq (x - v)^2$. First, we compute the expression of $I(x)$ which we use to verify the applicability of Theorem C.1 with the above parameters.

**Lemma H.1** (**Expression for** $I_\alpha(x)$). *Consider an instance $\mathcal{I} = (\Omega, f_{\mathcal{G}}, \ell, \alpha, \tau)$ of PrimalOpt where $\Omega = \mathbb{R}$, $\ell(x, v) \coloneqq (x - v)^2$, and $f_{\mathcal{G}}$ is the Gaussian density with mean $m$ and variance $\sigma^2$. Then*

$$I(x) = (\alpha - 1)\sigma^2 \left( (w^2 + 1)\Phi(w) + w\phi(w) \right) + \sigma^2 \left( w^2 + 1 \right),$$

*where $w \coloneqq \frac{x-m}{\sigma}$, $\phi(x) \coloneqq \frac{1}{\sqrt{2\pi}} e^{-\frac{x^2}{2}}$, and $\Phi(x) \coloneqq \int_{-\infty}^{x} \phi(x) d\mu(x)$ denotes the cumulative distribution function of the Gaussian density.*

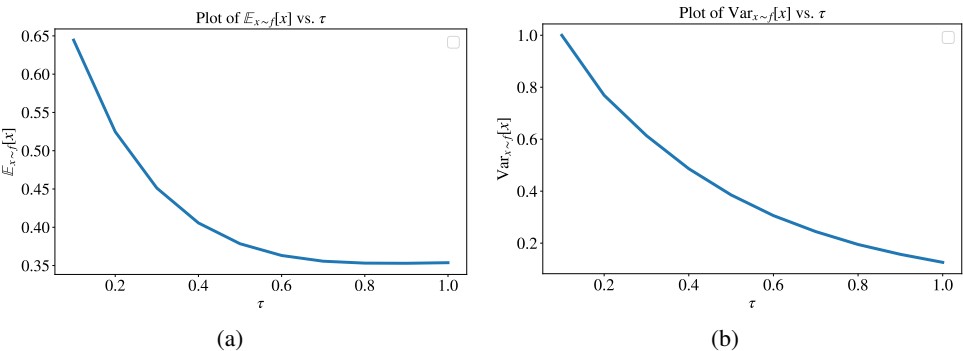

Figure 3: Mean and variance of the output density $f$, i.e., $\mathbb{E}_{x \sim f}[x]$ and $\text{Var}_{x \sim f}[x]$, as a function of $\tau$ when $f_{\mathcal{D}}$ is the standard normal density and $\alpha$ is fixed to 2.

*Proof.* By definition,

$$I(x) = \alpha \int_{-\infty}^{x} (x-v)^2 f_{\mathcal{G}}(v) d\mu(v) + \int_{x}^{\infty} (x-v)^2 f_{\mathcal{G}}(v) d\mu(v).$$

We first make a change of variables. Let $w := \frac{x-m}{\sigma}$ and $y := \frac{v-m}{\sigma}$. Thus, the density of $y$ is Gaussian with mean 0 and variance 1. We denote this density by $\phi$. The above integral becomes

$$\alpha\sigma^2 \int_{-\infty}^{w} (w-y)^2 f_{\mathcal{G}}(v) d\mu(v) + \sigma^2 \int_{w}^{\infty} (w-y)^2 f_{\mathcal{G}}(v) d\mu(v)$$

$$= \alpha\sigma^2 \int_{-\infty}^{w} (w^2 + y^2 - 2wy)\phi(y) d\mu(y) + \sigma^2 \int_{w}^{\infty} (w^2 + y^2 - 2wy)\phi(y) d\mu(y)$$

$$= \alpha\sigma^2 \int_{-\infty}^{w} (w^2 + y^2 - 2wy)\phi(y) d\mu(y) - \sigma^2 \int_{-\infty}^{w} (w^2 + y^2 - 2wy)\phi(y) d\mu(y)$$

$$\quad + \sigma^2 \int_{-\infty}^{\infty} (w^2 + y^2 - 2wy)\phi(y) d\mu(y)$$

$$= (\alpha-1)\sigma^2 \left( w^2\Phi(w) + \Phi(w) - w\phi(w) + 2w\phi(w) \right)] + \sigma^2 \left( w^2 + 1 \right)$$

$$= (\alpha-1)\sigma^2 \left( (w^2 + 1)\Phi(w) + w\phi(w) \right) + \sigma^2 \left( w^2 + 1 \right).$$

$\square$

**Applicability of Theorem C.1.** We verify that any instance $\mathcal{I} = (\Omega, f_{\mathcal{G}}, \ell, \alpha, \tau)$ of PrimalOpt defined by $\Omega = \mathbb{R}$, $f_{\mathcal{G}}$ as a Gaussian density, a finite $\alpha$, and $\tau$ satisfies the assumptions in Theorem C.1. Since $\Omega = \mathbb{R}$, **(A0)** holds for any finite $\tau$. **(A1)** and **(A2)** hold due to the choice of the loss function. **(A3)** holds since $f_{\mathcal{G}}$ has a finite variance. **(A4)** holds with, e.g., $R = \sigma^2$. In Lemma H.1, we compute

$$I(x) = (\alpha-1)\sigma^2 \left( (w^2 + 1)\Phi(w) + w\phi(w) \right) + \sigma^2 \left( w^2 + 1 \right),$$

where $w = \frac{x-m}{\sigma}$. From this expression, it follows that $I(x)$ is differentiable and

$$\frac{\partial^2 I(x)}{\partial^2 x} = 2 + 2(\alpha-1)\Phi(w).$$

Since $\alpha \geq 1$ and $\Phi(\cdot)$ is non-negative, it follows that $I(x)$ is strongly convex and, hence, has a unique global minimum. Therefore, **(A5)** holds. Since assumptions **(A0)–(A5)** hold, we invoke Theorem C.1 to deduce the form of $f^{\star}$.

Figure 3 plots the mean and the variance of the output density $f^{\star}$ as a function of the parameter $\tau$ when the input density $f_{\mathcal{D}}$ is the standard normal density. Figure 4 plots the mean and the variance of the output density as a function of the parameter $\alpha$ in this setting.

**Theorem H.2 (Expression for $f^{\star}$ when $\alpha = 1$).** *Consider an instance $\mathcal{I} = (\Omega, f_{\mathcal{D}}, \ell, 1, \tau)$ of PrimalOpt where $\Omega = \mathbb{R}$, $\ell(x, v) := (x-v)^2$, and $f_{\mathcal{G}}$ is the Gaussian density with mean $m$ and variance $\sigma^2$. Let $f^{\star}$ be the optimal solution of $\mathcal{I}$. Then $f^{\star}$ is a Gaussian with mean $m$ and variance $\frac{1}{2\pi}e^{2\tau-1}$.*

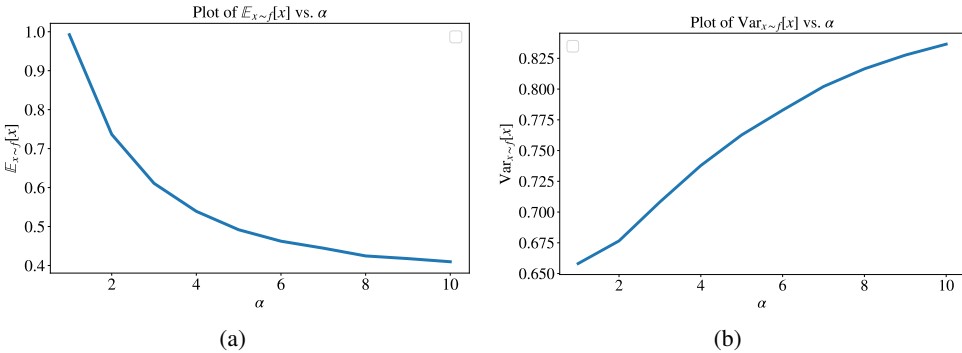

Figure 4: Mean and variance of the output density $f$, i.e., $\mathbb{E}_{x \sim f}[x]$ and $\text{Var}_{x \sim f}[x]$, as a function of $\alpha$ when $f_{\mathcal{D}}$ is the standard normal density and $\tau$ is fixed to the entropy of $f_{\mathcal{D}}$.

Thus, for $\alpha = 1$, increasing $\tau$ does not change the mean, but increases the variance of the output density.

*Proof.* For $\alpha = 1$, Lemma H.1 implies that

$$I(x) = (x - m)^2 + \sigma^2.$$

As shown earlier, by Theorem C.1, $f^\star$ has the following form

$$f^\star(x) \propto e^{\frac{-(x-m)^2 + \sigma^2}{\gamma^\star}} \propto e^{\frac{-(x-m)^2}{\gamma^\star}}.$$

where the proportionality constant and $\gamma^\star$ are determined by $\int_{\mathbb{R}} f^\star(x) d\mu(x) = 1$ and $\text{Ent}(f^\star) = \tau$. $f^\star$ is a Gaussian density with mean $m$ and variance $\frac{\gamma^\star}{2}$ and, hence, $\text{Ent}(f^\star) = \frac{1}{2} + \frac{1}{2} \ln(\pi \gamma^\star)$ [148]. Since $\text{Ent}(f^\star) = \tau$, the previous equality implies that $\gamma^\star = \frac{1}{\pi} e^{2\tau - 1}$. It follows that $f^\star(x) = \frac{1}{\sqrt{2e^{2\tau-1}}} \exp\left(-\pi \frac{(x-m)^2}{e^{2\tau-1}}\right)$ which is the Gaussian density with mean $m$ and variance $\frac{1}{2\pi} e^{2\tau-1}$. $\qquad\square$

**Shift in mean with increase in $\alpha$.** We consider the effect of the parameter $\alpha$ on the mean of the output density. We consider an instance $\mathcal{I}_1 = (\Omega, f_{\mathcal{D}}, \ell, \alpha, \tau)$ where $\Omega = \mathbb{R}$, $f_{\mathcal{D}}$ is the normal density $N(\mu, \sigma^2)$, $\ell(v, x) = (x - v)^2$, $\tau = \text{Ent}(f_{\mathcal{D}}) = \ln(\sigma\sqrt{2\pi e})$ and $\alpha = 1$. We know from the proof of Theorem H.2 that the output density $f^\star$ is the same as $f_{\mathcal{D}}$ and hence, has mean 0.

Now consider an instance $\mathcal{I}_2$, that corresponds to the disadvantaged group and has the same parameters as that of $\mathcal{I}_1$, except that the parameter $\alpha$ is larger than 1. Let $f_\alpha^\star$ denote the corresponding output density. We know from Theorem C.16 that the output density is proportional to $e^{-I_\alpha(x)/\gamma_\alpha^\star}$, where $\gamma_\alpha^\star$ is the optimal dual variable for the entropy constraint and Lemma H.1 shows that $I_\alpha(x)$ is given by:

$$I_\alpha(x) = (\alpha - 1)\sigma^2 \left((w^2 + 1)\Phi(w) + w\phi(w)\right) + \sigma^2\left(w^2 + 1\right),$$

where $w = (x - \mu)/\sigma$. For sake of brevity, let $g(w)$ denote $(w^2 + 1)\Phi(w) + w\phi(w)$, and assume w.l.o.g. that $\mu = 0$ and $\sigma = 1$, i.e., the input density is standard normal. Thus, $I_\alpha(x)$ can be written as $(\alpha - 1)g(x) + (x^2 + 1)$. When $\alpha$ is large, the first term here dominates the second term as long as $\Phi(x) \gg 1/\alpha$. Since $\Phi(x) \geq 1/2$ for all $x \geq 0$, $I_\alpha(x)$ is relatively large for all $x \geq 0$, and hence, $f_\alpha^\star(x)$ goes to 0 for $x \in [0, \infty)$ as $\alpha$ increases. In fact, $g(x)$ is much larger than $1/\alpha$ when $x$ is larger than $-\Omega(\sqrt{\ln(\alpha)})$ and, hence, $f_\alpha^\star$ should be small for such values of $x$ as well. Therefore, we expect the mean of the output density $f_\alpha^\star$ to go to $-\infty$ as $\alpha$ goes to $\infty$. Further, the mean of $f_\alpha^\star$ should decrease at a logarithmic rate with respect to $\alpha$. We verify these observations numerically in Figure 5.

**Deriving the implicit variance model.** We show that the implicit variance model can be derived from our optimization framework. In particular, we prove the following result:

**Theorem H.3.** *Consider an instance of the implicit variance model given by parameters $\mu, \sigma, \sigma_0$. Consider instances $I_1 = (\Omega, f_{\mathcal{G}}, \ell, \alpha, \tau_1)$ and $I_2 = (\Omega, f_{\mathcal{G}}, \ell, \alpha, \tau_2)$ of (OptProg), where $\Omega = \mathbb{R}$, $f_{\mathcal{G}}$ is the normal density $N(\mu, \sigma_0^2)$, $\alpha = 1$, $\ell(v, x) = (x - v)^2$, $\tau_1 = \frac{1}{2}(1 + \ln(2\pi\sigma_0^2))$ and*

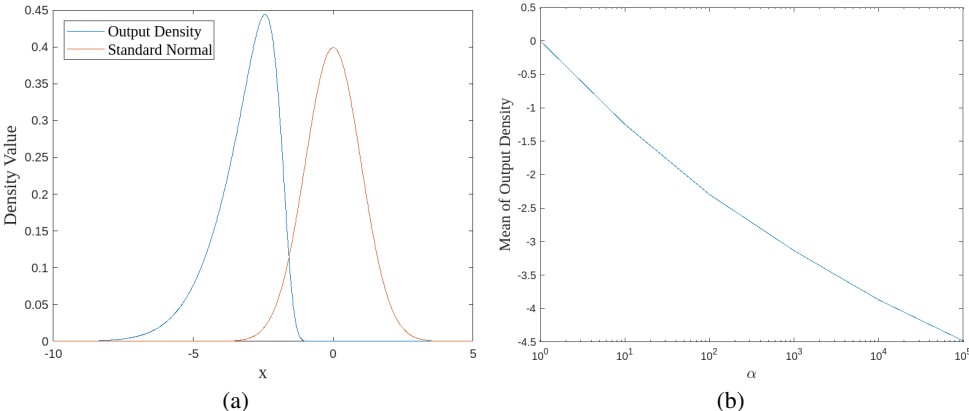

Figure 5: Figure (a) plots the input density, i.e., standard normal density, and the output density when the parameter $\alpha = 1000$. Figure (b) plots the mean of the output density as a function of the parameter $\alpha$ (on a log scale).

$\tau_2 = \frac{1}{2}(1 + \ln(2\pi(\sigma_0^2 + \sigma^2)))$. *Then the output density of* (OptProg) *on* $I_1$ *is* $N(\mu, \sigma_0^2)$ *and the output density of* (OptProg) *on* $I_2$ *is* $N(\mu, \sigma_0^2 + \sigma^2)$.

*Proof.* The result follows from Theorem H.2. For the instance $\mathcal{I}_1$, Theorem H.2 shows that the output density of OptProg is the normal density with mean $\mu$ and variance

$$\frac{1}{2\pi}e^{2\tau_1 - 1} = \sigma_0^2.$$

Similarly, it follows that the output density for the instance $\mathcal{I}_2$ is normal with mean $\mu$ and variance

$$\frac{1}{2\pi}e^{2\tau_2 - 1} = \sigma_0^2 + \sigma^2.$$

$\square$

## I  Pareto density

The Pareto density is defined as follows over $\Omega = [1, \infty)$ and has a parameter $\beta > 1$:

$$f_{\mathcal{P}}(x) := \frac{\beta}{x^{\beta+1}}, \quad x \in [1, \infty).$$

The mean of this density is $\frac{\beta}{\beta-1}$. Thus, the mean is finite only when $\beta > 1$. Its differential entropy is $1 + \frac{1}{\beta} + \ln\frac{1}{\beta}$ [148]. We consider the loss function $\ell(x, v) := \ln x - \ln v$. First, we compute the expression for $I(x)$ which we use to verify the applicability of Theorem C.1 with the above parameters.

**Lemma I.1 (Expression for $I_\alpha(x)$).** *Consider an instance* $\mathcal{I} = (\Omega, f_{\mathcal{P}}, \ell, \alpha, \tau)$ *of* PrimalOpt *where* $\Omega = [1, \infty)$, $\ell(x, v) := \ln x - \ln v$, *and* $f_{\mathcal{P}}$ *is the Pareto density with parameter* $\beta > 1$. *Then*

$$I(x) = \alpha \ln x + \frac{\alpha - 1}{\beta x^\beta} - \frac{\alpha}{\beta}.$$

*Proof.* We use integration by parts to derive the following expression for $I(x)$:

$$I(x) = \alpha \int_1^x (\ln x - \ln v)\frac{\beta}{v^{\beta+1}} d\mu(v) + \int_x^\infty (\ln x - \ln v)\frac{\beta}{v^{\beta+1}} d\mu(v)$$

$$= \alpha \left[\frac{\ln v - \ln x}{v^\beta}\right]_1^x - \alpha \int_1^x \frac{1}{v^{\beta+1}} d\mu(v) + \left[\frac{\ln v - \ln x}{v^\beta}\right]_x^\infty + \int_x^\infty \frac{1}{v^{\beta+1}} d\mu(v)$$

$$= \alpha \ln x + \frac{\alpha}{\beta}\left(\frac{1}{x^\beta} - 1\right) - \frac{1}{\beta x^\beta}.$$

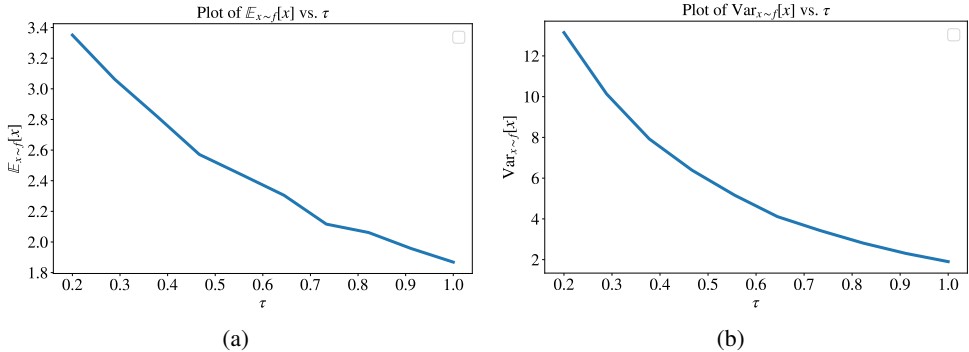

Figure 6: Mean and variance of the output density $f$, i.e., $\mathbb{E}_{x \sim f}[x]$ and $\text{Var}_{x \sim f}[x]$, as a function of $\tau$ when $f_{\mathcal{D}}$ is the Pareto distribution with parameter 3 and $\alpha$ is fixed to 2.

□

**Applicability of Theorem C.1.** Next, we verify that assumptions **(A0)–(A5)** hold. Since $\Omega = [1, \infty)$, **(A0)** holds for any finite $\tau$. **(A1)** and **(A2)** hold due to the choice of the loss function. **(A3)** can be shown to hold since $f_{\mathcal{P}}$ is a Pareto density: To see this note that for any finite $x \geq 1$

$$\int_{\Omega} |\ell(x, v)| \, f_{\mathcal{P}}(v) d\mu(v) = \int_1^x (\ln(x) + \ln(v)) \, f_{\mathcal{P}}(v) d\mu(v) + \int_x^{\infty} (\ln(v) - \ln(x)) \, f_{\mathcal{P}}(v) d\mu(v)$$

$$\leq \int_1^{\infty} (\ln(x) + \ln(v)) \, f_{\mathcal{P}}(v) d\mu(v)$$

$$= \ln(x) + \int_1^{\infty} \ln(v) f_{\mathcal{P}}(v) d\mu(v)$$

$$= \ln(x) + \left[ -\frac{\ln(v)}{v^{\beta}} \right]_1^{\infty} + \int_1^{\infty} \frac{1}{v^{\beta+1}} d\mu(v)$$

$$= \ln(x) + \left[ -\frac{v^{-\beta}}{\beta} \right]_1^{\infty}$$

$$= \ln(x) + \frac{1}{\beta}$$

$$< \infty.$$

Thus, **(A3)** holds. **(A4)** holds with, e.g., $R = 2$. By Lemma I.1,

$$I(x) = \alpha \ln x + \frac{\alpha - 1}{\beta x^{\beta}} - \frac{\alpha}{\beta}.$$

Thus, $I(x)$ is differentiable at each $x \in \Omega$. Moreover, for all $x \in \Omega$

$$\frac{\partial I(x)}{\partial x} = \frac{\alpha}{x} - \frac{\alpha - 1}{x^{\beta+1}} = \frac{1}{x} \left( \alpha - \frac{\alpha - 1}{x^{\beta}} \right) \overset{x \geq 1}{>} 0.$$

Since the derivative is positive for all $x \in \Omega = [1, \infty)$, it follows that $I(x)$ has a unique global minimum at $x = 1$. Therefore, **(A5)** holds. Since assumptions **(A0)–(A5)** hold, we invoke Theorem C.1 to deduce the form of $f^{\star}$. Figure 6 plots the mean and the variance of the output density $f^{\star}$ as a function of the parameter $\tau$ when the input density $f_{\mathcal{D}}$ is the standard normal density. Figure 7 plots the mean and the variance of the output density as a function of the parameter $\alpha$ in this setting.

**Theorem I.2 (Expression for $f^{\star}$ with $\alpha = 1$).** *Consider an instance $\mathcal{I} = (\Omega, f_{\mathcal{P}}, \ell, 1, \tau)$ of PrimalOpt where $\Omega = [1, \infty)$, $\ell := \ln x - \ln v$, and $f_{\mathcal{P}}$ is the Pareto density with parameter $\beta > 1$. Let $f^{\star}$ be the optimal solution of instance $\mathcal{I}$. $f^{\star}$ is a Pareto density with parameter $\beta_{\tau}$ satisfying the following condition:*

$$1 + \frac{1}{\beta_{\tau}} - \ln \beta_{\tau} = \tau.$$

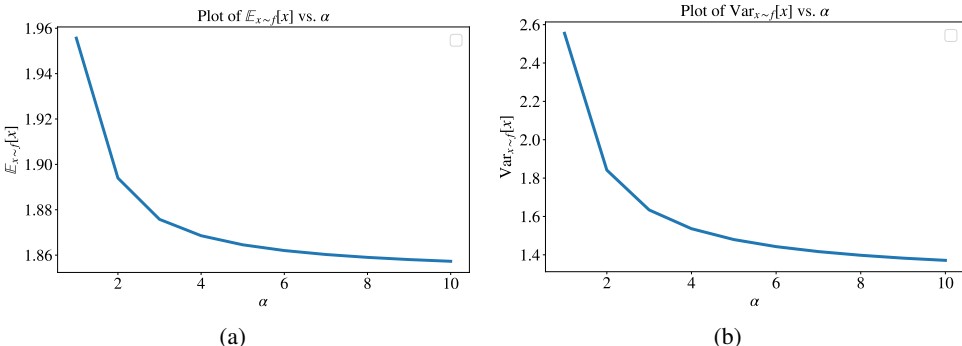

(a)                            (b)

Figure 7: Mean and variance of the output density $f$, i.e., $\mathbb{E}_{x\sim f}[x]$ and $\mathrm{Var}_{x\sim f}[x]$, as a function of $\alpha$ when $f_\mathcal{D}$ is the Pareto distribution with parameter 3 and $\tau$ is fixed to the entropy of $f_\mathcal{D}$.

*Let $m_\tau$ and $\sigma_\tau^2$ be the mean and variance of $f_\tau^\star$ as a function of $\tau$. It holds that $m_\tau$ is monotonically increasing in $\tau$ and $\sigma_\tau^2$ is either infinite or monotonically increasing in $\tau$.*

*Proof.* For $\alpha = 1$, Lemma I.1 implies that

$$I(x) = \ln(x) - \frac{1}{\beta}.$$

As shown earlier, one can invoke Theorem C.1 for instance $\mathcal{I}$ for any finite $\tau$, which implies that $f^\star$ has the following form

$$f^\star(x) \propto e^{\frac{-\ln(x)+\frac{1}{\beta}}{\gamma^\star}} \propto x^{-\frac{1}{\gamma^\star}}.$$

where the proportionality constant and $\gamma^\star$ are determined by $\int_\Omega f^\star(x)d\mu(x) = 1$ and $\mathsf{Ent}(f^\star) = \tau$. $f^\star$ is a Pareto density with parameter $\beta_\tau = \frac{1}{\gamma^\star} - 1$ and, hence, has a differential entropy of $1 + \frac{1}{\beta_\tau} + \ln\frac{1}{\beta_\tau}$ [148]. This combined with the condition $\mathsf{Ent}(f^\star) = \tau$ implies that $f^\star$ is the Pareto density with $\beta_\tau$ satisfying

$$1 + \frac{1}{\beta_\tau} + \ln\frac{1}{\beta_\tau} = \tau.$$

Since $1 + \frac{1}{\beta_\tau} + \ln\frac{1}{\beta_\tau}$ is a decreasing function of $\beta_\tau$, it follows that increasing $\tau$ monotonically decreases $\beta_\tau$. Note that $\beta_\tau > 1$ for any finite $\tau$. The mean and variance of $f^\star$ are $m_\tau = \frac{\beta_\tau}{\beta_\tau - 1}$ and $\sigma_\tau^2 = \frac{\beta_\tau}{(\beta_\tau - 1)^2(\beta_\tau - 2)}$ respectively. Since $m_\tau$ is a monotonically decreasing function of $\beta_\tau$ (for $\beta_\tau > 1$) and $\beta_\tau$ is a monotonically decreasing function of $\tau$, it follows that $m_\tau$ is a monotonically increasing function of $\beta_\tau$. The variance $\sigma_\tau^2$ is finite when $\beta_\tau > 2$. Moreover, if $\beta_\tau > 2$, then $\sigma_\tau^2$ is a monotonically decreasing function of $\beta_\tau$. Since $\beta_\tau$ is a monotonically decreasing function of $\tau$, it follows that $\sigma_\tau^2$ is either infinite or a monotonically increasing function of $\beta_\tau$. $\qquad\square$

**Reduction in mean with increase in $\alpha$.** We show how our framework can capture a similar phenomenon as the multiplicative-bias model of [90]. Recall that in the multiplicative-bias model, the estimated utility of the disadvantaged group is scaled down by a factor $\rho > 1$.

Fix a parameter $\beta > 1$. Consider an instance $\mathcal{I}_1$ given by the parameters $I_1 = (\Omega = [1, \infty), f_\mathcal{D}, \ell, \alpha, \tau)$ where $f_\mathcal{D}$ is the Pareto density with parameter $\beta$, $\tau = \mathsf{Ent}(f_\mathcal{D}) = 1 + \frac{1}{\beta} + \ln\frac{1}{\beta}$, $\alpha = 1$ and $\ell(x, v) = \ln x - \ln v$. As shown in Theorem I.2, the output density $f^\star$ is the same as $f_\mathcal{D}$. The proof of this result also shows that the output density $f^\star(x)$ is proportional to $e^{-I(x)/\gamma^\star}$, where $I(x) = \ln x - \frac{1}{\beta}$ and $\gamma^\star$ is the optimal dual variable for the corresponding entropy constraint.

The disadvantaged group is modeled by an instance $\mathcal{I}_2$ which has the same parameters as that of $\mathcal{I}_1$ except that the parameter $\alpha$ is larger than 1. In this case, Theorem I.2 shows that the optimal density $f_\alpha^\star(x)$ is proportional to $e^{-I_\alpha(x)/\gamma_\alpha^\star}$, where $I_\alpha(x) = \alpha\ln x + \frac{\alpha-1}{\beta x^\beta} - \frac{\alpha}{\beta}$ and $\gamma_\alpha^\star$ is the corresponding

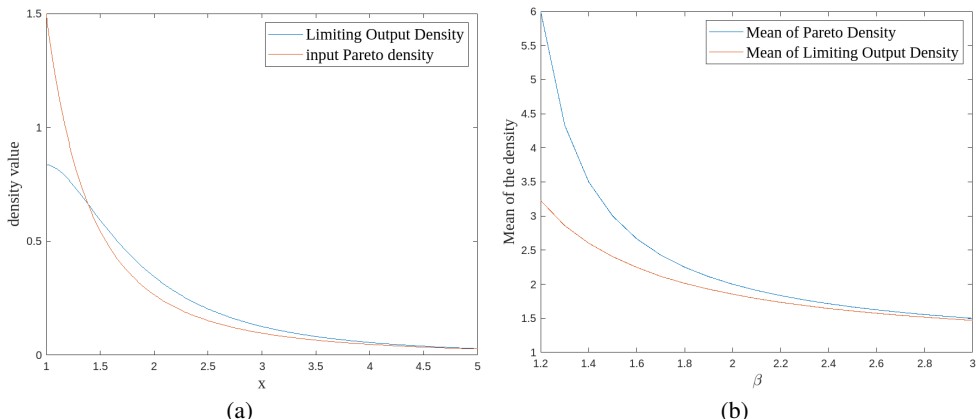

Figure 8: Figure (a) plots the input Pareto density with parameter $\beta = 1.5$ and the corresponding limiting output density. Figure (b) plots the mean of the input Pareto density and the corresponding limiting output density as a function of the parameter $\beta$.

optimal dual variable. For large $\alpha$, $I_\alpha(x) \sim \alpha(I(x) + \frac{1}{\beta x^\beta})$. Hence, the output density (for large $\alpha$) is given by

$$f_\alpha^\star(x) = K \cdot e^{-\frac{\alpha}{\gamma_\alpha^\star}\left(I(x) + \frac{1}{\beta x^\beta}\right)},$$

where $K$ is a normalization constant. The normalization constant $K$ and the ratio $\gamma_\alpha^\star/\alpha$ are given by two constraints: (i) integral of $f_\alpha^\star$ over the domain $\Omega$ should be 1, and (ii) the entropy of $f_\alpha^\star$ should be equal to $\tau = \mathsf{Ent}(f_\mathcal{D})$. This shows that the ratio $\gamma_\alpha^\star/\alpha$ tends to a constant for large $\alpha$, and hence, the output density converges to the density given by

$$g^\star(x) = K e^{-C\left(\ln x + \frac{1}{\beta x^\beta}\right)}.$$

We plot the output density $g^\star(x)$ for $\beta = 1.5$ in Figure 8(a). In Figure 8(b), we observe that for all considered values of the parameter $\beta$, the mean of the limiting output density $g^\star$ always remains below that of the input density $f_\mathcal{D}$. It is also worth noting that for large $\beta$, the gap between the mean of the input density and that of the (limiting) output density diminishes. Intuitively, this happens because as $\beta$ increases, the mean of the input (Pareto) density gets closer to 1 (recall that the mean of a Pareto density with parameter $\beta$ is equal to $\frac{\beta}{\beta-1}$). Now the output density also places more mass closer to 1, but gets restricted because of two conditions: (i) the entropy of the output density must be the same as that of the corresponding input density, and (ii) it cannot place any probability mass on values below 1. Hence, there is not much "room" for the output density to place extra probability mass on values close to 1 (as compared to the corresponding Pareto density). Hence its mean cannot go much below that of the corresponding Pareto density.

## J   Exponential density

The exponential density is defined as follows over $\Omega = [0, \infty)$ and has a parameter $\lambda > 0$:

$$f_{\mathrm{Exp}}(x) := \lambda e^{-\lambda x}, \quad x \in [0, \infty).$$

$\lambda$ is referred to as the "rate" parameter. The mean of the exponential density is $\frac{1}{\lambda}$. The differential entropy of $f_{\mathrm{Exp}}$ is $1 - \ln \lambda$ [148]. We consider the loss function $\ell(x, v) := x - v$. First, we compute the expression of $I(x)$ which we use to verify the applicability of Theorem C.1 with the above parameters.

**Lemma J.1 (Expression for $I(x)$).** *Consider an instance $\mathcal{I} = (\Omega, f_{\mathrm{Exp}}, \ell, \alpha, \tau)$ of PrimalOpt where $\Omega = [0, \infty)$, $f_{\mathrm{Exp}}$ is the Exponential density with rate parameter $\lambda$, and $\ell(x, v) := x - v$. Then*

$$I(x) = \frac{1}{\lambda}\left(\alpha(\lambda x - 1) + (\alpha - 1)e^{-\lambda x}\right).$$

*Proof.* The desired integral is

$$\lambda\alpha \int_0^x (x-v)e^{-\lambda v}d\mu(v) + \lambda \int_x^\infty (x-v)e^{-\lambda v}d\mu(v).$$

We do a change of variable, with $y := x - v$ above, to get

$$\lambda\alpha \int_0^x ye^{\lambda(y-x)}d\mu(y) + \lambda \int_{-\infty}^0 ye^{\lambda(y-x)}d\mu(y) = \lambda\alpha e^{-\lambda x}\left[e^{\lambda y}\frac{\lambda y-1}{\lambda^2}\right]_0^x + \lambda e^{-\lambda x}\left[e^{\lambda y}\frac{\lambda y-1}{\lambda^2}\right]_{-\infty}^0$$

$$= \frac{\alpha(\lambda x-1)}{\lambda} + \frac{\alpha e^{-\lambda x}}{\lambda} - \frac{e^{-\lambda x}}{\lambda}$$

$$= \frac{1}{\lambda}\left(\alpha(\lambda x-1) + (\alpha-1)e^{-\lambda x}\right).$$

$\square$

**Applicability of Theorem C.1.** We show that assumptions **(A0)–(A5)** hold. Since $\Omega = [0,\infty)$, **(A0)** holds for any finite $\tau$. **(A1)** and **(A2)** hold due to the choice of the loss function. **(A3)** can be shown to hold since $f_{\mathrm{Exp}}$ is an Exponential density: To see this note that for any finite $x \in \Omega$

$$\int_\Omega |\ell(x,v)|\, f_{\mathrm{Exp}}(v)d\mu(v) = \int_1^x (x-v)\, f_{\mathrm{Exp}}(v)d\mu(v) + \int_x^\infty (v-x)\, f_{\mathrm{Exp}}(v)d\mu(v)$$

$$\leq \int_1^\infty (|x|+|v|)\, f_{\mathrm{Exp}}(v)d\mu(v)$$

$$= |x| + \frac{1}{\lambda}$$

$$< \infty.$$

Thus, **(A3)** holds. **(A4)** holds with, e.g., $R = \frac{1}{\lambda}$. By Lemma J.1,

$$I(x) = \frac{1}{\lambda}\left(\alpha(\lambda x-1) + (\alpha-1)e^{-\lambda x}\right).$$

From this expression, it follows that $I(x)$ is differentiable at each $x \in \Omega$. Moreover, for all $x \in \Omega$

$$\frac{\partial^2 I(x)}{\partial^2 x} = \lambda(\alpha-1)e^{-\lambda x}.$$

For any $\alpha > 1$, this derivative is positive for all $x \in \Omega$ and, hence, $I(x)$ strictly convex whenever $\alpha > 1$ and, thus, it has a unique global minimum. If $\alpha = 1$, then $I(x) = x - \frac{1}{\lambda}$. This function has a unique global minimum at $x = 0$ over $\Omega = [0,\infty)$. Combining with the $\alpha > 1$ case, it follows that **(A5)** holds. Since assumptions **(A0)–(A5)** hold, one can invoke Theorem C.1 to deduce the form of $f^\star$.

**Theorem J.2 (Expression for $f^\star$ when $\alpha = 1$).** *Consider an instance $\mathcal{I} = (\Omega, f_{\mathrm{Exp}}, \ell, 1, \tau)$ of PrimalOpt where $\Omega = [0,\infty)$, $\ell(x,v) := x - v$, and $f_{\mathrm{Exp}}$ is the Exponential with rate parameter $\lambda$. Let $f^\star$ be the optimal solution of $\mathcal{I}$. Then $f^\star$ is the Exponential density with mean $e^{\tau-1}$*

Thus, for $\alpha = 1$, increasing $\tau$ increases the rate parameter of the output density.

*Proof.* Since $\alpha = 1$, Lemma J.1 implies that

$$I(x) = x - \frac{1}{\lambda}.$$

As shown earlier, one can invoke Theorem C.1 for instance $\mathcal{I}$ for any finite $\tau$, which implies that $f^\star$ has the following form

$$f^\star(x) \propto \exp\left(\frac{-x - \frac{1}{\lambda}}{\gamma^\star}\right) \propto \exp\left(\frac{-x}{\gamma^\star}\right).$$

where the proportionality constant and $\gamma^\star$ are determined by $\int_\Omega f^\star(x)d\mu(x) = 1$ and $\mathrm{Ent}(f^\star) = \tau$. Since $f^\star$ is an Exponential density with rate parameter $\frac{1}{\gamma^\star}$, its entropy is $\mathrm{Ent}(f^\star) = 1 + \ln(\gamma^\star)$ [148]. Since $\mathrm{Ent}(f^\star) = \tau$, the previous equality implies that $\gamma^\star = e^{\tau-1}$. It follows that $f^\star(x) = e^{1-\tau} \cdot \exp\left(-\frac{x}{\exp(\tau-1)}\right)$ which is the Exponential density with mean $e^{\tau-1}$. $\square$

# K  Laplace density

The Laplace density is defined as follows over $\Omega = \mathbb{R}$ and has parameters $a \in \mathbb{R}$ and $b > 0$:

$$f_{\mathcal{L}}(x) := \frac{1}{2b} e^{-\frac{1}{b}|x-a|}, \quad x \in \mathbb{R}.$$

$a$ is referred to as the "location" parameter and $b$ as "diversity." The differential entropy of $f_{\mathcal{L}}$ is $1 + \ln(2b)$ [148]. We consider the loss function to be $\ell(x,v) := |x - a| - |v - a|$ for $a \in \mathbb{R}$. First, we compute the expression for $I(x)$ that we use to show that Theorem C.1 is applicable with the above parameters.

**Lemma K.1** (**Expression for** $I(x)$)**.** *Consider an instance $\mathcal{I} = (\Omega, f_{\mathcal{L}}, \ell, \alpha, \tau)$ of PrimalOpt where $\Omega = \mathbb{R}$, $f_{\mathcal{L}}$ is the Laplace density with parameters $a \in \mathbb{R}$ and $b > 0$, and $\ell(x,v) := |x - a| - |v - a|$. Then*

$$I(x) = \begin{cases} \alpha b(w - 1) + b\frac{(\alpha-1)}{2}e^{-w} & \text{if } x \geq a \\ -b(w + 1) + b\frac{(1-\alpha)}{2}e^{w} & \text{otherwise} \end{cases},$$

*where $w := \frac{x-a}{b}$.*

*Proof.* The desired integral is

$$\frac{\alpha}{2b} \int_{-\infty}^{x} (|x - a| - |v - a|)e^{-|v-a|/b}d\mu(v) + \frac{1}{2b} \int_{x}^{\infty} (|x - a| - |v - a|)e^{-|v-a|/b}d\mu(v).$$

We perform a change of variables with $w := \frac{x-a}{b}$ and $y = \frac{v-a}{b}$. Then the above integral becomes

$$\frac{\alpha b}{2} \int_{-\infty}^{w} (|w| - |y|)e^{-|y|}d\mu(y) + \frac{b}{2} \int_{w}^{\infty} (|w| - |y|)e^{-|y|}d\mu(y).$$

Now two cases arise (i) $w \geq 0$, or (ii) $w \leq 0$. First, consider the case when $w \geq 0$. Then the above integral becomes:

$$\frac{\alpha}{2} \int_{-\infty}^{0} (w + y)e^{y}d\mu(y) + \frac{\alpha}{2} \int_{0}^{w} (w - y)e^{-y}d\mu(y) + \frac{1}{2} \int_{w}^{\infty} (w - y)e^{-y}d\mu(y)$$
$$= \frac{\alpha(w - 1)}{2} + \frac{\alpha(w + e^{-w} - 1)}{2} - \frac{e^{-w}}{2}$$
$$= \alpha b(w - 1) + \frac{(\alpha - 1)b}{2}e^{-w}.$$

In the second case, we get

$$\frac{\alpha}{2} \int_{-\infty}^{w} (-w + y)e^{y}d\mu(y) + \frac{1}{2} \int_{w}^{0} (-w + y)e^{y}d\mu(y) + \frac{1}{2} \int_{0}^{\infty} (-w - y)e^{-y}d\mu(y)$$
$$= -\frac{\alpha e^{w}}{2} + \frac{e^{w} - w - 1}{2} - \frac{1 + w}{2}$$
$$= -(w + 1)b + \frac{(1 - \alpha)e^{w}b}{2}.$$

$\square$

**Applicability of Theorem C.1.** We show that for any finite $a \in \mathbb{R}$ and $b > 0$, assumptions **(A0)**–**(A5)** hold for the instance $\mathcal{I} = (\Omega, f_{\mathcal{L}}, \ell, \alpha, \tau)$ of PrimalOpt where $\Omega = \mathbb{R}$, $f_{\mathcal{L}}$ is the Laplace density with parameters $a \in \mathbb{R}$ and $b > 0$, and $\ell(x,v) = |x - a| - |v - a|$. Since $\Omega = \mathbb{R}$, **(A0)** holds for any

finite $\tau$. **(A1)** and **(A2)** hold due to the choice of the loss function. **(A3)** can be shown to hold since $f_{\mathcal{L}}$ is a Laplace density: To see this note that for any finite $x \in \Omega$

$$\int_{\Omega} |\ell(x,v)| \, f_{\mathcal{L}}(v) d\mu(v) \leq \int_{0}^{\infty} (|x - a| + |v - a|) \, f_{\mathcal{L}}(v) d\mu(v)$$
$$= |x - a| + \frac{b}{2}$$
$$< \infty.$$

Thus, **(A3)** holds. **(A4)** holds with, e.g., $R = b \ln(2)$. By Lemma K.1,

$$I(x) = \begin{cases} \alpha b(w - 1) + b \frac{(\alpha-1)}{2} e^{-w} & \text{if } x \geq a \\ -b(w + 1) + b \frac{(1-\alpha)}{2} e^{w} & \text{otherwise} \end{cases},$$

where $w = \frac{x-a}{b}$. One can check that $I(x)$ is continuous and differentiable at each $x \in \Omega \setminus \{a\}$. Moreover, for all $x < a$, $\frac{\partial I(x)}{\partial x} < 0$ and for all $x \geq a$, $\frac{\partial I(x)}{\partial x} \geq 0$. Hence, it follows that $I(x)$ has a unique global minimum at $x = a$. Therefore, **(A5)** holds. Since assumptions **(A0)–(A5)** hold, we invoke Theorem C.1 to deduce the form of the optimal density.

**Theorem K.2 (Expression for $f^{\star}$ when $\alpha = 1$).** *Consider an instance $\mathcal{I} = (\Omega, f_{\mathcal{L}}, \ell, 1, \tau)$ of PrimalOpt where $\Omega = \mathbb{R}$, $f_{\mathcal{D}}$ is the Laplace density with parameters $a \in \mathbb{R}$ and $b > 0$, and $\ell(x,v) = |x - a| - |v - a|$. Let $f^{\star}$ be the optimal solution of $\mathcal{I}$. Then $f^{\star}$ is the Laplace density with parameters $(a, e^{\tau-1}/2)$.*

Thus, for $\alpha = 1$, increasing $\tau$ does not change the location parameter, but increases the "diversity" parameter of the output density.

*Proof.* Since $\alpha = 1$, Lemma K.1 implies that

$$I(x) = |x - a| - \frac{b}{2}.$$

As shown earlier, one can invoke Theorem C.1 for instance $\mathcal{I}$ for any finite $\tau$, which implies that $f^{\star}$ has the following form

$$f^{\star}(x) \propto \exp\left(\frac{-|x - a| - \frac{b}{2}}{\gamma^{\star}}\right) \propto \exp\left(\frac{-|x - a|}{\gamma^{\star}}\right).$$

where the proportionality constant and $\gamma^{\star}$ are determined by $\int_{\mathbb{R}} f^{\star}(x) d\mu(x) = 1$ and $\mathsf{Ent}(f^{\star}) = \tau$. Clearly, $f^{\star}$ is a Laplace density with the diversity parameter $\gamma^{\star}$, its entropy is $\mathsf{Ent}(f^{\star}) = 1 + \ln(2\gamma^{\star})$ [148]. On the other hand, since $\mathsf{Ent}(f^{\star}) = \tau$, the previous equality implies that $\gamma^{\star} = \frac{1}{2} e^{\tau-1}$. It follows that $f^{\star}(x) = e^{1-\tau} \cdot \exp\left(-\frac{2|x-a|}{\exp(\tau-1)}\right)$ which is the Laplace density with parameters $(a, \frac{1}{2} e^{\tau-1})$. $\square$

## L   Implementation details and additional empirical results

In this section, we present additional discussions and evaluations of intervention in the JEE setting (Appendix L.1), plots omitted from Section 4 (Appendix L.2), and implementation details of our model (Appendix L.3). The code for this paper is available at `https://github.com/AnayMehro tra/Bias-in-Evaluation-Processes`.

### L.1   Case Study: Evaluating bias-mitigating interventions in IIT-JEE admissions

In this section, we continue our study of the effectiveness of different interventions in a downstream selection task. Like in Section 4, we consider selection based on the JEE 2009 scores, but here consider representational constraints actually used in admissions to IITs. We also discuss additional interventions being implemented by the Indian state and central governments to reduce inequity in JEE scores.

Recall that, the Indian Institutes of Technology (IITs) are a group of engineering institutes in India. In 2009, there were 15 IITs and today this has grown to 23. Undergraduate admissions at IITs are decided based on the scores of candidates in the Joint Entrance Exam (JEE). JEE is conducted once every year. In 2009, the scores, (binary) genders, and birth categories of all candidates who appeared in JEE 2009 were released in response to a Right to Information application filed in June 2009 [91]. The birth category of the candidates is an official socioeconomic status label recognized by the government of India [135].

Here, we focus on two groups of candidates: the candidates in the general (GEN) category (the most privileged) and candidates not in the general category. We begin by discussing some of the interventions in place to reduce inequity in JEE scores and subsequent admissions at IITs.

**Interventions used in IIT admissions.** The Indian constitution allows the central government and state governments to enforce affirmative action in the form of quotas or lower bound constraints for official SES groups at educational institutes, employments, and political bodies [82, 146]. In 2005 lower-bound interventions were introduced in the admissions process at the IITs. Concretely, in 2009, out of the 7,440 seats, 3,688 (49.6%) were reserved for students who are not in the GEN category. This means that at least 3,688 out of the $7,440$ students admitted into IITs must not be in the GEN category. Note that this allows more than 3,688 or even all admitted students to be outside the GEN category. We call this constraint the `Reservation` constraint and, in this section, we study its effectiveness compared to other forms of interventions.

Apart from reservations, a number of other interventions have also been proposed and/or implemented to reduce biases in the JEE. We discuss two other types of interventions next.

**Interventions to reduce skew.** Private coaching institutes that train students for JEE have been criticized for being exorbitantly expensive and, hence, inaccessible for students in low SES groups [44]. Lack of accessibility to training resources can reduce the scores of candidates in low SES groups–creating a skew in the scores. To improve accessibility to training, in 2022, the Delhi government established a new program that will provide free training to students enrolled in government-funded schools [53]. Similar programs have also been introduced in other states [52, 141] and by school education boards that span multiple states [26]. In the context of our model, these interventions can be seen as reducing this skew in the evaluation process.

**Interventions to reduce information constraint.** A criticism of JEE is that it is only offered in the English and Hindi languages. This is undesirable because only 44% of Indians report English or Hindi as their first language and, according to the 2011 census, less than 68% of Indians list one of these languages among the three languages they are most comfortable with [103]. IITs have been repeatedly criticized for not offering the exam in regional languages [128, 7, 47]. The main concern is that the current exam reduces the performance of students less familiar with English and Hindi. In the context of our model, this can be thought of as placing a stronger information constraint on candidates who do not speak English or Hindi as a first language: these students would need to spend a higher cognitive load to understand the questions. This constraint not only acts during the exam but also during the preparation period because students speaking regional languages (and not English or Hindi), have to devote additional time to learning either English or Hindi in addition to the technical material for the exam.

While the JEE exam itself has not been offered in regional languages yet. Recently, in 2021, the screening test that candidates have to clear before appearing in JEE was offered in 11 regional languages in addition to English and Hindi [134].

In this section, we compare the effectiveness of the above three interventions – `Reservation` for lower SES groups, interventions to reduce skew (change $\alpha$ by $\Delta_\alpha$ percent), and interventions to reduce information-constraint (change $\tau$ by $\Delta_\tau$ percent).

**Setup (Group sizes and $k$).** Admissions into IITs are highly selective. For instance, in 2009, 384,977 applicants (232,334 from GEN; 60%) took the exam and just 7,440 (2%) were admitted to IITs. The admission is based on the candidates' All India Rank (henceforth just rank)–which denotes the candidate's position in the list of candidates ordered in decreasing order of their scores in JEE. Let $G_1$ be the group of students in GEN category and $G_2$ be all other students. To study the impact of different interventions for admissions into IITs, we fix group sizes and $k$ to match real numbers: $|G_1| = 232,334$, $|G_2| = 152,643$, and $k = 7,400$. We focus on the set of candidates who scored at least 80 (out of 480) on the exam. (The threshold 80 ensures that at least 10k candidates outside

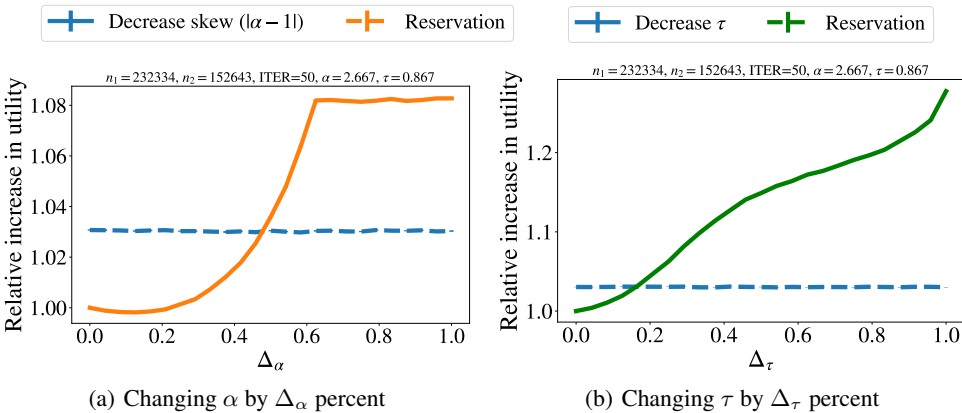

(a) Changing $\alpha$ by $\Delta_\alpha$ percent  (b) Changing $\tau$ by $\Delta_\tau$ percent

Figure 9: *Effectiveness of different interventions on the selection-utility–as estimated by our model:* We vary the strengths of the interventions ($\Delta_\alpha \in [0,1]$ and $\Delta_\tau \in [0,1]$) and report the expected utilities of the subset output by all three interventions. The $x$-axis shows the strength of the intervention changing $\alpha$ (Figure 9(a)) or $\tau$ (Figure 9(b)). The $y$-axis shows the ratio of the (true) utility of the subset output with an intervention to the (true) utility of the subset output without any intervention. Our main observation is that for each of the three interventions, there is a value of the percentage change in $\alpha$ and $\tau$ (i.e., $\Delta_\alpha$ and $\Delta_\tau$ respectively) for which the intervention outperforms the other two interventions. Hence, depending on the amount of change a policymaker expects a specific intervention (e.g., providing free coaching) to have on the parameters $\alpha$ and $\tau$, they can use our framework as a tool to inform their decision about which intervention to enforce. Error bars represent the standard error of the mean over 100 repetitions.

the GEN category are considered and this is significantly lower than the $k$-th highest score of 167). We fix $f_\mathcal{D}$ to be the density of utilities of all candidates in $G_1$ who scored at least 80. Since $f_\mathcal{D}$ has a Pareto-like density (see Figure 10), we fix $\ell(x,v) = \ln(x) - \ln(v)$. We fix $\Omega$ to be the set of all possible scores and $f_{G_2}$ to be the density of all candidates in $G_2$ who scored at least 80. As in Section 4, we select $\alpha$ and $\tau$ that lead to the density closest in TV distance to $f_{G_2}$. The rest of the setup is the same as in Section 4.

Unlike the main body, here, we only consider high-scoring candidates (those with a score of at least 80) because JEE is highly selective ($k/n \leq 0.02$) and, hence, to have meaningful results the estimated density $f_\mathcal{E}$ should have a good fit to the density from the real-data on the top 2% quantile, i.e., the right tail. To ensure this, we specifically consider the right tail of the distribution (by dropping candidates with a score below 80).

**Observations and discussion.** We vary $\Delta_\alpha \in [0,1]$ and $\Delta_\tau \in [0,1]$ and report the expected utilities of the subset output by all three interventions over 100 iterations in Figure 9. Our main observation is that for each of the three interventions, there is a value of the percentage change in $\alpha$ and $\tau$ (i.e., $\Delta_\alpha$ and $\Delta_\tau$ respectively) for which the intervention outperforms the other two interventions. Hence, depending on the amount of change a policymaker expects a specific intervention (e.g., providing free coaching) to have on the parameters $\alpha$ and $\tau$, they can use our framework as a tool to inform their decision about which intervention to enforce. Further, we observe that, as expected, increasing $\Delta_\alpha$ and $\Delta_\tau$, i.e., the percentage of change in $\alpha$ and $\tau$, improves the utility achieved by the corresponding interventions.

**Limitations and further discussion.** Next, we discuss some of the limitations of our study. First, we note that interventions such as increasing the accessibility of education can not only reduce inequity

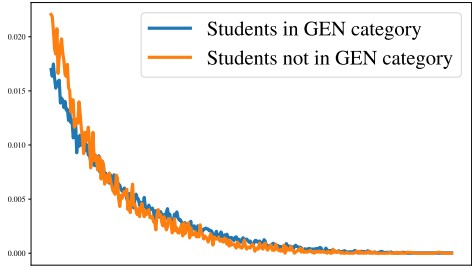

Figure 10: The densities of scores of students in GEN category (blue) and students not in GEN category (orange) in JEE-2009–only for students scoring at least 80 points out of 480.

in JEE but can also have positive effects on other exams and hiring. Hence, such interventions can have a larger positive (or negative) impact than suggested by our simulations. Studying these auxiliary effects is beyond the scope of this paper. Further, our study also does not model the response of the students, e.g., how do interventions affect the students' incentive to invest in skill development? Finally, our model only predicts the effect of $\alpha$ and $\tau$ on utility distributions. These predictions may not be accurate and a careful post-deployment evaluation may be required to accurately assess the effectiveness of different interventions.

## L.2    Additional plots for simulations in Section 4

In this section, we present plots of the best-fit densities output by our framework on different datasets.

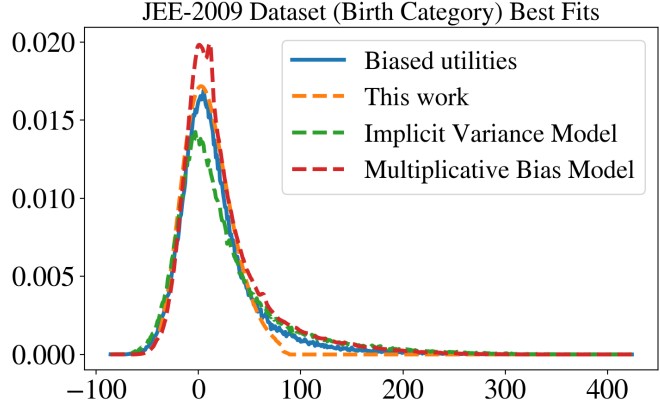

(a) Best-fit distribution ($\alpha = 3 \cdot 10^{-4}$ and $\tau = 1.51$) with JEE-2009 (Birth category)

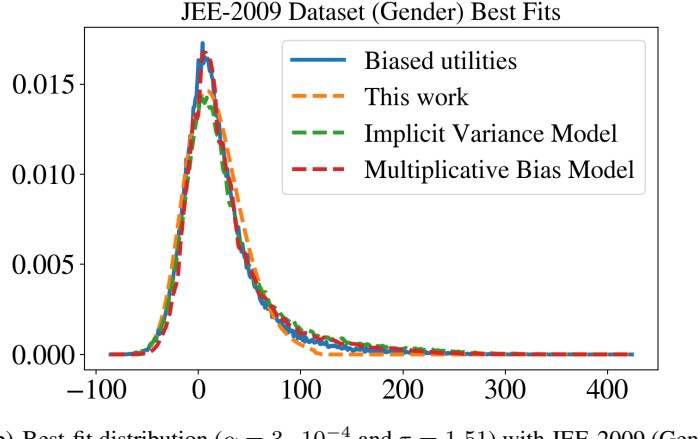

(b) Best-fit distribution ($\alpha = 3 \cdot 10^{-4}$ and $\tau = 1.51$) with JEE-2009 (Gender)

Figure 11:  Illustration of the best-fit distribution output by our framework for the JEE-2009 dataset. Captions of subfigures report the best fit $\alpha$ and $\tau$.

## L.3    Implementation details

### L.3.1    Our framework and other models

In this section, we give implementation details of our model. Recall that our model outputs the density which is the optimal solution of the following optimization program.

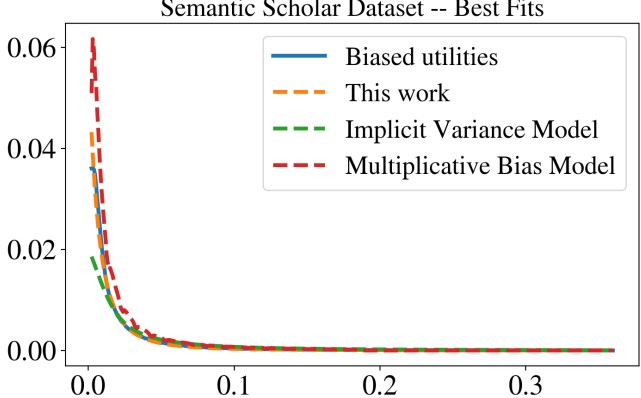

(a) Best-fit distribution ($\alpha = 22.78$ and $\tau = 2.09$) with Semantic Scholar Open Research Corpus

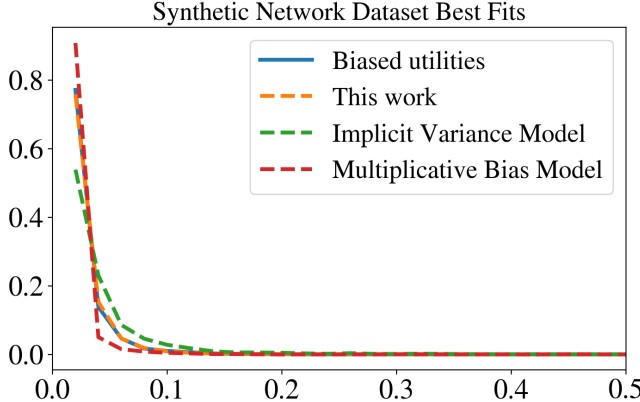

(b) Best-fit distribution ($\alpha = 1.92$ and $\tau = 3.19$) with synthetic network data

Figure 12: Illustration of the best-fit distribution output by our framework for the Semantic Scholar Open Research Corpus. Captions of subfigures report the best fit $\alpha$ and $\tau$.

$$\operatorname{argmin}_{f:\,\text{density on }\Omega} \; \operatorname{Err}_{\ell,\alpha}(f_{\mathcal{D}}, f) := \int_{\Omega}\left[\int_{\Omega}\ell_{\alpha}(x,v)f(x)d\mu(x)\right]f_{\mathcal{D}}(v)d\mu(v), \text{(OptProg-App)}$$
$$\text{such that} \quad -\int_{\Omega}f(x)\log f(x)d\mu(x) \geq \tau.$$

An instance of this program is specified by the following parameters.

1. A domain $\Omega \subseteq \mathbb{R}$ (e.g., $\Omega = \mathbb{R}$ and $\Omega = [1, \infty)$);

2. A true density $f_{\mathcal{D}}$ over $\Omega$ with respect to the Lebesgue measure $\mu$;

3. A loss function $\ell \colon \Omega \times \Omega \to \mathbb{R}$ (e.g., $\ell(x,v) = (x - v)^2$ and $\ell(x,v) = \ln(x/v)$);

4. A risk-averseness (or risk-eagerness) parameter $\alpha > 0$; and

5. A resource-information parameter $\tau > 0$.

Recall that $\ell_{\alpha}$ is a risk-averse loss defined by $\ell$ and $\alpha$ as in (4). For our simulations, we consider the shifted variant of $\ell_{\alpha}$ mentioned in Section 2: given a shift parameter $v_0 \in \mathbb{R}$, a loss function $\ell \colon \Omega \times \Omega \to \mathbb{R}$, and parameter $\alpha > 0$

$$\ell_{\alpha,v_0}(x,v) = \begin{cases} \alpha \cdot \ell(x, v + v_0) & \text{if } x > v + v_0, \\ \ell(x, v + v_0) & \text{otherwise.} \end{cases}$$

Let $f^\star_{\alpha,\tau,v_0}$ be the optimal solution to the instance $\mathcal{I}_{\alpha,\tau,v_0} = (\Omega, f_\mathcal{D}, \ell, \alpha, \tau, v_0)$ of (OptProg).

**Algorithmic task.** Given a "target" density $f_\mathcal{T}$ (denoting the density of biased utilities in the data), risk-averse loss function $\ell_\alpha$, and true density $f_\mathcal{D}$, the goal of our implementation is to find $\alpha^\circ$, $\tau^\circ$, and $v_0^\circ$ that minimize the total variation distance between $f_\mathcal{T}$ and $f^\star_{\alpha,\tau,v_0}$:

$$(\alpha^\circ, \tau^\circ, v_0^\circ) \coloneqq \operatorname*{argmin}_{\alpha,\tau,v_0} d_{\mathrm{TV}}(f^\star_{\alpha,\tau,v_0}, f_\mathcal{T}).$$

**Algorithmic approach and implementation.** We perform grid-search over all three parameters $\alpha, \tau$, and $v_0$. Given a specific $\alpha, \tau$, and $v_0$, to solve the above problem, we use the characterization in Theorem 3.1 to find $f^\star_{\alpha,\tau,v_0}$. Recall that the optimal solution of (OptProg) is of the following form

$$f^\star_{\alpha,\tau,v_0}(x) = C \cdot \exp\left(-I_{\alpha,v_0}(x)/\gamma^\star\right)$$

where $I_{\alpha,v_0}(x) \coloneqq \int_\Omega \ell_{\alpha,v_0}(x,v) f_\mathcal{D}(x) d\mu(x)$ and $C, \gamma^\star > 0$ are constants that are uniquely specified by the following two equations

$$\int_\Omega f^\star_{\alpha,\tau,v_0}(x) d\mu(x) = 1 \quad \text{and} \quad -\int_\Omega f^\star_{\alpha,\tau,v_0}(x) \log\left(f^\star_{\alpha,\tau,v_0}(x)\right) d\mu(x) = \tau.$$

Algorithmically, finding $C$ and $\gamma^\star$ requires computing a double integral over $\Omega$. In all of the simulations in Section 4, $\Omega$ is a discrete domain, so these integrals reduce to summations and we compute them exactly. We also provide an implementation of our algorithm for continuous domains. The implementation for continuous domains uses the quad function in scipy to compute the integrals. For the grid search itself, we varied $\alpha$ over $[10^{-4}, 10^2]$, $\tau$ over $[10^{-1}, 10]$, and $v_0$ over $\Omega$. We found this range to be sufficient for our simulation, but it would be interesting to design a principled way of specifying the ranges given other parameters and target density $f_\mathcal{T}$.

**Implementation details of multiplicative bias model [90] and implicit variance model [61].** Recall that the multiplicative bias and the implicit variance models are specified by parameters $\rho$ and $\sigma$ respectively: given a fixed true value $v \in \mathbb{R}$, the output of the multiplicative bias model is $v/\rho$ and the output of the implicit variance model is $v + \zeta$ where $\zeta$ is a zero-mean normal random variable with variance $\sigma^2$. In addition, we allow both models to introduce a shift $v_0$. For the multiplicative bias model, given a true density $f_\mathcal{D}$ and a target density $f_\mathcal{T}$, we compute $(\rho^\circ, v_0^\circ)$ that solves $\operatorname{argmin}_{\rho,v_0} d_{\mathrm{TV}}(f_{\rho,v_0}, f_\mathcal{T})$ where $f_{\rho,v_0}$ is the density of $(v/\rho) + v_0$ for $v \sim f_\mathcal{D}$. For the implicit variance model, given a true density $f_\mathcal{D}$ and a target density $f_\mathcal{T}$, we compute $(\sigma, v_0)$ that solves $\operatorname{argmin}_{\sigma,v_0} d_{\mathrm{TV}}(f_{\sigma,v_0}, f_\mathcal{T})$ where $f_{\sigma,v_0}$ is the density of $v + v_0 + \zeta$ for $v \sim f_\mathcal{D}$ and a zero-mean normal random variable $\zeta$ with variance $\sigma^2$. For both models, we compute the optimal parameters using grid search: we vary $v_0$ over $\Omega$, $(1/\rho)$ over $[0, 1]$, and $\sigma$ over $[10^{-2}, 10]$.

### L.3.2 Computational resources used

All simulations were run on a MacBook Pro with 16 GB RAM and an Apple M2 Pro processor.

### L.3.3 JEE-2009 Scores

**Additional discussion of the dataset.** The JEE-2009 test scores were released in response to a Right to Information application filed in June 2009 [91]. This dataset contains the scores of all students from JEE-2009 (384,977 total) [91]; we used the version available provided by [40]. In addition to the scores, for each student, the data contains their self-reported (binary) gender and their birth category. The birth category of a student is an officially designated indicator of their socioeconomic group, where the general (GEN) category is the most privileged; see [135, 19] for more details.

We observe that students not in the GEN category have significantly lower average scores than students in the GEN category (18.2 vs. 35.1); this may not imply that students not in the GEN category would perform poorly if admitted. Indeed, among students of equal true "potential," those from underprivileged groups are known to perform poorer on standardized tests [60]. In the Indian context, this could be due to many reasons, including that in India, fewer students outside the GEN category attend primary school compared to students from the general category, and on average a lower amount of money is spent on the education of students in the non-general category compared to the general category [92].

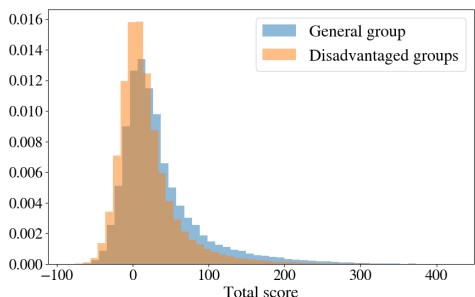

Figure 13: Distribution of scores in the JEE dataset for different protected groups based on birth category. See Section 4 for a discussion of the dataset.

### L.3.4 Semantic Scholar Open Research Corpus

**Cleaning and predicting author names.** We follow the procedure used by [40]. Concretely, we remove papers without publication year (1.86% of total) and predict author gender using their first name from a publicly available dataset [3], containing first names and gender of everyone born between 1890 to 2018 and registered with the US social security administration (USSSA). We remove authors whose first name has 2 or fewer characters, as these names are likely to be abbreviations (retaining 75% of the total), and then categorize an author as female (respectively male) if more than $\phi = 0.9$ fraction of the people of the same first name are female (respectively male) in the USSSA data. We drop all uncategorized authors (32.25% of the remaining). This results in 3,900,934 women and 5,074,426 men (43.46% females). We present the tradeoff between the total number of authors retained and $\phi$ in Figure 14.

**Counting the number of citations.** We aim to ensure that the citation counts we compute correspond to the total citations received by an author over their lifetime (so far). Since the dataset only contains citations from 1980 onwards, we remove authors who published their first paper before 1980 as the dataset does not have information about their earlier citations. This is the same as the cleaning procedure used by [40]. We present the resulting citation-distributions for male and female authors respectively in Figure 15.

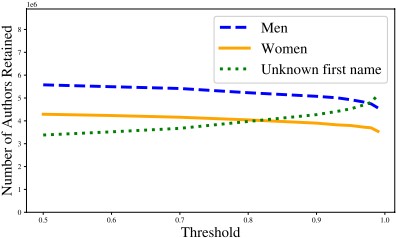

Figure 14: The tradeoff between the threshold $\phi$ used for clearing the Semantic Scholar Open Research Corpus and the number of authors retained. Details appear in Appendix L.3.4.

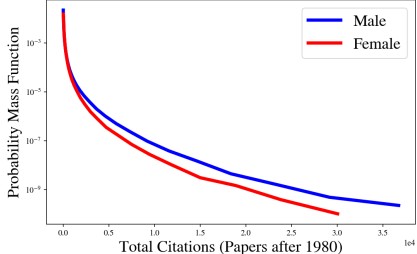

Figure 15: Distributions of total citations of men and women in the Semantic Scholar Open Research Corpus. Details appear in Appendix L.3.4.

