# OpenReview forum: "Bias in Evaluation Processes: An Optimization-Based Model"
_NeurIPS.cc/2023/Conference — NeurIPS 2023 poster_

### Official Review · Reviewer_Aa19 · 2023-06-27

**Soundness:** 2 fair
**Presentation:** 2 fair
**Contribution:** 2 fair
**Rating:** 4
**Confidence:** 3

**Summary:**

This paper studies the issue of biases present in evaluation processes like hiring and school admissions. The authors propose a model that estimates the distribution of utility that incorporates two main features: resource constraints for information, and risk-averseness of the decision-maker. They formulate an optimization problem to estimate the utility distribution with two parameters that represent the aforementioned two features. They study the effect of these two parameters on the solution of the optimization problem, and they conduct a numerical study to study the effect of interventions in a downstream selection task.

**Strengths:**

The problem of bias in evaluation processes is a significant problem, and I like the approach of understanding how bias can emerge in this process via a stylized model. The authors model two important phenomena, resource constraints and risk aversion, that have been shown to arise in many settings and can contribute to bias. The authors use real-world datasets to validate their study.

**Weaknesses:**

The exposition of the paper was poor, making it challenging to comprehend the main message of the work. I understood the paper as positing a stylized model of an evaluation process that incorporates resource-information and risk averseness, and the main question that is studied is to understand how these two features contribute to the emergence of bias. However, I did not get a satisfactory understanding of this question from reading this work.

1. The main point of confusion was the lack of a formal model of bias, and an interpretation of what the main framework, OptProg, represents in reality. OptProg outputs a density of scores, $f_{\epsilon}$, where $f_{D}$ is a “true” distribution of scores. I interpret this via the following example: the distribution of SAT scores across all students ($f_{\epsilon}$) is not equal to the distribution of “true ability” of same students ($f_{D}$). However, only looking at the distribution seems insufficient for understanding “bias”, since it does not specify how _each_ true score gets mapped to each “biased” score. This is the source of the issue in some of my next comments.

2. Section 3.2 studies how tau and alpha influence the solution to OptProg, which seems to be the main contribution of this work. However, this section relies on specific distribution examples and numerical investigations, with long discussions and alot of notation that was difficult to follow. Moreover, it fails to explain why mean and variance are the relevant statistics of interest and how they capture the notion of "bias". Providing explicit theorem statements would have greatly improved this section.

3. The numerical study had several parts that I found confusing:
- in Section 4.1, fitting the OptProg model to real-world datasets and evaluating the TV distance does not demonstrate the model's ability to capture "Ability to capture biases in data" (section title). Why does a small TV distance imply bias?
- The model fits the best tau and alpha, but wouldn’t tau = -\infty always be the best tau? Also, is it the case that alpha > 1 is a better fit than alpha <= 1?
- The interventions aimed at changing tau and alpha from the best fit would increase the error in estimation (increase TV). Why would these interventions improve utility? The confusion arises from the lack of a formal model of utility and bias.
- There was no clear conclusion from the numerical results: each intervention was the best in some regime. What should the reader take away from this, and how should a DM use these results?
- To strengthen the study, I suggest including numerical analyses that (a) validate the model and (b) rigorously estimate alpha and tau for different groups to demonstrate the existence of this type of bias.

**Questions:**

Please let me know if I have any misinterpretations in my review.

**Limitations:**

Yes.

---

> ### Author Rebuttal · Authors · 2023-08-10
>
> Thank you for your feedback. We are glad that you like our approach. There do seem to be some misinterpretations that we have clarified. We hope that you will increase your support for the paper.
>
>
> **"...lack of a formal model...what...OptProg...represents"** OptProg is an abstract model of how an input distribution of true utility of a group of individuals gets transformed by an evaluation process to an output distribution in the presence of risk-averseness and information constraint. Since the corresponding parameters $\alpha$ and $\tau$ depend on socio-economic factors, we chose to focus on the input/output of OptProg at a population level. However, the intuition for OptProg does come from understanding how an individual's true ability $u$ gets mapped to a biased score. Indeed, we can recover this behavior from OptProg by plugging in $f_\mathcal{D}$ to be the Dirac delta distribution around $u$ and setting $\alpha$ and $\tau$ appropriately. We will clarify this.
>
> **"...mean and variance..."** Our work is rooted in several prior works that have argued that mean and variance of population level distributions are relevant statistics in understanding bias. E.g., [81] points out that in the distributions of lifetime citations of men and women scientists, the average number of citations is lower for women than men. As for variance, quoting from [57] -- *different groups of candidates may exhibit different variability when their quality is estimated through a given test*.
>
> Several theorems on mean and variance appear in the Supplementary Material (Theorems D.1, D.5, D.7, E.1, E.2). These theorems discuss the effect of changing $\tau$ and $\alpha$ on the mean/variance of the output distribution. Based on your suggestion, we will include informal versions of them in the main body.
>
>
> **"...Why does a small TV distance imply bias?"** Sorry, this is a typo. We intended to say that a small TV distance between the best-first distribution generated by our model and the distribution of utilities in a dataset (which is already known to be biased; see the note below) implies that our model can generate distributions that capture biases arising in real-world datasets. We will fix this.
>
> Note: All of the real-world datasets that we consider are already known to have biases in utilities across the  protected groups (defined by gender and/or birth category for the JEE-2009 dataset and defined by gender for the semantic scholar dataset); see, for instance, [41] and [57].
>
> **"...wouldn’t $\tau = -\infty$ always be the best tau?"** Setting $\tau=-\infty$ just leads to the output density having the least amount of uncertainty. E.g., when $f_{\cal D} = N(\mu,\sigma^2)$, $\tau=-\infty$,  $\alpha=1$, $\ell(v,x)=(v-x)^2$, the output of OptProg is the Dirac-delta function at $\mu$. This is not the best fit solution. Whereas if we pick $\tau$ to be equal to the entropy of $f_{\cal D}$, the output is same as $f_{\cal D}$ which, by definition, is the best-fit distribution.
>
> We report the best-fit $\tau$ in Figure 6 of the Supplementary Material.
>
> **..is it the case that $\alpha > 1$ is a better fit than $\alpha \leq 1$?"** Not in general. For the Semantic Scholar dataset and the synthetic network data, $\alpha>1$ is a better fit than $\alpha\leq 1$. However, for the JEE-2009 dataset, the opposite holds: $\alpha\leq 1$ is a better fit than $\alpha>1$. We report the best-fit $\alpha$ in Figure 6 of the Supplementary Material. Concretely, Table 1 shows that the TV-distances with $\alpha=1$ are poorer than with the best-fit $\alpha$ (0.12, 0.11, 0.08, and 0.02 compared to 0.08, 0.07, 0.03, and 0.01 respectively).
>
>
>
> **"...Why would these interventions improve utility?"** Changing $\tau$ and $\alpha$ from their best-fit values (for the biased distribution values) does increase the error in estimation from the biased distribution of values, but it may decrease the error in estimation from the true distribution of values. Since utility is measured as the sum of true (and not biased) values of the selected individuals (Lines 352 – 359), changing $\tau$ and $\alpha$ from their best-fit values can improve utility. We will emphasize this further in the final version.
>
> **"...each intervention was the best in some regime...how should a DM use these results?"** That is correct. Given the regime that is of interest to a policymaker, the policymaker can use our model to study the effects in order to systematically identify the best intervention and, based on its assessed efficacy, decide whether to enforce it (Lines 395-397). As mentioned in the abstract (and the introduction), our work provides a tool for the policymaker to guide the deployment of interventions to mitigate biases. Empirical sections provide use cases that may help policymakers to see how they can use our model. Prescribing which intervention to use in which context is outside the scope of this work.
>
> **"...I suggest including numerical analyses that (a) validate the model and (b) rigorously estimate alpha and tau for different groups..."** Perhaps there is some confusion, we do validate our model: We estimate $\alpha$ and $\tau$ for utilities of disadvantaged groups in three real-world datasets (Lines 12 - 13).  We discuss implementation details of the estimation in Lines 303-310, report the resulting TV-distances between the best-fit distribution in Table 1, and report the best-fit values and distributions in Figure 6 in the Supplementary Material. We observe that the resulting distributions output by our framework are close in TV distance (they have TV-distances $\leq 0.08$; Table 1) to the distributions of biased utilities in the real-world data. Moreover, our model has a better fit than the implicit variance model [57] and the multiplicative bias model [81] on datasets where utilities have skew (Rows 1, 3, and 4 in Table 1) (Lines 115-116).

---

> > ### Comment · Reviewer_Aa19 · 2023-08-16
> >
> > Thank you to the authors for their response.
> >
> > I don't quite understand the authors’ first response, which was my main concern that there was no stated mechanism that maps "true ability” to “biased ability”. Could the authors spell this out further?
> >
> > Next, if there is such a mapping, then the goal of these interventions should be to _recover_ the density of true abilities, from the biased abilities (i.e. "correct" the bias). But it doesn’t seem like the interventions are aiming to measure "recovery". For example, increasing tau increases both the mean and variance (Section 3.2), but what does that mean in terms of correcting bias? Let me know if this is not the goal of the interventions being studied.

---

> > > ### Author Response · Authors · 2023-08-17
> > > **A concrete example of how "true ability" gets mapped to "biased ability"**
> > >
> > > Thanks for reading our rebuttal and responding. We answer your question below and will be happy to provide further clarifications.
> > >
> > > **1)"...mechanism that maps true ability to biased ability. Could the authors spell this out further?"**
> > >
> > > Certainly. Consider the setting where a single individual with "true ability" $u$ is being evaluated. One concrete way in which our framework can model this is by setting the input density $f_{\mathcal{D}}$ to be concentrated on $u$ and letting the loss function to be $\ell_2^2$-loss (see Lines 204-205 for a definition) over the domain $\mathbb{R}$. For fixed parameters $\tau$ and $\alpha$, the output distribution $f^\star$ (over the "biased ability") can be shown to have a mean of $u - \sqrt{\frac{\gamma^\star}{2}}\cdot \frac{\sqrt{\alpha}-1}{\sqrt{\alpha}+1}$. Here $\gamma^\star > 0$ is as in Theorem 3.1 and is a function of $\tau$ and $\alpha$.
> > >
> > > Thus, our framework allows one to derive a mapping from "true ability" to the (mean of the) "biased ability".
> > >
> > > This mapping can be used to understand how the parameters $\tau$ and $\alpha$ in the evaluation process transform  the true ability. For instance, using the fact that $\gamma^\star$ increases as $\tau$ increases, and that $\frac{\sqrt{\alpha}-1}{\sqrt{\alpha}+1}$ is positive for $\alpha > 1$, it follows that decreasing $\tau$ pushes the expected biased ability of the individual up towards $u$, their true ability.
> > >
> > > Similar mappings can be derived for other loss functions as well and, if one is interested in the distribution of biased abilities, Theorem 3.1 gives a characterization.
> > >
> > >
> > > **2)"...then the goal of these interventions should be to recover the density of true abilities, from the biased abilities (i.e. "correct" the bias)."**
> > >
> > > Yes, that could be one goal, however, it may neither be necessary nor directly achievable. Obtaining estimates of abilities is not an end to themselves: they are used in downstream tasks such as selection. The interventions (in our paper and in prior works) try to ensure that the outcomes of the downstream tasks with biased abilities and interventions are (approximately) the same as the outcomes with true abilities.
> > >
> > > **3)"For example, increasing tau increases both the mean and variance (Section 3.2), but what does that mean in terms of correcting bias?"**
> > >
> > > Concretely, as discussed in the example point (1) above, decreasing $\tau$ moves (mean of) the biased ability of the individual up toward their true utility. Moreover, as can be seen by the expression of the mean of the biased utility in point (1) (i.e., $u - \sqrt{\frac{\gamma^\star}{2}}\cdot \frac{\sqrt{\alpha}-1}{\sqrt{\alpha}+1}$), moving $\alpha$ towards $1$ also ensures that the mean of the biased ability of the individual approaches their true ability.

---

> > > > ### Comment · Reviewer_Aa19 · 2023-08-21
> > > >
> > > > Thanks to the authors for their response. I have raised my score to a 4. I am still concerned about the presentation of the paper (I still find the paper to be very confusing, even with the author’s clarifications), which I don’t think can be addressed in a small revision.

---

> > > > > ### Author Response · Authors · 2023-08-21
> > > > > **Plan for revision**
> > > > >
> > > > > Thank you for raising your score.
> > > > >
> > > > > Our discussion with the reviewers has been invaluable, and because the revisions are a matter of clarifications (which we have already outlined in the rebuttal) and not of developing new material or empirical results, we are confident that we can complete them in a timely manner.
> > > > >
> > > > > We provide a detailed summary of changes below and believe that we will be able to incorporate these changes within a week. In addition, we will correct typos and address any minor comments omitted in the list below. Note that one extra page is allowed for the final version, which allows sufficient space for these changes. If the AC allows, we would be happy to upload our revision by early next week.
> > > > >
> > > > > **Section 2**
> > > > >
> > > > > * We will explain the mechanism that maps true ability to biased ability in Section 2. (Included in responses to Aa19.)
> > > > >
> > > > > * We will further explain the use of max-entropy constraint and contrast our approach with density estimation in Section 2. (See the response to 7ABm.)
> > > > >
> > > > > * In Section 2, we will add a remark on how our model extends to multiple sensitive groups with overlapping attributes (We present the idea in the response to 7ABm.)
> > > > >
> > > > >
> > > > > **Section 3**
> > > > >
> > > > > * We will add an overview of the proof of Theorem 3.1 in Section 3.1. (Presented in the "Author Rebuttal by Authors" above.)
> > > > >
> > > > > * We will add informal versions of key results in Sections D and E (on how changing $\alpha$ and $\tau$ affect the mean and the variance of the output density) in Section 3.2.
> > > > >
> > > > > * We will add two theorems that show how our framework captures the implicit variance and the beta-bias models in Section 3.2 (the statements appear in the response to G1nQ). The proofs follow from the results in Sections F and G in the supplementary material and will be added as a separate section after Section G in the supplementary material. (Presented in the response to G1nQ.)
> > > > >
> > > > > * In Section 3.2, we will replace the plots in Figure 1 with 2-dimensional plots which show the effect of  $\alpha$ or $\tau$ on the mean of the distributions. These plots are also shown in the attached 1-page pdf file. If space is a constraint, we will move them to the Supplementary material.
> > > > >
> > > > >
> > > > > **Section 4**
> > > > >
> > > > > * We will add a version of Figure 6 that compares the best fits of our model and the best fits of previous models with the distribution of biased utilities in Section 4.
> > > > >
> > > > > * In Section 4 (and corresponding sections in Supplementary Material), we will update the experimental validation with the train-test split. (See the response to gzsA and the table in the one-page pdf attached with the rebuttal.)
> > > > >
> > > > > * Time permitting, we will incorporate the suggestion of gzsA to include further empirical evaluation to give qualitatively better insights to compare different policies.
> > > > >
> > > > > **New section between Section 4 and Section 5**
> > > > >
> > > > > * We will add a section between Sections 4 and 5 to include specific examples of mechanisms by which information constraints and risk aversion lead to bias. Here we will briefly discuss examples of concrete interventions (details are given in Section H) and how policymakers can use our model. (Presented in the "Author Rebuttal by Authors" and in the response to G1nQ.)
> > > > >
> > > > > **Section 5**
> > > > >
> > > > > * We will add a short discussion on the limitations of our model when it comes to long-term applicability in Section 5.
> > > > >
> > > > > * We will add a comment on the potential misuse of our model and its negative societal impact in Section 5. (See the response to G1nQ.)
> > > > >
> > > > > * We will mention the limitation of our work in understanding the effect of the interventions considered in our paper in the long term in Section 5.
> > > > >
> > > > > * We will add a discussion on the limitations of our model when there are differences in the distribution of true utility among the two groups at the population level in Section 5. (See the response to gzsA.)
> > > > >
> > > > > * We will add a comment in Section 5 on how the output of our model can be used by policymakers to assess the impact of interventions in this supermodular setting as well. (See the response to 2dLt.)

---

### Official Review · Reviewer_gzsA · 2023-07-05

**Soundness:** 3 good
**Presentation:** 3 good
**Contribution:** 2 fair
**Rating:** 6
**Confidence:** 4

**Summary:**

The paper proposes an optimization-based framework for modeling bias in evaluations. The perspective of the paper is to provide a well-founded and interpretable model of evaluations that can replicate biases observed in real settings without invoking an intrinsic utility for producing biased evaluations. This model is a generalization of previous models for evaluation bias which assume a particular parameterized model for the output density (e.g. Gaussian or Pareto). The primary utility of this model is to help policy-makers study the effectiveness of different interventions aimed to reduce bias, such as requiring proportional fairness of evaluations or decreasing the informational cost of evaluations. The authors apply their model to data such as IIT JEE-2009 scores, obtaining a closer fit with realized scores than previous models, and test out different interventions.

Specifically, the paper models the density of evaluations f_{\mathcal{E}} as the output of an optimization problem, in which the evaluator seeks to produce an evaluation as close to the true value v of the individual as possible, which is drawn from a density f_{D}. Closeness is measured by a loss function \ell(x,v) between evaluation x and quality v. This optimization problem is subject to constraints and adjustments which introduce bias, which are well-founded in the literature. The first is information constraints. Evaluations are noisy in practice because it is difficult and costly to obtain a clear evaluation signal, and this cost can vary across different groups (e.g. groups that speak different languages). This is modeled as a constraint requiring that the entropy of f_{\mathcal{E}} be lower-bounded by some \tau. The second source of bias is risk-aversion: the evaluator's loss may not be symmetric between over and under-estimation of the true value, and may penalize over-estimation more. This is modeled in the loss function between the evaluation and the true value v, in which loss aversion (parameterized by \alpha\geq0) causes the evaluator to penalize over-estimation of v more than under-estimation by a factor of \alpha. They derive an exponential functional form for the solution of this optimization problem, which is standard by the maximum-entropy principle. They also perform sensitivity analysis of how the solution changes with \tau and \alpha.

This modeling framework generalizes the implicit variance and multiplicative bias models. The only inputs to the model are the information constraint \tau, risk aversion parameter \alpha and loss function \ell. For empirical validation, they consider two real-life datasets and one synthetic example of scores in different contexts. They fix a group G_{1} of individuals to be a baseline group and use the distribution of scores from that group to represent the true distribution of values. They consider another group G_{2} whose scores are potentially biased (and thus arising from f_{\mathcal{E}}), and they compare how well different models of evaluation bias reproduce the distribution of G_{2} using scores from G_{1} as the true values. They find that their more generalized model can improves the fit of the score distribution of G_{2} across all datasets. They then use their model to assess various interventions for subset selection tasks, based on whether they reduce bias and allow the evaluator to pick a subset of individuals with higher expected value. These interventions include requiring proportional or equal representation, increasing \tau, and lowering \alpha, which all are related to interventions considered in practice. They find that optimal interventions depend case by case based on different parameters of the subset selection task (e.g. how many individuals to hire, etc.), and their model can guide policy-makers to understand when certain interventions will be more effective than others.

**Strengths:**

The greatest strength of the paper is in its conception. Invoking the maximum entropy principle obtains a natural generalization of previous models in the literature, without involving an excess of extra parameters. Including risk aversion makes sense too, given that it is a well-studied source of bias and better enables the framework to model skewed distributions. The method is computationally feasible since evaluation scores are 1-D and ultimately discrete, and produces realistic evaluation distributions. The optimization-based formulation retains much of the interpretability of simpler models, while crucially allowing for greater modeling capacity through computation. In contrast to previous papers, which are primarily concerned with producing simple models that illustrate a particular source of bias, this paper provides a computational framework designed to be applied to data (I believe this aspect should be emphasized more in the paper).

The paper is clearly written and the empirical benchmarking is solid with compelling examples (JEE-2009 and Semantic Scholar). The evaluation of different interventions is also a nice illustration of the usefulness and interpretability of the model.

**Weaknesses:**

Overall the paper could benefit from more exposition. While the paper references key sources in the literature, it does not clearly explain the mechanism by which information constraints and risk aversion lead to bias. This could be clarified by 1 or 2 specific examples. Without this, it is difficult to understand why these are the particular sources of bias incorporated in the framework and why others are not. It would also be better to spend more time discussing concrete interventions, e.g. moving some of the material from Supplemental Material H to the main text. To aid in this, would it be possible to evaluate an intervention like 'change score from out of 100 to out of 10' or 'truncate the range of scores'? If so, this could help in demonstrating the applicability of the model for helping with realistic policy decisions.

The theoretical results on the evaluator's optimization problem seem to be fairly standard and straight-forward implications of the maximum entropy principle. This is not a bad thing, but at least this section could be trimmed down and streamlined. The proof of Theorem 1 seems fairly involved, and it would be useful to explain how this setting departs from standard maximum entropy settings. The results on sensitivity analysis with respect to \tau and \alpha are also not surprising and this section could be shortened and crystalized to better capture how \tau and \alpha affect the output density (with greater focus on how the mean changes). Figure 1 is difficult to interpret and could either be removed entirely or replaced.

I think the biggest need for improvement is in showing the strength of the contribution compared to previous models. I think a strong case can be made using supplemental material, but from the current draft alone the strength of the contribution is not very clear. First, it would help to use some supplemental material (e.g. Figure 6 from Supplemental Material H) into the paper to show that a wider class of models is actually necessary for modeling real life score distributions (comparing the best fits from other models). One could also look at a synthetic example with exponential or laplacian tails, but this is not necessary.

And since the framework a generalization of previous models, it is perhaps unsurprising that it achieves greater fit to data. More than fitting the data, ultimately what matters are implications for intervention evaluation/sensitivity analysis for policy-making. If the conclusions are identical to what one obtains from less sophisticated but simpler models, it's unclear whether a more powerful modeling class actually helps. What would have been more compelling is if there are instances where the current model gives different answers for what interventions work well or not compared to simpler models, and justifying why the current model's conclusions are more sensible. In that light, it would be useful to evaluate how well the framework assesses interventions, which can be done through synthetic examples (e.g. intervening on the synthetic example and seeing how well the model predicts the effect of the intervention).

**Questions:**

When fitting \alpha and \tau to minimize total variation distance with the G_{2} distribution, is there any train-test splitting, or are they fit on all the data? Otherwise there might be a concern about overfitting.

In the empirical evaluation, the underlying assumption is the distribution of true values for the G_{2} individuals is exactly the same as the distribution for G_{1}. While this assumption makes sense for convenience since we don't observe the true values of G_{2}, if this assumption were false in practice, how could this affect the validity when assessing different interventions?

**Limitations:**

The paper primarily mentions the limitation of the work as being that it only concerns with scalar values, and mentions that the framework can be extended to multivariate values. The paper also mentions that there may be other models out there that achieve the same fit as the maximum entropy one. This is all reasonable. It does not include the limitation that the distribution of true values for G_{2} is assumed to be identical for G_{1}, in which case one cannot disentangle evaluation bias from natural differences in these distributions.

---

> ### Author Rebuttal · Authors · 2023-08-10
>
> Thank you for your feedback. We are glad that you think that our model can guide policy-makers to understand when certain interventions will be more effective than others. Thanks also for appreciating the invocation of the maximum entropy principle to generalize previous models.
>
> **"proof of Theorem 1...”** Please see our response [here](https://openreview.net/forum?id=7b4oobeB4w&noteId=2eKb6fBuEX).
>
>
> **"…specific examples…"** While we do explain how changing $\alpha$ and $\tau$ give rise to biases, we present a couple of concrete settings to show how $\alpha$ and $\tau$ may arise and give rise to biases in an evaluation process [here](https://openreview.net/forum?id=7b4oobeB4w&noteId=2eKb6fBuEX).
>
> **“…concrete interventions…e.g. moving material from Supplemental Material H.”** Thanks. We will move the discussion on interventions from Supplementary Material H (Lines 1362-1381) to the main body.
>
>
> While such interventions (e.g. conducting the exam in multiple languages) can be assessed by our framework, to evaluate them, one needs a model of how the intervention affects $\tau$ and $\alpha$, and designing such models is beyond the scope of this work.
>
> That said, for interventions that directly manipulate the values, such as truncating the range of the values as suggested by you, one can evaluate their effect on $\alpha$ and $\tau$ by computing the best-fit distributions to the truncated value distribution and, hence, use our framework to assess the efficacy of these interventions.
>
>
> **"...could be shortened...to better capture how \tau and \alpha affect the output..."** Thanks for the suggestions.  We will trim the discussion of sensitivity analysis with respect to $\tau$ and $\alpha$, which focus on their effect on the variance and the  mean of the output density. Even though the sensitivity analysis with respect to $\tau$ and $\alpha$ shows that the variance and the mean of the output density change as expected, proving these results are non-trivial. The key reason is that we do not have a closed form expression for the optimal density in terms of $\alpha$ and $\tau$. We can only express the optimal density in terms of the optimal dual variable $\gamma^\star$ (see Thereom 3.1), but understanding the sensitivity properties of $\gamma^\star$ turns out to be  non-trivial. We draw intuition from the analogy with Gibbs equation (see equation (6) and Section C in the supplementary material), which helps us in predicting the properties of $\gamma^\star$. Sections D, E, F, and G in the Supplementary Material are dedicated to the proofs of these results.
>
> **"Figure 1...could be...replaced."** Thanks, we simplified Figure 1: instead of the three-dimensional plots, we present two-dimensional plots that show the effect of $\alpha$ (respectively $\tau$) on the mean of the distributions for a fixed value of $\tau$ (respectively $\alpha$). The updated figure is in the one-page PDF attached with our rebuttal.
>
>
> **“...contribution compared to previous models…would help to use...Figure 6…”** Thanks for your suggestion, we will add a version of Figure 6 that compares the best fits of our model and the best fits of previous models with the distribution of biased utilities in the final version. As expected from the TV-distance values in Table 1, the best fit achieved by our framework is at least as good as the best fit of the implicit variance model [57] and the multiplicative bias model [81], and is a better fit than these models [57, 81] on datasets in which utilities have skew (Rows 1, 3, and 4 in Table 1).
>
> **“...instances where the current model gives different answers...compared to simpler models…”** Yes, indeed, for certain interventions that affect both $\tau$ and $\alpha$, our model may lead to different assessments than simpler models. For instance, in the context of standardized testing, consider the intervention that requires conducting the exam in multiple languages. This intervention can reduce the information constraint faced by the disadvantaged group (by reducing the cognitive load on the non-native speakers during the examination) and, at the same time, may also reduce risk-aversion parameter $\alpha$ (by eliminating the need for non-native speakers to enroll in additional training for the examination language). For such interventions, simpler models–that only consider the effect of the intervention along one dimension–may underestimate its positive impact while our framework could give a more accurate assessment. We will include a discussion on this in the final version.
>
> **"...train-test splitting"** Thank you for your suggestion. We did not use a train-test split when fitting $\alpha$ and $\tau$. We have repeated the simulation with an 80-20 train-test split and will include them in the final version.
>
> The results with the train-test split are similar to the results in the paper: across all datasets, the densities output densities by our model are close to the density of biased utilities in the datasets (TV distance $\leq 0.09$) and our model has a better fit than the implicit variance model [57] and the multiplicative bias model [81] on datasets where utilities have skew (Rows 1, 3, and 4 below).
>
> Concretely, we report the TV-distances with a train-test split in the one-page PDF attached with our rebuttal.
>
> **"..G_{2}...is ...same as...G_{1}....if this assumption were false..."** This assumption builds on the premise that there are no differences (at a population level) between $G_{1}$ and $G_{2}$; see, e.g., [81,57,41]. If this premise is false, then the effectiveness of interventions can be either underestimated or overestimated which may lead a policymaker to select a suboptimal intervention. That said, if all the considered interventions reduce risk aversion and/or resource constraints, then the chosen intervention should still have a positive impact on the disadvantaged group. We will add a brief discussion in the limitations section on this.

---

> > ### Comment · Reviewer_gzsA · 2023-08-17
> >
> > Thank you for the thoughtful response. I appreciate the explanation for the proof of Theorem 1. I also really appreciate the explanations for why \alpha and \tau would affect bias, they help to illustrate why these parameters are invoked in the model and why other parameters are not. All of my concerns and questions have been addressed by the above response.
> >
> > I do still wish there were a more extensive empirical evaluation that could clearly show that this more computationally powerful method gives _qualitatively_ better insights into which policies would work better than others. And also if there is any way to concretely interpret the fitted values of \tau and \alpha. I think this is a mostly a matter of rewriting the paper so that it's usefulness to potential policymakers (or even possibly empirical social scientists) is more apparent.
> >
> > I will stick to my rating since I believe it is appropriate. Once again, I like the core idea of the paper, my comments are only on the presentation.

---

### Official Review · Reviewer_2dLt · 2023-07-06

**Soundness:** 3 good
**Presentation:** 4 excellent
**Contribution:** 3 good
**Rating:** 5
**Confidence:** 3

**Summary:**

This work presents a theoretical model to quantify bias in the task of evaluating candidates (ie minimizing loss while subject to an information constraint). It presents a formula/representation of the problem, parametrized by roughly "real-world" factors of 1) resource-information tradeoffs; and 2) risk-aversion. After presenting some properties of this model, the work loosely applies it to quantify types of bias in real-world datasets (eg standardized testing by class/gender; citations; etc). Finally, it explores how different real-world-inspired interventions (eg Rooney Rule for representational constraint, structured interviews for standardization) could impact the state of bias in the parametrized models.

edit: I have had the rebuttal and feel better about the derivations. I remain borderline accept

**Strengths:**

- I quite like the design decision to choose model parameters and intervention types that are inspired by plausible tradeoffs and concerns in evaluation bias.
- This paper would not be as strong if it were just the formulas/models without trying to measure any grounding in real-world datasets.

**Weaknesses:**

- Although engaging with real-world datasets empirically is commendable, the analysis conducted was rather light. It didn't offer particularly novel insights or alternative ways of thinking about bias in the data (eg could try to demonstrate how the model can help policymakers with actionable interventions; I don't think a policymaker would be able to gain such insight with the current presentation of results/discussion)

**Questions:**

- Overall this paper seems very interesting, though I couldn't follow all of the derivations/formulas. The empirical part, alone, doesn't have enough insight, but I do like applying the theoretical parametrized model to the real-world datasets to discuss how to interpret them. I'm inclined to accept, though I would feel more comfortable about this paper if one of the other reviewers were able to speak to the derivations' validity and contribution.

**Limitations:**

- Some of the modeling choices were noticeably reductive. For instance, on page 8, Section 4.2 uses college admissions as an example of a subset selection task. However, it suggests modeling the task is the sum of each accepted applicant's individual values, rather than considering network effects (eg Scott Page's "The Diversity Bonus" where a team of different people can contribute different kinds of knowledge to solution resulting in a better result than if the top-2 individual values had high overlap & therefore didnt offer complementary strengths). Of course, that is just nitpicking one example, however my general concern is that this work may or may not ultimately prove useful enough for policymakers. It might end up being the case that any tasks that are tractable enough to model mathematically are poor fits for social science dynamics in practice)

---

> ### Author Rebuttal · Authors · 2023-08-10
>
> Thank you for appreciating the model, the empirical work, and the presentation of the paper. We have addressed your questions and concerns below and hope that you will consider supporting our paper further.
>
>
> **"I'm inclined to accept .. "**  Thanks. Please take a look at the review of reviewer gzsA concerning the validity and contributions of the derivations.
>
> **" ... how the model can help policymakers with actionable interventions ..."** Most of the prior works on interventions in selection settings have focused on adding representational constraints for the disadvantaged groups. Such constraints, often framed as a form of affirmative action, could be beneficial but may not be possible to implement in certain contexts. E.g., in a landmark 2023 ruling, the US Supreme Court effectively prohibited the use of race-based affirmative action in college admissions. Our model posits that reason for the emergence of differences in distributions of abilities of different groups could be because of the difference in the values of the information-resource parameter $\tau$ and/or the risk-averseness parameter $\alpha$ of the group. The theoretical results establish the impact of these parameters on the mean/variance of the output distributions. Thus, in all, the model allows a  policymaker to evaluate interventions beyond affirmative action that focus on procedural fairness; this allows working towards diversity and equity goals without placing affirmative-action-like constraints.
>
>
> In this framing, we can consider decreasing either $\alpha$ or $\tau$. Improving either would work towards equity, but which ones to target or to what extent and via which method would be context dependent and vary in cost. A decrease in $\alpha$ can be achieved by reducing risk-averseness in the evaluation process; e.g. by investment in better facilities for disadvantaged groups, or making the evaluation process blind to group membership. A reduction in $\tau$ may follow by allocating additional resources to the evaluation process, e.g., by letting a candidate choose an evaluation in their native language. Our framework allows a policymaker to study these trade-offs, and we discuss specific examples in Supplementary Material H.
>
>
>
>
> **"a team of different people can contribute different kinds of knowledge to solution resulting in a better result"** Sorry for the confusion, summing up the utility of each accepted applicant's individual utility in the example of subset selection task is not a modeling choice made in this paper. It is borrowed from prior works [81,57,41].
>
> The focus of this paper is to give a model of how population level differences in utility distributions can arise in evaluation processes. In the empirics, the main goal was to show that the model can explain biases in real-world data sets. The goal of showing the subset selection application was to guide policymakers on the deployment of interventions to mitigate biases.
>
> That said, we appreciate your suggestion. Indeed, works in sociology (such as the one you suggest) have shown that one of the benefits of diverse teams is that complementary skill sets increase the overall production -- the sum is more than the parts. This corresponds to having a supermodular set aggregation function in the selection task. The output of the model can be used by policymakers to assess the impact of interventions in this supermodular setting as well. We will mention this in the final version.

---

### Official Review · Reviewer_7ABm · 2023-07-06

**Soundness:** 2 fair
**Presentation:** 1 poor
**Contribution:** 2 fair
**Rating:** 4
**Confidence:** 1

**Summary:**

The paper studies how to examine the group distributional difference using loss minimization. The authors propose a loss with a max-entropy constraint.

**Strengths:**

The paper studies an important problem of how to examine the evaluation bias in many applications such as hiring and school admissions. The authors nicely motivate the problem and have a detailed related work on how such biases arise in practice.

**Weaknesses:**

The paper became quite hard to read after related work. I had a hard time understand what is the formulated problem and the reasons for many design choices are not clear to me. I do not follow the loss formulation, since it seems that the authors are trying to do a density estimation task with certain constraints on the density function. Why not minimize classic metrics like f-divergence, IPM, or use methods like GMM or kernel density estimation? There is also little explanations on why we should use max-entropy as constraint.

Evaluation: It seems the experiments are a density estimation task for two groups, why usual density estimation methods cannot be applied here?

I am also not convinced the proposed method can be used to examine effect of interventions, since they are dynamic and more information is needed to see the effect of interventions in the long run.

Typo:  “risk averseness:”, incomplete sentence L121-122

**Questions:**

See weaknessses.

**Limitations:**

1. The paper only considers binary sensitive groups, while in practice, sensitive groups are often overlapping with multiple attributes.
2. How would the estimation error and model misspecification lead to negative impact?

---

> ### Author Rebuttal · Authors · 2023-08-10
>
> Thank you for appreciating the paper. We address your questions and concerns below and hope you will strengthen your support for the paper.
>
> **"..since it seems that the authors are trying to do a density estimation task with certain constraints on the density function. Why not minimize classic metrics like f-divergence, IPM, or use methods like GMM or kernel density estimation? There is also little explanations on why we should use max-entropy as constraint.."**
>
> Thanks for this question and sorry for the confusion. Our goal is not to do *density estimation*, but to *model* how an input density representing the utility of the population gets *transformed* by an evaluation process to an output density in the presence of risk-averseness and information constraint.  in Section 2, our model modifies the classical maximum entropy framework (Equation 2) to incorporate the resource-information parameter $\tau$ (Equation 3) and the risk-averseness parameter $\alpha$ (Equations 4).
>
> We do use an $f$-divergence (differential entropy) in our formulation, albeit in the constraint as it allows us to incorporate the resource-information parameter $\tau$. However, methods like GMMs or kernel density estimation are not relevant to what we are trying to achieve.
>
> In Section 2, we will add further on the use of max-entropy as a constraint and also contrast our approach with density estimation.
>
>
> **"It seems the experiments are a density estimation task for two groups, why usual density estimation methods cannot be applied here?"** The empirical results are trying to find the values of the parameters $\alpha$ and $\tau$ in our model that best-fit the density of the biased values rather than estimating density itself, so the usual methods for density estimation do not seem useful.
>
>
> **"I am also not convinced the proposed method can be used to examine effect of interventions, since they are dynamic and more information is needed to see the effect of interventions in the long run."** As in many prior works [81,57,41], the proposed model can be used to understand the effect of interventions in one round. Indeed, to understand the effect of interventions in the long term, additional work would be required, perhaps evaluated as in [40], and would be an important direction for future work. We will add this as a limitation in the final version.
>
>
> **"The paper only considers binary sensitive groups, while in practice, sensitive groups are often overlapping with multiple attributes."** Indeed, we limit our discussion and simulations to binary-sensitive groups. However, our model easily extends to multiple sensitive groups by considering a group-specific risk-aversion parameter and a group-specific information constraint. Prior models of biased densities [81, 57] also limit their discussion to binary-sensitive groups, and similar extensions (with group-specific parameters) have also been proposed for them. For example, if two groups $G_1,G_2$ overlap, then we can consider three disjoint subgroups $G_1\cap G_2,$ $G_1\backslash G_2$ and $G_2 \backslash G_1$. We will add a remark in the final version.
>
>
> **"How would the estimation error and model misspecification lead to negative impact?"** Estimation errors and model misspecification can lead to errors in the assessments of interventions that possibly, in turn, leads to the deployment of suboptimal interventions. These errors are not specific to our evaluation method and one can try to assess and reduce their negative impact via validation methods such as train-test splits, control trials, and ablation studies. We will include a discussion about such errors in the final version.

---

> > ### Comment · Reviewer_7ABm · 2023-08-16
> >
> > Thank authors for the responses. I will keep my score.

---

### Official Review · Reviewer_G1nQ · 2023-07-07

**Soundness:** 3 good
**Presentation:** 2 fair
**Contribution:** 3 good
**Rating:** 6
**Confidence:** 2

**Summary:**

In this paper, the authors model evaluation processes that estimate the density of value for an individual (on a task) as a loss minimization problem subject to constraints. The authors proceed to derive various properties of the output densities of their model and evaluate it on two real world datasets.

**Strengths:**

- This is a good solution that seems to provide clarity to a difficult and important problem
- Strong (though limited) empirical section, and ablation studies on the effects of $\tau$ and $\alpha$

**Weaknesses:**

- I found it really hard to read this paper as someone not too familiar with the field, in particular I thought the intro could use some more clarity. There's a bit of measure theory  too that I wonder is necessary -
- Id like to see a theorem about the performance of this model relative to others, it seems like its fairly to compare it to some of the Related Work mentioned. Though this might be fixed by adding more clarifications --

**Questions:**

- So how would a policy maker use this model? It isnt immediately clear to me as someone unfamiliar with the literature

**Limitations:**

The authors note some limitations to their work - though I didn't see much discussion of the misuse of their model and it's negative societal impact.

---

> ### Author Rebuttal · Authors · 2023-08-10
>
> Thanks for your feedback. We are glad that you find our work as a good solution to an important and difficult problem. We address your specific questions below.
>
> **"hard to read this paper as someone not too familiar with the field"**  We apologize that you found it hard to read parts of the paper. Given the multitude of disciplines touched by this paper, it was difficult to write. We chose a style similar to some of the most related works [81,57,41]. We take your feedback seriously and will try to simplify some of the expressions while preparing the final version.
>
> **"...theorem about the performance of this model relative to others..."** Thanks for the suggestion. The two most-related prior models are the implicit variance model of [57] and the beta-bias model of [81]. In the implicit variance model, the observed utility of the advantaged group is a Gaussian random variable with mean $\mu$ and variance $\sigma_0^2$, and the observed utility of the disadvantaged group is a Gaussian random variable with mean $\mu$ and variance $\sigma_0^2 + \sigma^2$. This model is a special case of our framework  as captured by the following theorem (this is a direct corollary of Lemma F.2 and we will include it in the final version):
>
>
> **Theorem:** Consider an instance of the implicit variance model given by parameters $\mu, \sigma, \sigma_0$. Consider instances  $I_1=(\Omega, f_{\mathcal{G}}, \ell, \alpha, \tau_1)$ and $I_2=(\Omega, f_{\mathcal{G}}, \ell, \alpha, \tau_2)$ of OptProg, where  $\Omega=\mathbb{R}$, $f_{\mathcal{G}}$ is the normal density $N(\mu, \sigma_0^2)$, $\alpha=1$, $\ell(v,x) = (x-v)^2$, $\tau_1=\frac{1}{2}(1+\ln (2 \pi \sigma_0^2))$ and  $\tau_2 = \frac{1}{2}(1 + \ln(2 \pi (\sigma_0^2+ \sigma^2)))$. Then the output density of OptProg on $I_1$ is $N(\mu, \sigma_0^2)$ and the output density of OptProg on $I_2$ is $N(\mu, \sigma_0^2 + \sigma^2)$.
>
> In the $\beta$-bias model, the true utility of both groups is drawn from a Pareto distribution and the output utility for the disadvantaged group is obtained by scaling down the true utility by a factor $\beta > 1$. This changes the domain of the distribution to $[\frac{1}{\beta},\infty)$ from $[1,\infty)$ and, hence, does not fit exactly in our model which does not change the domain. Nevertheless, we can show that, for any fixed $\tau$,  increasing $\alpha$ reduces the *mean* of the output density, effectively capturing the $\beta$-bias model at a population level. Formally, the following theorem follows from the calculations in Section G; we will include it in the final version. Recall that the Pareto density with parameter $\gamma$ over $[1, \infty)$ is defined as $f_\gamma(x) = \frac{\gamma}{x^{\gamma+1}}, x \in [1, \infty).$
>
>
> **Theorem:** Consider an instance $\cal I$ of the $\beta$-bias model specified by the Pareto distribution $f_\gamma$ with parameter $\gamma$ over $[1, \infty)$ and a parameter $\beta > 1$. Consider the instance  $I_1 =  (\Omega=[1, \infty), f_\gamma, \ell, \alpha_1 = 1, \tau)$ of OptProg, where $\tau = 1 + \frac{1}{\gamma} - \ln \gamma$ and $\ell(x) = \ln (x/v).$  Then the optimal solution to $I_1$ is given by $f_\gamma$. Further, there exist values $\alpha_2 \geq 1$ and $\beta_\gamma > 1$, which is a function of $\gamma$ only, such that if $\beta \leq \beta_\gamma$, then the output density of OptProg on the instance $I_2 = (\Omega, f_\gamma, \ell, \alpha_2, \tau)$ has expectation $1/\beta$ times the expectation of $f_{\gamma}$.
>
>
> **"how would a policymaker use this model?"** Most of the prior works on interventions in selection settings have focused on adding representational constraints for disadvantaged groups. Such constraints, often framed as a form of affirmative action, could be beneficial but may not be possible to implement in certain contexts. E.g., in a landmark 2023 ruling, the US Supreme Court effectively prohibited the use of race-based affirmative action in college admissions. Our work, via a more refined model of how bias might arrive at a population level in evaluation processes, allows for evaluating additional interventions that focus on procedural fairness; this allows working towards diversity and equity goals without placing affirmative-action-like constraints.
>
> In this framing, we can consider decreasing either $\alpha$ or $\tau$. Improving either would work towards equity, but which ones to target or to what extent and via which method would be context dependent and vary in cost. A decrease in $\alpha$ can be achieved by reducing risk-averseness in the evaluation process; e.g. by investment in better facilities for disadvantaged groups, or making the evaluation process blind to group membership. A reduction in $\tau$ may follow by allocating additional resources to the evaluation process, e.g., by letting a candidate choose an evaluation in their native language. Our framework allows a policymaker to study these trade-offs, and we discuss specific examples in Supplementary Material H.
>
> **"I didn't see much discussion of the misuse of their model and it's negative societal impact."** In any work that focuses on debiasing, the ideas and models could be used adversarially to achieve the opposite goal. We need third-party evaluators, legal protections, and available recourse for affected parties -- crucial components of any system -- though beyond the scope  of this work. We will add a brief discussion to acknowledge these important points in the final version.

---

### Author Rebuttal · Authors · 2023-08-10

We thank the area chair for their time and effort in engaging with the reviewers and considering our rebuttal.

We thank all the reviewers for their excellent suggestions which will help improve the paper and for considering our rebuttal. We take the feedback of reviewers seriously and have addressed their specific questions and concerns in individual responses.

Based on suggestions by Reviewer gzsA, we give two responses below that may be of general interest. The first is an overview of the proof of Theorem 3.1 and the second is some specific examples of mechanisms by which information constraints and risk aversion lead to bias.

We also attach a pdf file that contains a new figure that we reference in response to Reviewer gzsA.

We will include all these changes in the final version.


**1. Overview of the proof of Theorem 3.1.**


(i) We first consider the dual of OptProg and show that strong duality holds (see Sections B.2 and B.3). The proof of strong duality (Theorem B.3) uses Slater's condition. This step allows us to derive the form of the optimal density to OptProg and is standard (e.g., in formulations that maximize entropy subject to additional constraints).

(ii) The next step is to show that the optimal solution (density) of OptProg exists and is unique. This requires proving that the dual variable $\gamma^\star$ (corresponding to the entropy constraint) is *positive* -- while this variable is always nonnegative, the challenge is to show that it is *nonzero*.  There are instances to OptProg where $\gamma^\star$ is zero, and an optimal solution does not exist (or an optimal solution exists, but it is not unique). Proving that $\gamma^\star$ is nonzero under general conditions is the main technical difficulty.

(iii) The next step is reducing the proof of $\gamma^\star \neq 0$ to understanding the properties of the following integral that captures the expected loss when the estimated utility is $x$ (see lines 1026--1027): $I(x) = \int_{\Omega} \ell_\alpha(x,v) f_{\cal D}(v) dv.$ The objective function of OptProg is equal to the expectation of $I(x)$ over the output density $f^\star(x)$.

(iv) In section B.4, we show that $I(x)$ can be expressed as a sum of two monotone functions (see Theorem B.10). This decomposition allows us to show that the optimal value of OptProg is finite and $I(x)$ has a global minimizer $x^\star$ (Claim B.12).

(v) In Section B.5, we show that the optimal value of OptProg is strictly larger than $I(x^\star)$ (Lemma B.13, B.14). This requires us to understand the interplay between  the growth rate of the expected loss function and the entropy of a density as we place probability mass away from $x^\star$.

(vi) Finally, in Theorem B.15, we show that $\gamma^\star$ is nonzero. This follows from the fact that if $\gamma^\star=0$, then the optimal value of OptProg is equal to $I(x^\star)$, which contradicts the claim in (v) above. Once we show $\gamma^\star > 0$, the expression for the (unique) optimal solution follows from equation (13) in line  1128.

To summarize, the technical difficulty in proving Theorem 3.1 lies in showing $\gamma^\star \neq 0$ and the conceptual difficulty lies in formulating general conditions under which this property holds. Proving that these conditions hold for a general class of loss functions is non-trivial. We carry out these steps for Gaussian, Pareto, Exponential, and Laplace densities in Sections F, G, I, and J respectively.

**2. Specific examples of mechanisms by which information constraints and risk aversion lead to bias.**

One context is college admissions. It is well known that SAT scores are implicitly correlated with income (and, hence, test preparation) in addition to student ability ("Is Income Implicit in Measures of Student Ability?", Budget Model, Penn Wharton). While the true ability may be $v$, the score is skewed depending on amount/quality of test preparation, which depends on socio-economic status. The parameter $\alpha$ in our model can be used to encode this. As for $\tau$, while an evaluator may know what a GPA means at certain universities well known to them, they may not understand what GPA means for students from  lesser-known school to them. This lack of knowledge can be overcome, but takes effort/time, and without effort entrenches the status quo.


Another example is evaluation of candidates using a standardized test.  In time-constrained settings, high  value of the resource-information parameter $\tau$ for the disadvantaged group indicates that such candidates may not be able to comprehend a question as well as someone from an advantaged group. This could be due to various factors including less familiarity with the language used in the test or the pattern of questions, as opposed to someone who had the resources to invest in a training program for the test.  Similarly, a high value of the risk-averseness parameter captures that an evaluator, when faced between a choice of awarding low or high marks to an answer given by a candidate from the disadvantaged group, is less likely to give high marks. More concretely, suppose there are several questions in a test, where each question is graded either 0 or 1. Assume that a candidate has true utility $v \in [0,1]$, and hence, would have received expected score $v$ for each of the questions if one were allowed to award grades in the continuous range $[0,1]$. However, the fact that the true scores have to be rounded to either 0 or 1 can create a bias for the disdvantaged group. Indeed, the probability that an evaluator rounds such an answer to 1 may be less than $v$ -- the risk-averseness parameter measures the extent to which this probability gets scaled down.

---

### Decision · Program_Chairs · 2023-09-21

**Decision:**

Accept (poster)

**Comment:**

This work proposes a new optimization-based approach to modeling bias in evaluations. Reviewers have raised several important concerns on the lack of clarity, and the authors present a detailed plan to address them. The proposed model is an interesting contribution, and the paper's impact can be maximized by putting further work on the exposition. I would like to echo Reviewers gzsA's comments on expanding the discussion on implications for policy-making.